# Row-Stochastic Matrices Can Provably Outperform Doubly Stochastic Matrices in Decentralized Learning

**Bing Liu** [1]  **Boao Kong** [2]  **Limin Lu** [1]  **Kun Yuan** [3]  **Chengcheng Zhao** [1]

## Abstract

Decentralized learning often involves a weighted global loss with heterogeneous node weights $\lambda$. We revisit two natural strategies for incorporating these weights: (i) embedding them into the local losses to retain a uniform weight (and thus a doubly stochastic matrix), and (ii) keeping the original losses while employing a $\lambda$-induced row-stochastic matrix. Although prior work shows that both strategies target the same $\lambda$-weighted global loss, it remains unclear whether the Euclidean-space guarantees are tight and what fundamentally differentiates their behaviors. To clarify this, we develop a weighted Hilbert-space framework $L^2(\lambda; \mathbb{R}^d)$ and obtain convergence rates that are strictly tighter than those from standard Euclidean analysis. In this geometry, the row-stochastic matrix becomes *self-adjoint* whereas the doubly stochastic one does not, creating additional *penalty terms* that amplify consensus error, thereby slowing convergence. Consequently, the difference in convergence arises not only from spectral gaps but also from these penalty terms. We then derive sufficient conditions under which the row-stochastic design converges faster even with a smaller spectral gap. Finally, by using a Rayleigh-quotient and Loewner-order eigenvalue comparison, we further obtain topology conditions that guarantee this advantage and yield practical topology-design guidelines.

## 1. Introduction

The ever-increasing scale of data and models has made distributed learning a central paradigm for large-scale op-

timization. Among its variants, decentralized learning has attracted growing attention for its robustness to single-point failures, low communication overhead, and strong scalability. Most existing analyses, however, assume uniform node weights and symmetric, doubly stochastic mixing (Lian et al., 2017; Koloskova et al., 2020). In practical systems, differences in data volumes and distributions often lead to heterogeneous node weights. Therefore, following (McMahan et al., 2017; Yuan et al., 2018a; Zhu et al., 2025), we study decentralized learning over a network of $n$ nodes with *prescribed heterogeneous* weights $\lambda = [\lambda_1, \ldots, \lambda_n]^\top$, which are fixed throughout optimization. Each node $i$ accesses its local data distribution $\mathcal{D}_i$ and collaboratively solves the weighted optimization problem

$$\min_{\theta \in \mathbb{R}^d} F(\theta) = \frac{1}{n} \sum_{i=1}^n \lambda_i \left[ F_i(\theta) := \mathbb{E}_{\xi_{i,j} \sim \mathcal{D}_i} \left[ f_i(\theta, \xi_{i,j}) \right] \right],$$
(1)

where $\lambda_i > 0$ and $\sum_{i=1}^n \lambda_i = n$, and $f_i(\theta, \xi_{i,j})$ denotes the instantaneous loss of sample $\xi_{i,j}$ evaluated at $\theta$.

Two natural decentralized designs arise to solve problem (1).

- **Strategy I** absorbs the weights into the local losses, i.e., each $F_i$ is replaced by $\lambda_i F_i$. This transformation recovers uniform node weights $(1/n)$ and leads to the standard framework with a doubly stochastic mixing.

- **Strategy II**, in contrast, keeps the original losses and incorporates $\lambda$ into a row-stochastic mixing matrix whose stationary distribution equals $\lambda/n$.

Existing analyses (Sayed, 2014; Ying & Sayed, 2016) show that both strategies converge to stationary points of the same weighted optimization problem in (1). However, several fundamental questions remain open:

(Q1) *Does the convergence rate obtained under the standard Euclidean framework remain tight under heterogeneous node weights?*

(Q2) *Do the differences between the two strategies arise solely from the spectral gaps of their mixing matrices, or also from other key factors?*

[1]College of Control Science and Engineering, Zhejiang University [2]Center for Data Science, Peking University [3]Center for Machine Learning Research, Peking University. Correspondence to: Chengcheng Zhao <chengchengzhao@zju.edu.cn>.

*Proceedings of the 43$^{rd}$ International Conference on Machine Learning*, Seoul, South Korea. PMLR 306, 2026. Copyright 2026 by the author(s).

(Q3) *Given a weight vector $\lambda$, under what conditions should we prefer one strategy over the other?*

**Main contributions.** This work advances the understanding of decentralized learning with heterogeneous weights by addressing the open questions above. Our main contributions are summarized as follows:

(C1) We develop a new analytical framework built upon a weighted Hilbert space $L^2(\lambda; \mathbb{R}^d)$. Because of the heterogeneous node weights, the error recursions of both strategies naturally reside in this weighted space rather than in the standard Euclidean one (Appendix C.2). Within this framework, we derive tight convergence rates for decentralized stochastic gradient tracking under both strategies and show that the Euclidean analysis leads to strictly looser bounds.

(C2) Within the $L^2(\lambda; \mathbb{R}^d)$ space, the $\lambda$-induced row-stochastic matrix becomes self-adjoint, whereas the doubly stochastic matrix does not. This lack of self-adjointness generates additional *penalty terms* in the convergence rates, increasing the consensus error and tightening the admissible step-size range of Strategy I. Consequently, the difference between the two strategies is determined not only by their spectral gaps but also by these penalty terms. Hence, Strategy II can converge faster even when its spectral gap is smaller.

(C3) From the derived convergence rates, we first obtain spectral-gap conditions under which Strategy II converges strictly faster than Strategy I. We then use Rayleigh-quotient and Loewner-order arguments to express these conditions as degree–weight constraints on the topology. These constraints further yield simple topology-design guidelines: node degrees should scale proportionally with their associated weights.

## 2. Related Works

**Decentralized learning with uniform weights.** Early works in optimization studied decentralized gradient descent with diminishing step sizes, time-varying directed graphs, and later fixed-step convergence guarantees (Nedic & Ozdaglar, 2009; Nedić & Olshevsky, 2014; Yuan et al., 2016). In machine learning, Lian et al. (2017) initiated the study of decentralized stochastic gradient descent (SGD), showing linear speedup comparable to centralized SGD while substantially reducing communication. Subsequent research improves robustness to data heterogeneity via gradient tracking and related techniques (Pu & Nedić, 2021; Yuan et al., 2023), reduces communication through efficient protocols (Tang et al., 2018; Koloskova et al., 2019), and extends decentralized learning to broader settings, including bilevel optimization (Zhu et al., 2024; Kong et al., 2025) and

Markovian sampling settings (Johansson et al., 2010; Sun et al., 2023). Nevertheless, most existing analyses assume *uniform* node weights and thus rely on doubly stochastic mixing matrices, leaving the heterogeneous-weight setting relatively less understood.

**Decentralized learning with heterogeneous weights.** In federated learning, heterogeneous weights are standard (McMahan et al., 2017; Li et al., 2020) because a server can directly implement weighted aggregation. In decentralized settings, however, the lack of a global server makes weighting more challenging, and only a few works treat it directly (Chen & Sayed, 2013; Yuan et al., 2018b;c; Cyffers et al., 2024; Zhu et al., 2025). From them, Cyffers et al. (2024) encode weights via the stationary distribution of random walks to strengthen differential privacy, but the convergence analysis remains in a standard (uniform-weight) framework. Zhu et al. (2025) introduce a *Data Influence Cascade* metric to quantify node impact, without rate analysis. The diffusion framework (Sayed, 2014; Chen & Sayed, 2013; 2015) handles heterogeneous node weights either via row-stochastic mixing or by folding the weights into the local losses, reducing the problem to uniform weights. Exact diffusion (Yuan et al., 2018b;c) further removes the bias introduced by left-stochastic mixing and constant step sizes through additional communication and computation. Both diffusion and exact diffusion are developed for strongly convex losses. In contrast, we study when *row-stochastic* mixing with heterogeneous weights can outperform doubly stochastic mixing *without extra steps*, and we introduce a new framework for comparing their convergence rates.

**Column-/row-stochastic mixing under uniform weights.** On directed graphs, nodes may only know in-/out-degrees, enabling only column-/row-stochastic matrices and introducing bias. For column-stochastic mixing, the bias and rates have been extensively characterized (Nedić & Olshevsky, 2014; Xi & Khan, 2017; Assran et al., 2019; Liang et al., 2025b), with Liang et al. (2025b) giving effective metrics, tight bounds, and showing that multiple communication rounds can mitigate asymmetry and recover optimal rates. For row-stochastic mixing, most prior results focus on deterministic, strongly convex problems (Mai & Abed, 2016; Xin et al., 2019a;b; Ghaderyan et al., 2023). In the stochastic, non-convex setting, Liang et al. (2025a) propose effective metrics, establish a lower bound, and show with multiple communication rounds the upper bound matches the lower bound up to logarithmic factors, yielding near-optimal rates.

## 3. Preliminaries

### 3.1. Notations

Let $\mathbb{R}^d$ denote the $d$-dimensional Euclidean space and $\mathbb{R}^{m \times d}$ the set of real $m \times d$ matrices. For a vector $x$, $\|x\|_2$

and $\langle x, y \rangle$ denote its Euclidean norm and standard inner product, respectively. Let $\mathbf{1}$ denote the all-ones vector and $I$ denote the identity matrix. For a matrix $M$, let $M_{i,j}$, $M^\top$, $\|M\|_2$, and $\|M\|_F$ denote its $(i,j)$-th entry, transpose, spectral norm, and Frobenius norm. A matrix $M$ is *row-stochastic* if $M_{i,j} \geq 0$ and $\sum_j M_{i,j} = 1$ for all $i$; if both $M$ and $M^\top$ are row-stochastic, it is *doubly stochastic*. For a square matrix $W \in \mathbb{R}^{n \times n}$, let $\sigma(W) = \{\sigma_i(W)\}_{i=1}^m$ denote its eigenvalues in non-increasing order, and define the spectral radius as $\rho(W) = \max_i |\sigma_i(W)|$. When $W$ is row-stochastic, $\rho(W) = 1$, and its spectral gap is defined as $1 - \max\{|\sigma_2(W)|, |\sigma_n(W)|\}$. We define the filtration $\mathcal{F}^{(t)}$ as the filtration **before** the stochastic gradient evaluation in the $t$-th step. We use $a \lesssim_d b$ to indicate that there exists a $C \geq 0$ that is independent with $d$ such that $a \leq Cb$.

### 3.2. Markov Chains

Consider a finite Markov chain with row-stochastic transition matrix $P$.

**Irreducibility.** The chain is *irreducible* if any two states communicate, i.e., there exists $t \geq 0$ such that $\Pr\{X_t = j \mid X_0 = i\} > 0$.

**Aperiodicity.** The period of a state $i$ is the largest common divisor of $\{t : \Pr\{X_t = i \mid X_0 = i\} > 0\}$. A chain is *aperiodic* if all states have period one.

**Stationary distribution.** If the chain is irreducible and aperiodic, there exists a unique stochastic row vector $\pi$ such that $\pi\mathbf{1} = 1$ $\pi P = \pi$. The equality $\pi_i = \sum_{j=1}^n \pi_j P_{j,i}$ expresses the balance of probability flows, and the stronger *detailed balance* condition requires $\pi_i P_{i,j} = \pi_j P_{j,i}, \; \forall i,j$. Moreover, $P^t \to \mathbf{1}\pi$ as $t \to \infty$, and the convergence rate is governed by the spectral gap of $P$.

## 4. Algorithm Overview

We consider the decentralized optimization problem in (1) and employ *decentralized stochastic gradient tracking (GT)* to mitigate data heterogeneity. Our framework follows the standard GT structure (Xu et al., 2017), with two lightweight extensions to incorporate heterogeneous node weights. We consider learning over an undirected network $\mathcal{G} = (\mathcal{V}, \mathcal{E})$, where $\mathcal{V} = \{1, \ldots, n\}$ and $\mathcal{E}$ denotes the edge set; each node $i$ communicates only with its neighbors $\mathcal{N}_i = \{j : (i,j) \in \mathcal{E}\}$. For brevity, we denote $g_i^{(t)} := \nabla f_i(\theta_i^{(t)}; \xi_i^{(t)})$. The overall procedure is summarized in Algorithm 1.

Let $\Theta^{(t)} = [\theta_1^{(t)}, \ldots, \theta_n^{(t)}]^\top$, $Y^{(t)} = [y_1^{(t)}, \ldots, y_n^{(t)}]^\top$, and $\nabla F_\xi^{(t)} = [g_1^{(t)}, \ldots, g_n^{(t)}]^\top$. The algorithm iterates as

$$\begin{aligned} \Theta^{(t+1)} &= W_\lambda[\Theta^{(t)} - \alpha Y^{(t)}], \\ Y^{(t+1)} &= W_\lambda Y^{(t)} + G_\lambda[\nabla F_\xi^{(t+1)} - \nabla F_\xi^{(t)}], \end{aligned} \quad (2)$$

---

**Algorithm 1** Weighted Decentralized Gradient Tracking

1: **Input:** step size $\alpha$, batch sizes $\{b_i\}$, total iterations $T$, and system matrices $W_\lambda, G_\lambda$.
2: **Initialize:** initial states $\Theta^{(0)}$; $y_i^{(0)} = g_i^{(0)}$ for Strategy II, or $y_i^{(0)} = \lambda_i g_i^{(0)}$ for Strategy I.
3: **for** $t = 0$ **to** $T - 1$ **do**
4:     **for** each node $i \in \mathcal{V}$ **do**
5:         Receive $\{\theta_j^{(t)}, y_j^{(t)}\}_{j \in \mathcal{N}_i}$ from neighbors.
6:         **Model update:**
$$\theta_i^{(t+1)} = \sum_{j=1}^n [W_\lambda]_{i,j}(\theta_j^{(t)} - \alpha y_j^{(t)}).$$
7:         **Tracker update:**
$$y_i^{(t+1)} = \sum_{j=1}^n [W_\lambda]_{i,j} y_j^{(t)} + [G_\lambda]_{i,i}(g_i^{(t+1)} - g_i^{(t)}).$$
8:     **end for**
9: **end for**

---

where $(W_\lambda, G_\lambda)$ encode different weighting mechanisms:

### Strategy I (Weighted loss, uniform mixing):

$(W_\lambda, G_\lambda) = (W^{\mathrm{ds}}, D_\lambda)$, where $W^{\mathrm{ds}}$ is symmetric and doubly stochastic, and $D_\lambda = \mathrm{diag}(\lambda_1, \ldots, \lambda_n)$.

### Strategy II (Uniform loss, weighted mixing):

$(W_\lambda, G_\lambda) = (W, I)$, where $W$ is row-stochastic with $\frac{\lambda^\top}{n}$ being its stationary distribution, i.e., $\frac{\lambda^\top}{n} W = \frac{\lambda^\top}{n}$.

### 4.1. Aggregate Dynamics under the Weighted Loss

Under the initialization in Algorithm 1, the gradient-tracking property ensures that $\frac{\lambda^\top}{n} Y^{(t)} = \frac{1}{n} \sum_{i=1}^n \lambda_i g_i^{(t)}$ for Strategy II, and $\frac{\mathbf{1}^\top}{n} Y^{(t)} = \frac{1}{n} \sum_{i=1}^n \lambda_i g_i^{(t)}$ for Strategy I. Therefore, although the two strategies encode the weights differently, both induce aggregate recursion dynamics that are aligned with the same weighted loss in (1). We present the details for both strategies below.

### Strategy I: Weighted local losses.

Each node scales its loss as $F_i'(\theta) = \lambda_i F_i(\theta)$, and uses a doubly stochastic matrix $W^{\mathrm{ds}}$. The global average $\bar\theta^{(t)} = \frac{1}{n} \sum_i \theta_i^{(t)}$ evolves as

$$\bar\theta^{(t+1)} = \frac{\mathbf{1}^\top}{n} W^{\mathrm{ds}}[\Theta^{(t)} - \alpha Y^{(t)}] = \bar\theta^{(t)} - \frac{\alpha}{n} \sum_{i=1}^n \lambda_i g_i^{(t)},$$

which matches the weighting structure of the optimization problem in (1).

### Strategy II: Weighted mixing matrix.

Instead of modifying the local losses, we embed $\lambda_i$ in a row-stochastic $W$ such that $\frac{\lambda^\top}{n} W = \frac{\lambda^\top}{n}$. Defining the weighted

average $\bar{\theta}_\lambda^{(t)} = \frac{1}{n} \sum_i \lambda_i \theta_i^{(t)}$, we have

$$\bar{\theta}_\lambda^{(t+1)} = \frac{\lambda^\top}{n} W[\Theta^{(t)} - \alpha Y^{(t)}] = \bar{\theta}_\lambda^{(t)} - \frac{\alpha}{n} \sum_{i=1}^n \lambda_i g_i^{(t)},$$

which also matches the weighting structure of the optimization problem in (1).

We are interested in whether the two strategies, despite sharing the same weighting structure, exhibit the same error recursion and convergence behavior, and what fundamentally distinguishes them. We address this question by introducing a new analytical framework and deriving the corresponding convergence rates for both strategies.

# 5. The $L^2(\lambda; \mathbb{R}^d)$-Hilbert Space

This section gives a decentralized construction of a row-stochastic matrix $W$ whose stationary distribution equals $\lambda/n$. We then develop the $\lambda$-weighted geometry that underlies our analysis and the spectral identities used later.

## 5.1. Constructing $W$ from $\lambda$

The weighting vector is naturally encoded by the stationary distribution. Our goal is thus a decentralized row-stochastic $W$ whose stationary distribution matches $\lambda/n$.

We adopt a modified Metropolis–Hastings (MH) rule (Chib & Greenberg, 1995) to construct a transition matrix $P$ with stationary distribution $\lambda/n$:

$$P_{i,j} = \begin{cases} \frac{1}{d_i} \min\left(1, \frac{\lambda_j d_i}{\lambda_i d_j}\right), & \text{if } i \neq j \text{ and } j \in \mathcal{N}_i, \\ 1 - \sum_{k \in \mathcal{N}_i} P_{i,k}, & \text{if } i = j, \\ 0, & \text{otherwise.} \end{cases}$$

To preclude oscillations and ensure aperiodicity, we introduce a *laziness* parameter $\varepsilon \in (0,1)$:

$$W \leftarrow (1 - \varepsilon)P + \varepsilon I.$$

The resulting entries are

$$W_{i,j} = \begin{cases} \frac{1-\varepsilon}{d_i} \min\left(1, \frac{\lambda_j d_i}{\lambda_i d_j}\right), & \text{if } i \neq j \text{ and } j \in \mathcal{N}_i, \\ 1 - \sum_{k \in \mathcal{N}_i} W_{i,k}, & \text{if } i = j, \\ 0, & \text{otherwise.} \end{cases} \quad (3)$$

**Implementation overhead.** The construction in (3) is fully distributed on undirected graphs: each node only exchanges its prescribed weight $\lambda_i$ and degree $d_i$ with its neighbors. The positive diagonal prevents periodicity, and setting $\lambda \equiv \mathbf{1}$ recovers the standard doubly stochastic matrix $W^{\text{ds}}$. Compared with the standard MH construction for doubly stochastic matrices, it requires only one additional scalar

weight per neighbor and a constant number of extra scalar computation/communication per edge. Thus, the $\lambda$-induced row-stochastic construction has nearly the same practical feasibility as the doubly stochastic one.

## 5.2. $L^2(\lambda; \mathbb{R}^d)$-Hilbert Space

Heterogeneous node weights induce a weighted Euclidean geometry (cf. Appendix C.2). We therefore introduce the vector-valued Hilbert space $L^2(\lambda; \mathbb{R}^d)$ endowed with the inner product

$$\langle X, Y \rangle_{\lambda,d} = \sum_{i=1}^n \lambda_i \langle x_i, y_i \rangle, \quad X, Y \in (\mathbb{R}^d)^n.$$

The induced weighted Frobenius norm is $\|X\|_{\lambda,F} = (\sum_i \lambda_i \|x_i\|^2)^{1/2}$, and for the matrix (linear map) $W$, we use the corresponding weighted spectral norm $\|W\|_\lambda = \max_{\|X\|_{\lambda,F}=1} \|WX\|_{\lambda,F}$. Equivalently, $L^2(\lambda; \mathbb{R}^d) \cong L^2(\lambda) \widehat{\otimes} \mathbb{R}^d$. Under the construction (3), $W$ is self-adjoint in this space, admits an orthogonal eigen-decomposition, and allows for standard spectral analysis.

**Spectral relationships.** For the symmetric, doubly stochastic case $W^{\text{ds}}$, let $J = \frac{\mathbf{1}\mathbf{1}^\top}{n}$ denote the projection onto its stationary distribution. Then

$$\|W^{\text{ds}} - J\|_2 = \rho(W^{\text{ds}} - J) = \rho(W_J) := \rho_J,$$

where $\rho_J$ is the second-largest eigenvalue magnitude of the matrix $W^{\text{ds}}$.

For the $\lambda$-induced row-stochastic matrix $W$, its $\lambda$-weighted spectral norm satisfies

$$\|W - \Lambda\|_\lambda = \|W_\Lambda\|_\lambda = \|D_\lambda^{1/2} W_\Lambda D_\lambda^{-1/2}\|_2 = \|\widetilde{W}_\Lambda\|_2,$$

where $\Lambda = \frac{\mathbf{1}\lambda^\top}{n}$ and $\widetilde{W}_\Lambda$ is a symmetric similarity transform of $W_\Lambda$ that preserves its eigenvalues. The detailed balance condition ensures $\widetilde{W}_\Lambda = \widetilde{W}_\Lambda^\top$ (Lemma C.4), implying

$$\|W - \Lambda\|_\lambda = \rho(\widetilde{W}_\Lambda) = \rho(W_\Lambda) := \rho_\Lambda.$$

Hence, the $\lambda$-weighted spectral norm coincides with $\rho_\Lambda$, i.e., the second-largest eigenvalue magnitude of $W$. We also consider the $\lambda$-weighted spectral norm of $W^{\text{ds}} - J$:

$$\|W_J\|_\lambda = \|D_\lambda^{1/2} W_J D_\lambda^{-1/2}\|_2 \leq \sqrt{\frac{\lambda_{\max}}{\lambda_{\min}}} \rho_J := \kappa_\lambda \rho_J.$$

Here, $\kappa_\lambda > 1$ quantifies the metric distortion induced by the heterogeneous weights. Since a doubly stochastic matrix is generally non-self-adjoint in $L^2(\lambda; \mathbb{R}^d)$, the resulting non-normality introduces a multiplicative penalty greater than one. Further details on the weighted Hilbert space are provided in Appendix B.

## 6. Convergence Rate for Algorithm 1

In this section, we present our convergence rate for Algorithm 1 with different communication strategies.

### 6.1. Assumptions and Descent Lemma

To begin with, we present the following assumptions, which are commonly used in existing literature.

**Assumption 6.1** ($\beta$-smoothness). Each local loss $F_i(\theta)$ is differentiable and $\beta$-smooth, i.e.,

$$\|\nabla F_i(\theta_1) - \nabla F_i(\theta_2)\| \le \beta\|\theta_1 - \theta_2\|, \quad \forall \theta_1, \theta_2 \in \mathbb{R}^d.$$

**Assumption 6.2** (Gradient oracles). For each node $i \in \mathcal{V}$ and all $\theta \in \mathbb{R}^d$, it holds that:

$$\mathbb{E}[\nabla f_i(\theta; \xi)] = \nabla F_i(\theta), \ \mathbb{E}[\|\nabla f_i(\theta; \xi) - \nabla F_i(\theta)\|^2] \le v^2.$$

**Assumption 6.3** (Connected graph). The communication graph $\mathcal{G}$ is connected. Hence, the matrices $W$ and $W^{\mathrm{ds}}$ constructed via (3) are irreducible and aperiodic, and thus have positive spectral gaps: $\rho_\Lambda < 1$ and $\rho_J < 1$.

We now explain why the subsequent analysis is naturally carried out in the weighted Hilbert space $L^2(\lambda; \mathbb{R}^d)$. Under Assumptions 6.1–6.2 and a step size $\alpha \le 1/\beta$, the descent lemma for Algorithm 1 yields (See proof in Appendix C.2.):

$$\mathbb{E}[F(\bar{\theta}_*^{(t+1)})] \le \mathbb{E}[F(\bar{\theta}_*^{(t)})] - \frac{\alpha}{2}\mathbb{E}[\|\nabla F(\bar{\theta}_*^{(t)})\|^2]$$
$$+ \frac{\alpha\beta^2}{2n}\mathbb{E}[\|(I - M_*)\Theta^{(t)}\|_{F,\lambda}^2] + \frac{\alpha^2 c_\lambda \beta v^2}{2},$$

where $c_\lambda = n^{-2}\sum_{i=1}^n \lambda_i^2 < 1$, and $(\bar{\theta}_*^{(t)}, M_*)$ equals $(\bar{\theta}^{(t)}, J)$ in Strategy I and $(\bar{\theta}_\lambda^{(t)}, \Lambda)$ in Strategy II.

The key observation is that the consensus error enters the inequality solely through the weighted term $\|(I - M_*)\Theta^{(t)}\|_{F,\lambda}^2$. This term is exactly the squared norm in $L^2(\lambda; \mathbb{R}^d)$, indicating that the geometry induced by the heterogeneous node weights is governed by this space. Consequently, $L^2(\lambda; \mathbb{R}^d)$ provides the appropriate functional setting for analyzing consensus under heterogeneous weights.

### 6.2. Consensus Error

In this subsection, we analyze the parameter and tracker consensus errors across all nodes and introduce the following notation to jointly characterize them for both strategies:

$$E_{\mathrm{I}}^{(t)} = \begin{bmatrix} (I - J)\Theta^{(t)} \\ \alpha(I - J)Y^{(t)} \end{bmatrix}, \qquad E_{\mathrm{II}}^{(t)} = \begin{bmatrix} (I - \Lambda)\Theta^{(t)} \\ \alpha(I - \Lambda)Y^{(t)} \end{bmatrix}.$$

Building on (Koloskova et al., 2021), we derive closed-form spectral norms for the matrices in the recursion, enabling direct computation of the consensus error at any iteration $t$ without the window-averaging technique used in (Koloskova et al., 2021). This yields a simpler analysis and more transparent accumulated-error bounds:

**Proposition 6.4.** *Under Assumptions 6.1–6.3, the following accumulated consensus error bounds hold (See proof in Appendix C.3.):*

*(Strategy I).* If $\alpha < \frac{1}{2\lambda_{\max}\beta}\sqrt{\frac{1}{15B(\rho_J)}}$, then

$$\sum_{t=0}^{T-1} \mathbb{E}\left[\left\|E_{\mathrm{I}}^{(t)}\right\|_{F,\lambda}^2\right] \tag{4}$$
$$\le \left(\frac{\kappa_\lambda^2}{C_1'}\right) \frac{6(4\rho_J^4 - 5\rho_J^2 + 3)}{(1 - \rho_J^2)^3} \left\|E_{\mathrm{I}}^{(0)}\right\|_{F,\lambda}^2$$
$$+ \left(\frac{\lambda_{\max}^2}{C_1'}\right) 2n\alpha^2 B(\rho_J) \sum_{j=0}^{T-1} \mathbb{E}[\|\nabla F(\bar{\theta}^{(t)})\|^2]$$
$$+ \left(\frac{\lambda_{\max}^2}{C_1'}\right) 18n\alpha^2 v^2 T\left(A(\rho_J) + \frac{3}{2}c_\lambda\alpha^2\beta^2 B(\rho_J)\right),$$

*where* $A(\rho_J) := \frac{1+\rho_J^2}{(1-\rho_J^2)^3}$, $B(\rho_J) := \frac{2(1+3\rho_J^4)}{(1-\rho_J^2)^3(1-\rho_J)}$, $c_\lambda = \frac{1}{n^2}\sum_{i=1}^n \lambda_i^2 < 1$, *and* $C_1' = 1 - 60\lambda_{\max}^2\alpha^2\beta^2 B(\rho_J) > 0$.

*(Strategy II).* If $\alpha < \frac{1}{2\beta}\sqrt{\frac{1}{15B(\rho_\Lambda)}}$, then

$$\sum_{t=0}^{T-1} \mathbb{E}\left[\left\|E_{\mathrm{II}}^{(t)}\right\|_{F,\lambda}^2\right] \tag{5}$$
$$\le \frac{1}{C_1} \frac{6(4\rho_\Lambda^4 - 5\rho_\Lambda^2 + 3)}{(1 - \rho_\Lambda^2)^3} \left\|E_{\mathrm{II}}^{(0)}\right\|_{F,\lambda}^2$$
$$+ \frac{1}{C_1} 2n\alpha^2 B(\rho_\Lambda) \sum_{j=0}^{T-1} \mathbb{E}[\|\nabla F(\bar{\theta}_\lambda^{(t)})\|^2]$$
$$+ \frac{1}{C_1} 18n\alpha^2 v^2 \left(A(\rho_\Lambda) + \frac{3}{2}c_\lambda\alpha^2\beta^2 B(\rho_\Lambda)\right) T,$$

*where* $A(\rho_\Lambda) := \frac{1+\rho_\Lambda^2}{(1-\rho_\Lambda^2)^3}$, $B(\rho_\Lambda) := \frac{2(1+3\rho_\Lambda^4)}{(1-\rho_\Lambda^2)^3(1-\rho_\Lambda)}$, $c_\lambda = \frac{1}{n^2}\sum_{i=1}^n \lambda_i^2 < 1$, *and* $C_1 = 1 - 60\alpha^2\beta^2 B(\rho_\Lambda) > 0$.

We observe that Strategy I introduces several multiplicative constants exceeding unity, notably $\boldsymbol{\kappa_\lambda^2}$ and $\boldsymbol{\lambda_{\max}^2}$, which are absent in Strategy II. The constant $\kappa_\lambda^2$ originates from the non-self-adjoint property of $W^{\mathrm{ds}}$, while the $\lambda_{\max}^2$ factor emerges from the combined effect of the gradient-related matrix $D_\lambda$ and the non-self-adjoint nature of $W^{\mathrm{ds}}$, collectively amplifying the consensus error. For comparative analysis, we present the corresponding spectral norms below (See proof in Proposition C.14): $\|A_{\mathrm{I}}^t\|_\lambda = \frac{t+\sqrt{t^2+4}}{2}\kappa_\lambda\rho_J^t$, $\quad \|A_{\mathrm{II}}^t\|_\lambda = \frac{t+\sqrt{t^2+4}}{2}\rho_\Lambda^t$, and
$$\|W^{\mathrm{ds}}(I - J)D_\lambda\|_\lambda^2 \le \lambda_{\max}^2\rho_J^2, \qquad \|W(I - \Lambda)\|_\lambda^2 \le \rho_\Lambda^2.$$

Thus, under comparable spectral gaps ($\rho_J = \rho_\Lambda$), **Strategy II** yields strictly smaller consensus error.

### 6.3. Convergence rate

With the consensus error analysis, we now derive the convergence rates of Algorithm 1 under both strategies, sum-

marized in the following theorem.

**Theorem 6.5.** *Under Assumptions 6.1–6.3, the following convergence rates hold (See proof in Appendix C.4.):*

(Strategy I). *If $\alpha < \frac{1}{\lambda_{\max}\beta}\sqrt{\frac{1}{62B(\rho_J)}}$, then*

$$\frac{1}{T}\sum_{t=0}^{T-1}\mathbb{E}\left[\|\nabla F(\overline{\theta}^{(t)})\|^2\right] \tag{6}$$

$$\leq \left(\frac{1}{C_2'}\right)\frac{2[F(\overline{\theta}^{(0)}) - F(\theta^\star)]}{\alpha T} + \left(\frac{1}{C_2'}\right)\alpha c_\lambda \beta v^2$$

$$+ \left(\frac{\kappa_\lambda^2}{C_1'C_2'}\right)\frac{6(4\rho_J^4 - 5\rho_J^2 + 3)\beta^2}{(1-\rho_J^2)^3 nT}\left\|E_{\mathrm{I}}^{(0)}\right\|_{F,\lambda}^2$$

$$+ \left(\frac{\lambda_{\max}^2}{C_1'C_2'}\right)18\alpha^2\beta^2 v^2\left(A(\rho_J) + \frac{3}{2}c_\lambda\alpha^2\beta^2 B(\rho_J)\right),$$

*where $C_2' = 1 - \frac{2\lambda_{\max}^2\alpha^2\beta^2 B(\rho_J)}{C_1'} > 0$.*

(Strategy II). *If $\alpha < \frac{1}{\beta}\sqrt{\frac{1}{62B(\rho_\Lambda)}}$, then*

$$\frac{1}{T}\sum_{t=0}^{T-1}\mathbb{E}\left[\|\nabla F(\overline{\theta}_\lambda^{(t)})\|^2\right] \tag{7}$$

$$\leq \frac{1}{C_2}\frac{2[F(\overline{\theta}_\lambda^{(0)}) - F(\theta^\star)]}{\alpha T} + \frac{1}{C_2}\alpha c_\lambda \beta v^2$$

$$+ \frac{1}{C_1 C_2}\frac{6[4\rho_\Lambda^4 - 5\rho_\Lambda^2 + 3]\beta^2}{(1-\rho_\Lambda^2)^3 nT}\left\|E_{\mathrm{II}}^{(0)}\right\|_{F,\lambda}^2$$

$$+ \frac{1}{C_1 C_2}18\alpha^2\beta^2 v^2\left(A(\rho_\Lambda) + \frac{3}{2}c_\lambda\alpha^2\beta^2 B(\rho_\Lambda)\right),$$

*where $C_2 = 1 - \frac{2\alpha^2\beta^2 B(\rho_\Lambda)}{C_1} > 0$.*

*Remark* 6.6 (**Interpretation, scope, and tightness of the comparison**). Both strategies achieve the same asymptotic order $\mathcal{O}(1/T)$ in Theorem 6.5, but differ in their constants. Within the standard single-loop GT framework considered in this paper, this comparison is decisive in the following sense: both strategies are designed to solve the same weighted loss in (1), and their aggregate recursions share the same weighted gradient structure. Therefore, within this framework, the difference between the two bounds comes from the consensus-error recursion.

Specifically, as suggested by the consensus-error analysis in Section 6.2, Strategy I uses a doubly-stochastic matrix $W^{\mathrm{ds}}$, which is non-self-adjoint in the $\lambda$-weighted space. This leads to additional $\kappa_\lambda^2$- and $\lambda_{\max}^2$-type factors in (6), amplifying the consensus and stochastic-gradient terms. In contrast, Strategy II encodes the weights through a row-stochastic matrix that is self-adjoint in the weighted space, thereby avoiding these multiplicative penalties in (7). Hence, the rate gap identified in Theorem 6.5 is not merely an artifact of a loose upper bound, but a structural consequence of

the mixing matrix geometry under the same single-loop GT framework. In particular, even when $W^{\mathrm{ds}}$ has a larger spectral gap, Strategy II can still have a sharper non-asymptotic bound due to its smaller constants.

Meanwhile, this result should not be interpreted as a universal minimax statement over all decentralized algorithms, nor as a formal lower bound ruling out every doubly-stochastic variant. Formal lower bounds and algorithmic optimality guarantees in decentralized optimization typically depend on the algorithmic class, communication model, topology, and whether extra communication rounds or acceleration are allowed. Such modifications may change the comparison, but they belong to broader algorithmic classes beyond the scope of this paper and are left for future work.

**Recovery of linear-speedup under uniform weighting.** Our framework demonstrates consistency with established results in the uniform-weight setting. When the optimization problem (1) reduces to the uniform-weight scenario (i.e., $\lambda \equiv \mathbf{1}$), both Strategy I and Strategy II become equivalent, corresponding to classical decentralized SGD with doubly-stochastic mixing and Euclidean-space analysis. The asymptotic convergence rate is stated in the following corollary.

**Corollary 6.7.** *Under uniform weighting ($\lambda \equiv \mathbf{1}$), Algorithm 1 achieves the following asymptotic convergence rate:*

$$\frac{1}{T}\sum_{t=0}^{T-1}\mathbb{E}\left[\|\nabla F(\overline{\theta}^{(t)})\|^2\right] \lesssim_{n,T} \frac{1}{\sqrt{nT}}.$$

This result precisely matches the asymptotic linear speedup convergence established in prior work (Lian et al., 2017), confirming the consistency of our generalized framework with classical analysis in the uniform-weight setting.

### 6.4. Limitations of the Standard Euclidean Framework.

To ensure analytical rigor, the convergence rate for **Strategy I** presented in our main results is derived within the Hilbert space $L^2(\lambda; \mathbb{R}^d)$ framework. This approach reveals limitations of conventional Euclidean-space analysis, which does not fully capture the geometric structure of decentralized learning with heterogeneous node weights.

While the standard Euclidean framework can be applied to **Strategy I** through rescaling of smoothness and noise constants, it produces much looser convergence bounds. Specifically, in Strategy I, the rescaled loss $F_i' = \lambda_i F_i$ has effective local smoothness $\lambda_i \beta_i$, which is bounded by $\lambda_{\max}\beta$. Similarly, the gradient-noise variance is bounded by $\lambda_{\max}^2 v^2$. This gives the conservative estimates

$$\|\nabla F_i'(\theta_1) - \nabla F_i'(\theta_2)\| \leq \lambda_{\max}\beta\|\theta_1 - \theta_2\|,$$
$$\mathbb{E}\left[\|\nabla f_i'(\theta;\xi) - \nabla F_i'(\theta)\|^2\right] \leq \lambda_{\max}^2 v^2.$$

Then, the convergence rate becomes:

If $\alpha < \frac{1}{\lambda_{\max}\beta}\sqrt{\frac{1}{62B(\rho_J)}}$, then

$$
\begin{aligned}
&\frac{1}{T}\sum_{t=0}^{T-1}\mathbb{E}[\|\nabla F(\overline{\theta}^{(t)})\|^2] \\
&\leq \left(\frac{1}{C_2'}\right)\frac{2[F(\overline{\theta}^{(0)})-F(\theta^\star)]}{\alpha T}+\left(\frac{\lambda_{\max}^3}{nC_2'}\right)\alpha\beta\upsilon^2 \\
&\quad +\left(\frac{\lambda_{\max}^2}{C_1'C_2'}\right)\frac{6[4\rho_J^4-5\rho_J^2+3]\beta^2}{(1-\rho_J^2)^3nT}\left\|E_{\mathrm{I}}^{(0)}\right\|_{F,\lambda}^2 \\
&\quad +\left(\frac{\lambda_{\max}^4}{C_1'C_2'}\right)18\alpha^2\beta^2\upsilon^2 A(\rho_J) \\
&\quad +\left(\frac{\lambda_{\max}^6}{C_1'C_2'}\right)24c_\lambda\alpha^4\beta^4\upsilon^2 B(\rho_J).
\end{aligned}
$$

When compared with our refined result in Theorem 6.5, the differences are highlighted in brackets. The standard analysis introduces multiple $\lambda_{\max}$ factors (up to the sixth power) that substantially inflate the error bounds, showing that the weighted Hilbert-space framework gives a tighter characterization of the dependence on heterogeneous weights.

*Remark* 6.8. Our goal is not to design a fully $\beta_i$-aware variant, but to compare where a prescribed weight vector should be placed under a standard shared-smoothness GT framework. In addition, we do not report a standard Euclidean-space convergence rate for Strategy II: in this geometry, the row-stochastic matrix $W$ is non-self-adjoint and incurs additional penalty factors (Liang et al., 2025a). The analysis in $L^2(\lambda;\mathbb{R}^d)$-Hilbert space therefore provides a sharper characterization for Strategy II.

# 7. Head-to-Head Convergence: Spectral Conditions, Topology Design, and Step Sizes

Theorem 6.5 provides convergence rates for both strategies. A direct comparison at fixed $\alpha$ and $T$ is obscured by higher-order terms and constants. We therefore begin by deriving a sufficient spectral condition under which Strategy II achieves strictly faster convergence, then translate it into a topological condition, highlight its step-size advantage, and finally provide corresponding topology-design guidelines.

## 7.1. Spectral and Topological Conditions for Faster Convergence and Step Size Advantage

Strategy II can outperform Strategy I even when $\rho_\Lambda > \rho_J$, due to the self-adjointness of the $\lambda$-induced operator. This phenomenon is formalized below.

**Theorem 7.1.** *There exists a constant $\eta > 0$ such that if the spectral gaps satisfy*

$$
(1-\rho_\Lambda) \geq \min\left\{(1+\eta)\lambda_{\max}^{-1/2}, 1\right\}(1-\rho_J),
$$

*then Strategy II converges strictly faster.*

We denote $R = \min\left\{(1+\eta)\lambda_{\max}^{-1/2}, 1\right\} \in (0,1]$. Because spectral gaps of $W$ and $W^{\mathrm{ds}}$ are often difficult to interpret directly, we next translate the condition in Theorem 7.1 into a more transparent relationship between the network topology and the weight vector $\lambda$. Closed-form expressions for the spectral gaps are generally unavailable in high dimensions (Horn & Johnson, 2012); we therefore compare $\rho_\Lambda$ and $\rho_J$ via Rayleigh quotients (Golub & Van Loan, 2013).

**Theorem 7.2.** *A necessary and sufficient condition for Theorem 7.1 is that the weight vector $\lambda$ satisfies*

$$
R\cdot\inf_{\substack{z'\neq 0\\ \langle z',\mathbf{1}\rangle=0}}\frac{\sum_{i,j}\min\left\{\frac{1}{d_i},\frac{1}{d_j}\right\}(z_i'-z_j')^2}{\|z'\|_2^2} \tag{8}
$$

$$
\leq \inf_{\substack{z\neq 0\\ \langle z,D_\lambda^{1/2}\mathbf{1}\rangle=0}}\frac{\sum_{i,j}\min\left\{\frac{1}{d_i}\sqrt{\frac{\lambda_i}{\lambda_j}},\frac{1}{d_j}\sqrt{\frac{\lambda_j}{\lambda_i}}\right\}(z_i-z_j)^2}{\|z\|_2^2}.
$$

As condition (8) is difficult to evaluate directly, we also provide a simple sufficient condition.

**Corollary 7.3.** *If the weight vector $\lambda$ satisfies*

$$
R\min\left\{\frac{d_i}{d_j},1\right\} \leq \sqrt{\frac{\lambda_i}{\lambda_j}} \leq R^{-1}\max\left\{\frac{d_i}{d_j},1\right\} \quad \forall i,j, \tag{9}
$$

*then Theorem 7.1 holds. When the total degree is fixed, $1-\rho_\Lambda$ is maximized when $\lambda \propto d$, i.e., $\frac{\lambda_i}{\lambda_j} = \frac{d_i}{d_j}$ for all $i,j$ $(\exists\, c>0 : \lambda = cd)$.*

*Remark* 7.4. Under condition (9), the Laplacians satisfy $\mathcal{L}(\lambda) \succeq R\mathcal{L}(\mathbf{1})$ in the Loewner order (Horn & Johnson, 2012), implying $\sigma_i(\mathcal{L}(\lambda)) \geq R\,\sigma_i(\mathcal{L}(\mathbf{1}))$ for all $i$.

**Step-size advantage of Strategy II.** The step-size bounds in Theorem 6.5 further imply that

$$
1-\rho_\Lambda \geq \lambda_{\max}^{-1/2}(1-\rho_J) \quad\Longrightarrow\quad \alpha_{\max}^{\mathrm{II}} \geq \alpha_{\max}^{\mathrm{I}}.
$$

Thus, even when $1-\rho_\Lambda$ is smaller, Strategy II still admits larger step sizes. In practice, a broader admissible step size range allows more aggressive learning rates with faster per-iteration decrease, and further enhances robustness to stochastic noise and generalization performance (Ghadimi & Lan, 2013; Wu et al., 2022).

## 7.2. Design Principles and Degree Allocation

From Corollary 7.3, on regular graphs the uniform weight $\lambda \equiv \mathbf{1}$ maximizes the spectral gap; with heterogeneous weights, regularity is suboptimal.

**Design intuition.** Corollary 7.3 suggests matching degrees to $\lambda$: nodes with larger $\lambda_i$ should have higher connectivity. Therefore, we propose Algorithm 2[1], which constructs a connected simple graph whose degree sequence approximates the proportionality to the weights $\lambda$.

---

**Algorithm 2** Build Graph From Weights

---

**Require:** Node weight vector $\lambda$, target average degree $\bar{d}$
**Ensure:** A connected simple graph $\mathcal{G}_\lambda$ whose degrees approximately match the target degrees from the weights.
 1: $d \leftarrow \text{SCALETODEGREES}(\lambda_1, \ldots, \lambda_n, \bar{d})$
 2: **for** trial $= 1, 2, \ldots, K$ **do**
 3:    success, $\mathcal{G}_\lambda \leftarrow \text{HAVELHAKIMICONSTRUCT}(d)$
 4:    **if not** success **then**
 5:       **continue** {go to next trial}
 6:    **end if**
 7:    $\mathcal{G}_\lambda \leftarrow \text{MAKECONNECTED}(\mathcal{G}_\lambda)$
 8:    **if** $\mathcal{G}_\lambda$ is connected and $\deg_{\mathcal{G}_\lambda}(i) = d_i$ for all $i$ **then**
 9:       **return** $\mathcal{G}_\lambda$
10:    **end if**
11: **end for**
12: $\mathcal{G}_\lambda \leftarrow \text{FALLBACKCONNECTEDGRAPH}(d)$
13: **return** $\mathcal{G}_\lambda$

---

Algorithm 2 first rescales $\lambda$ into an integer degree sequence $d$ with even total sum and bounded degrees $d_i \in [1, n-1]$ using the subroutine SCALETODEGREES. It then repeatedly attempts to realize $d$ as a simple graph via a Havel–Hakimi procedure (HAVELHAKIMICONSTRUCT) (Hakimi, 1962), and applies a degree-preserving edge-swap routine (MAKECONNECTED) to merge disconnected components until a connected realization is obtained. If no connected realization with $\deg_{\mathcal{G}_\lambda}(i) = d_i$ for all $i$ is found in $K$ trials, the algorithm falls back to a simple scheme that first builds a ring and then greedily adds edges to match $d$ as closely as possible (FALLBACKCONNECTEDGRAPH). Detailed pseudocode for the subroutines is provided in Appendix E.

# 8. Experiments

We empirically validate the theoretical results on a synthetic least-squares task and CIFAR-10 (Krizhevsky et al., 2009). Experiments are conducted on 16-node networks with two weights $\lambda_A$ and $\lambda_B$, and are further scaled to 32 nodes with $\lambda_C$ and 64 nodes with $\lambda_D$ when applicable, with increasing weight imbalance. The least-squares experiments cover all three scales, while the CIFAR-10 experiments are run up to 32 nodes. For each weight vector, we test standard topologies and the tailored graphs generated by Algorithm 2. More details are provided in Appendix F.

---

[1]For $n$ nodes, Algorithm 2 runs in $O(n^2 \log n)$ time in typical cases and in $O(n^3)$ time in the worst case, and always returns a connected simple graph due to the fallback construction.

## 8.1. Least-Squares Quadratic Experiment

Following the setup of Koloskova et al. (2020), we conduct decentralized least-squares experiments with minor modifications, as detailed in Section F.3. Each local loss is augmented with an $\ell_2$ regularization term, and stochastic gradients are simulated by injecting zero-mean Gaussian noise into the true gradients. Figures 1 and 2 report the norm of the weighted average gradient $\left\| \sum_{i=1}^n \frac{\lambda_i}{n} \nabla F_i(\theta_i^{(t)}) \right\|$.

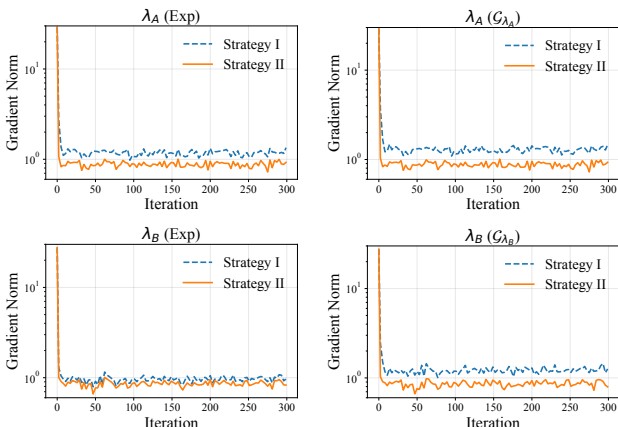

*Figure 1.* Weighted gradient norms for least-squares (16 nodes).

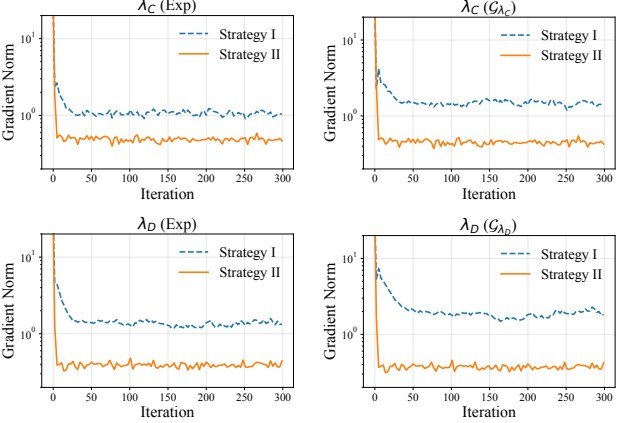

*Figure 2.* Weighted gradient norms for least-squares (32/64 nodes).

Figures 1 and 2 show that Strategy II consistently converges faster than Strategy I and reaches a tighter neighborhood of the same optimum $\theta^\star$, resulting in a smaller steady-state gradient norm. This is consistent with the penalty terms in Theorem 6.5. Additional plots in Appendix F.4 track the distance to $\theta^\star$ and further confirm this observation.

We scale the least-squares experiment from 16 nodes to 32 and 64 nodes. Figure 2 shows that the convergence gap becomes wider with stronger weight heterogeneity. Even when $W$ has a smaller spectral gap than $W^{ds}$, Strategy II can still converge faster, confirming that the gap is not determined by spectral gaps alone, but also by the self-adjointness of the mixing matrix.

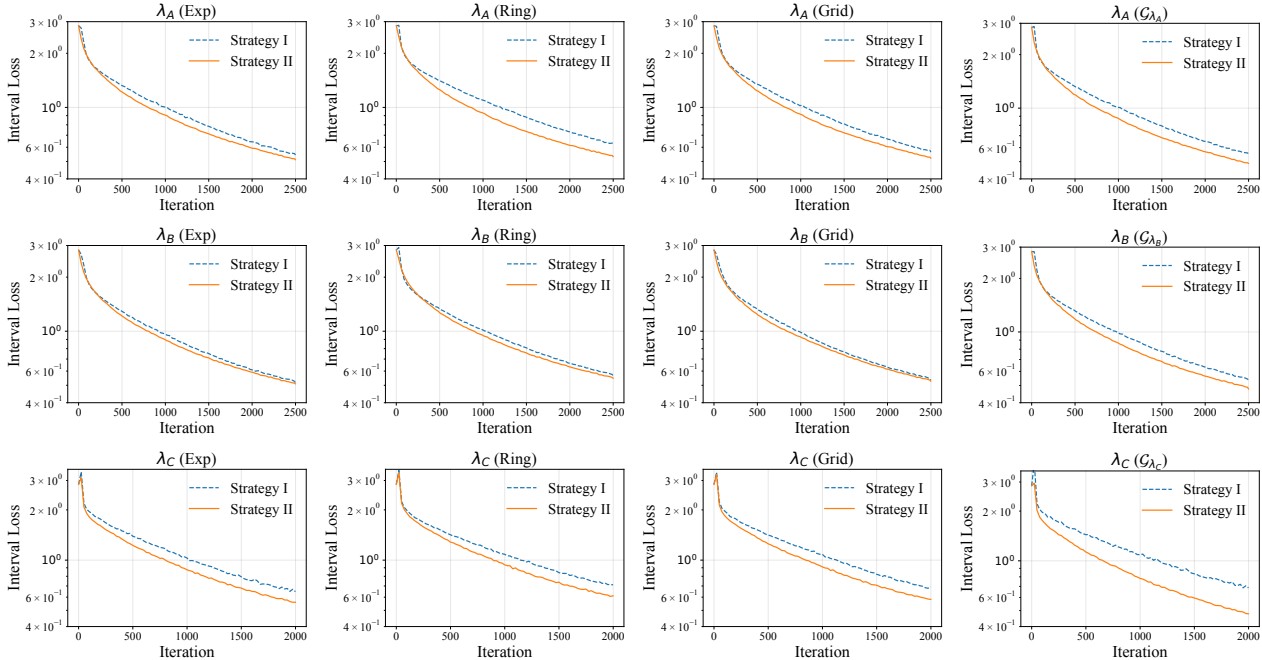

*Figure 3.* Interval losses for CIFAR-10 experiments. The first two rows report the 16-node results under $\lambda_A$ and $\lambda_B$, and the third row reports the 32-node result under $\lambda_C$.

## 8.2. Training ResNet-18 on CIFAR-10

To further validate the results, we evaluate the strategies on a deep learning task using the ResNet-18 model (He et al., 2016) on the CIFAR-10 dataset. The performance of Strategy I and Strategy II in Algorithm 1 is measured by the weighted average instantaneous loss $\sum_{i=1}^{n} \frac{\lambda_i}{n} f_i(\theta_i^{(t)}; \xi_i^{(t)})$ across nodes. The loss curves are averaged over 6 random seeds and smoothed with a 30-iteration moving average.

Figure 3 shows that Strategy II consistently achieves lower interval loss than Strategy I across different weights and topologies. The advantage expands in the $32-$node setting under $\lambda_C$, implying that the performance gap is not limited to the 16-node case. Moreover, Strategy II can remain faster even when $W^{\mathrm{ds}}$ has a larger spectral gap, supporting our conclusion that convergence is affected not only by spectral gaps, but also by the self-adjointness structure and the resulting multiplicative constants. Complete test accuracies and additional topology results are provided in Appendix F.5.

## 9. Conclusions and Limitations

We revisit decentralized learning with heterogeneous node weights and show that the standard Euclidean framework, due to a coarse rescaling argument, leads to loose convergence rates. In contrast, a proposed weighted Hilbert-space analysis yields tighter convergence rates and reveals that the row-stochastic matrix is self-adjoint while the doubly stochastic one is not, introducing penalty terms that slow

convergence and tighten step sizes. These insights show that performance gaps are not dictated by spectral gaps alone and lead to degree–weight guidelines for topology design. Extending the analysis to directed graphs and developing lower-bound or optimality characterizations are important directions for future work, as both require tools beyond the self-adjoint weighted-space framework developed here.

## Acknowledgments

The work of Liu, Lu, and Zhao is supported in part by the National Natural Science Foundation of China under Grant 62273305 and 62293515, in part by the Zhejiang Provincial Natural Science Foundation of China under Grant LR25F030002, in part by Zhejiang University Special Project of the State Key Laboratory of Industrial Control Technology under Grant ICT2025C05, and in part by the Fundamental Research Funds for Zhejiang Provincial Universities under Grant 226-2025-00221. Kun Yuan is supported by the National Key Research and Development Program of China (No. 2024YFA1012902).

## Impact Statement

This paper presents work whose goal is to advance the field of Machine Learning. There are many potential societal consequences of our work, none which we feel must be specifically highlighted here.

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

# Appendix

## A. Motivation for Heterogeneous Node Weights

In this paper, the weights $\lambda = [\lambda_1, \ldots, \lambda_n]^\top$ are prescribed by the target learning problem and remain fixed throughout the optimization process. They are not tunable variables. Instead, they specify how much each local loss contributes to the weighted global loss

$$F(\theta) = \frac{1}{n} \sum_{i=1}^{n} \lambda_i F_i(\theta).$$

Heterogeneous weights arise naturally in decentralized learning. A common example is data-size weighting (McMahan et al., 2017; Yuan et al., 2018a): if node $i$ owns $m_i$ samples and $F_i$ denotes its empirical risk averaged over these samples, then the empirical risk over all the data is

$$\frac{1}{\sum_{j=1}^{n} m_j} \sum_{i=1}^{n} m_i F_i(\theta),$$

which corresponds to choosing $\lambda_i \propto m_i$. Thus, when local data volumes are unequal, the desired global loss is generally weighted rather than uniformly averaged. Similar weighted losses also appear when nodes have different data distributions, priorities, or influence scores (Zhu et al., 2025).

Since uniform weighting can be viewed as a special case of heterogeneous weighting, some distributed optimization studies explicitly introduce node-specific weights in the problem formulation (Yuan et al., 2018b;c;a).

Heterogeneous weights can also be induced algorithmically. In decentralized methods with non-uniform mixing, the stationary distribution of the mixing matrix determines the effective contribution of each node to the global loss (Sayed, 2014; Ying & Sayed, 2016; Cyffers et al., 2024). Therefore, even when an algorithm is not explicitly weighted, local step-sizes or the stationary distribution of the mixing matrix may implicitly create a non-uniform weighting of local losses.

The role of this paper is not to design the weights, but to study how a prescribed weight vector should be incorporated into a decentralized gradient-tracking method. We compare two standard realizations of the same weighted loss: Strategy I absorbs $\lambda_i$ into the local losses, while Strategy II encodes $\lambda$ through the stationary distribution of the mixing matrix. Both strategies target the same loss above; they differ only in how the weights affect the consensus dynamics and, consequently, the convergence rates.

## B. More Details on the $L^2(\lambda; \mathbb{R}^d)$-Hilbert Space

### B.1. Definition of $L^2(\lambda)$ space

Let $S$ denote the index set of nodes and let

$$\lambda = (\lambda_i)_{i \in S}, \qquad \lambda_i > 0, \qquad \sum_{i \in S} \lambda_i = n > 1,$$

be the vector of heterogeneous node weights used in the decentralized optimization problem (1). Here $\lambda$ should be regarded as a *positive weight vector* associated with the network nodes, rather than a probability distribution.

The weighted Hilbert space $L^2(\lambda)$ is defined as

$$L^2(\lambda) := \Big\{ \phi : S \to \mathbb{R} \Big| \sum_{i \in S} |\phi(i)|^2 \lambda_i < \infty \Big\},$$

with inner product and induced norm

$$\langle \phi_1, \phi_2 \rangle_\lambda := \sum_{i \in S} \lambda_i \phi_1(i) \phi_2(i), \qquad \|\phi\|_\lambda := \big( \langle \phi, \phi \rangle_\lambda \big)^{1/2}.$$

If $\lambda$ is replaced by $c\lambda$ for any constant $c > 0$, then

$$\langle \phi_1, \phi_2 \rangle_{c\lambda} = c \langle \phi_1, \phi_2 \rangle_{\lambda}, \qquad \|\phi\|_{c\lambda} = \sqrt{c} \|\phi\|_{\lambda}.$$

Hence the element set of $L^2(\lambda)$ is unchanged, and the geometry of the Hilbert space is preserved up to a scaling factor.

In this paper, we focus on the matrix $W$ introduced in (3), which is constructed to satisfy the detailed balance condition with respect to $\lambda$:

$$\lambda_i W_{ij} = \lambda_j W_{ji}, \quad \forall i, j \in S.$$

As a result, $W$ is self-adjoint with respect to $\langle \cdot, \cdot \rangle_{\lambda}$, and its spectral properties are invariant to rescaling of $\lambda$. Equivalently, under the $\lambda$-similarity transform, $\widetilde{W} := D_{\lambda}^{1/2} W D_{\lambda}^{-1/2}$ is symmetric, which can be readily verified from its expression:

$$\widetilde{W}_{i,j} = \begin{cases} (1-\varepsilon) \min\left( \frac{1}{d_i} \sqrt{\frac{\lambda_i}{\lambda_j}}, \frac{1}{d_j} \sqrt{\frac{\lambda_j}{\lambda_i}} \right) & \text{if } i \neq j \text{ and } j \in \mathcal{N}_i \\ 1 - \sum_{k \in \mathcal{N}_i} W_{i,k} & \text{if } i = j \\ 0 & \text{otherwise.} \end{cases}$$

**Remark.** It is worth emphasizing that $\lambda$ in our setting is not required to be the stationary distribution of a Markov chain. Instead, it is a user-defined weight vector that induces the reversibility condition for $W$. While stationary distributions are a special case where $\lambda$ is normalized to sum to one, our construction applies more generally and does not rely on probabilistic normalization.

## B.2. $L^2(\lambda; \mathbb{R}^d)$-**Hilbert space.**

In addition to the scalar-valued space $L^2(\lambda)$, it is often convenient to consider its vector-valued extension

$$L^2(\lambda; \mathbb{R}^d) := \left\{ \phi : S \to \mathbb{R}^d \,\Big|\, \sum_{i \in S} \lambda_i \|\phi(i)\|^2 < \infty \right\}.$$

This space can be identified with the tensor product

$$L^2(\lambda; \mathbb{R}^d) \cong L^2(\lambda) \,\widehat{\otimes}\, \mathbb{R}^d.$$

This identification simply reflects that an element of $L^2(\lambda; \mathbb{R}^d)$ consists of $n$ vectors in $\mathbb{R}^d$, with geometry determined independently by the weighted node index $(L^2(\lambda))$ and the feature dimension $(\mathbb{R}^d)$.

The inner product between $\phi_1, \phi_2 \in L^2(\lambda; \mathbb{R}^d)$ is given by

$$\langle \phi_1, \phi_2 \rangle_{\lambda, d} = \sum_{i \in S} \lambda_i \langle \phi_1(i), \phi_2(i) \rangle,$$

where $\langle \cdot, \cdot \rangle$ denotes the standard Euclidean inner product in $\mathbb{R}^d$. The induced norm reads

$$\|\phi\|_{\lambda, d}^2 = \sum_{i \in S} \lambda_i \|\phi(i)\|^2.$$

This vector-valued extension inherits all Hilbert space properties of $L^2(\lambda)$ and is particularly useful when analyzing multi-dimensional signals or operator dynamics with matrix-valued actions. Equivalently, if $\phi$ is arranged as a $|S| \times d$ matrix with $\phi(i)$ as its $i$-th row, then $\|\phi\|_{\lambda, d}^2$ coincides with the $\lambda$-weighted Frobenius norm of this matrix, i.e., $\|\phi\|_{F, \lambda}^2$.

## B.3. Why introduce the $L^2(\lambda; \mathbb{R}^d)$ space?

The motivation for introducing this space is twofold. First, under heterogeneous weights, the tightest result from the descent lemma is naturally expressed in the $L^2(\lambda; \mathbb{R}^d)$ norm; using the standard Euclidean space instead would introduce looser bounds. Second, the matrix $W$ is generally not symmetric under the standard Euclidean inner product, so its eigenvectors

are not orthogonal. Analyzing in the Euclidean space thus inevitably requires coarse bounding, leading to convergence results that are not sufficiently tight.

By introducing the weighted Hilbert space $L^2(\lambda)$, endowed with the inner product

$$\langle \phi_1, \phi_2 \rangle_\lambda = \sum_{i \in S} \lambda_i \phi_1(i) \phi_2(i),$$

we recover orthogonality of eigenvectors when $W$ is reversible with respect to $\lambda$, i.e., when the detailed balance condition $\lambda_i W_{ij} = \lambda_j W_{ji}$ holds. In this case $W$ is self-adjoint with respect to $\langle \cdot, \cdot \rangle_\lambda$, and thus admits an orthogonal spectral decomposition in $L^2(\lambda)$. This structure enables the use of Hilbert space techniques, such as Pythagorean identities and spectral gap estimates, to quantify the rate at which the dynamics defined by $W$ converge to equilibrium.

Even when $\sum_i \lambda_i \neq 1$, the space $L^2(\lambda)$ remains well defined, since normalization of $\lambda$ only rescales the inner product without altering the spectral properties of $W$ or the convergence rate analysis.

### B.4. A Simple Consensus Problem

This subsection uses a simple consensus problem to illustrate why the two weighting strategies lead to different contraction factors in the $\lambda$-weighted space. Consider the pure consensus dynamics without gradient updates:

$$\Theta^{(t+1)} = P\Theta^{(t)},$$

where $\Theta^{(t)} \in \mathbb{R}^{n \times d}$ collects the local variables of all nodes. Recall that

$$\Lambda := \frac{\mathbf{1}\lambda^\top}{n}, \qquad J := \frac{\mathbf{1}\mathbf{1}^\top}{n}.$$

Here, $\Lambda$ is the projection induced by the stationary distribution $\lambda/n$, while $J$ is the standard averaging projection. We compare the consensus contraction of the two strategies in the $\lambda$-weighted Hilbert space.

**Strategy II: row-stochastic mixing.** For Strategy II, the consensus dynamics are

$$\Theta^{(t+1)} = W\Theta^{(t)},$$

where $W$ is row-stochastic and satisfies

$$\frac{\lambda^\top}{n}W = \frac{\lambda^\top}{n}.$$

Since $W\mathbf{1} = \mathbf{1}$ and $\frac{\lambda^\top}{n}W = \frac{\lambda^\top}{n}$, we have

$$W\Lambda = \Lambda, \qquad \Lambda W = \Lambda.$$

Therefore,

$$(I - \Lambda)W^t = (W - \Lambda)^t.$$

The consensus error satisfies

$$\left\| (I - \Lambda)\Theta^{(t)} \right\|_{F,\lambda}^2 = \left\| (I - \Lambda)W^t\Theta^{(0)} \right\|_{F,\lambda}^2 = \left\| (W - \Lambda)^t\Theta^{(0)} \right\|_{F,\lambda}^2 \leq \rho_\Lambda^{2t} \left\| \Theta^{(0)} \right\|_{F,\lambda}^2,$$

where

$$\rho_\Lambda := \| W - \Lambda \|_\lambda$$

denotes the contraction factor in the $\lambda$-weighted space. Since $W$ is self-adjoint in $L^2(\lambda; \mathbb{R}^d)$ under our construction, this contraction is directly characterized in the weighted norm.

**Strategy I: doubly stochastic mixing.** For Strategy I, the consensus dynamics are

$$\Theta^{(t+1)} = W^{\mathrm{ds}}\Theta^{(t)},$$

where $W^{\mathrm{ds}}$ is doubly stochastic. The corresponding consensus projection is $J$, and

$$W^{\mathrm{ds}}J = J, \qquad JW^{\mathrm{ds}} = J.$$

Thus,

$$(I - J)(W^{\mathrm{ds}})^t = (W^{\mathrm{ds}} - J)^t.$$

Evaluating the consensus error in the $\lambda$-weighted norm gives

$$\left\|(I - J)\Theta^{(t)}\right\|_{F,\lambda}^2 = \left\|(I - J)(W^{\mathrm{ds}})^t\Theta^{(0)}\right\|_{F,\lambda}^2 = \left\|(W^{\mathrm{ds}} - J)^t\Theta^{(0)}\right\|_{F,\lambda}^2 \leq \kappa_\lambda^2 \rho_J^{2t} \left\|\Theta^{(0)}\right\|_{F,\lambda}^2,$$

where $\rho_J$ denotes the standard consensus contraction factor associated with $W^{\mathrm{ds}} - J$, and $\kappa_\lambda > 1$ quantifies the metric distortion induced by heterogeneous weights.

This additional factor appears because $W^{\mathrm{ds}}$ is generally not self-adjoint in the $\lambda$-weighted space. As a result, its contraction in the weighted norm is naturally bounded with the factor $\kappa_\lambda$. In contrast, Strategy II directly aligns the stationary distribution of the mixing matrix with the weighted geometry, avoiding this multiplicative penalty.

This simple consensus example shows that the gap between the two strategies already appears at the consensus level. The additional $\lambda_{\max}^2$-type penalty in Theorem 6.5 further arises from the gradient-tracking dynamics and the interaction with the gradient-related scaling, rather than from the pure consensus step alone.

## C. Proof for the Convergence Rate

In this section, we present the proof of convergence for the proposed algorithms.

### C.1. Technical lemmas

To begin with, we introduce some technical lemmas.

The following lemma provides the eigenvalue of the row-stochastic matrix.

**Lemma C.1.** *Suppose $P \in \mathbb{R}^{n \times n}$ is a row-stochastic matrix and $\pi$ is its stationary distribution, then the eigenvalues of the matrix $P - \mathbf{1}\pi$ can be given by:*
$$\sigma(P - \mathbf{1}\pi) = \{0, \sigma_2(P), \ldots, \sigma_n(P)\}.$$

*Proof.* As $P$ is a row-stochastic matrix with stationary distribution $\pi$, then it holds that $P\mathbf{1} = \mathbf{1}$, $\pi P = \pi$, and $\pi\mathbf{1} = 1$. Denote $\Pi = \mathbf{1}\pi$, then $(P - \Pi)\mathbf{1} = P\mathbf{1} - \mathbf{1}(\pi\mathbf{1}) = \mathbf{1} - \mathbf{1} = 0$. Thus 0 is an eigenvalue of $P - \Pi$ with eigenvector $\mathbf{1}$.

Moreover, if $\sigma \neq 1$ is an eigenvalue of $P$ with respect to the vector $v$, it holds from $\pi P = \pi$ and $Pv = \sigma v$ that $\pi v = \pi(Pv) = \sigma\pi v$. Thus $(\sigma - 1)\pi v = 0$ and $\pi v = 0$ holds. Consequently, it holds that $\Pi v = \mathbf{1}\pi v = 0$, and $(P - \Pi)v = Pv = \sigma v$.

Therefore, combining the two aspects yields $\sigma(P - \mathbf{1}\pi) = \{0, \sigma_2(P), \ldots, \sigma_n(P)\}$. $\qquad\square$

**Lemma C.2.** *Suppose $\omega_1, \ldots, \omega_n > 0$, and let $x_1, \ldots, x_n \in \mathbb{R}^d$ be independent random vectors with zero expectation. Then, we have*

$$\mathbb{E}\left[\left\|\sum_{i=1}^n \omega_i x_i\right\|^2\right] = \sum_{i=1}^n \omega_i^2 \mathbb{E}\left[\|x_i\|^2\right] = \sum_{i=1}^n \omega_i^2 \operatorname{var}(x_i).$$

*Proof.* By independence and the zero-mean property of $x_1, \ldots, x_n$, we can derive that:

$$\mathbb{E}\left[\left\|\sum_{i=1}^{n}\omega_i x_i\right\|^2\right] = \mathbb{E}\left[\left\langle\sum_{i=1}^{n}\omega_i x_i, \sum_{j=1}^{n}\omega_j x_j\right\rangle\right] = \mathbb{E}\left[\sum_{i=1}^{n}\sum_{j=1}^{n}\omega_i\omega_j\langle x_i, x_j\rangle\right]$$

$$= \mathbb{E}\left[\sum_{i=1}^{n}\omega_i^2\langle x_i, x_i\rangle\right] = \sum_{i=1}^{n}\omega_i^2\mathbb{E}\left[\|x_i\|^2\right] = \sum_{i=1}^{n}\omega_i^2\,\text{var}(x_i).$$

Thus we finish the proof of this lemma. $\square$

The following lemma can be obtained from Young's inequality:

**Lemma C.3.** *Given two matrices $A, B \in \mathbb{R}^{m \times d}$, for any $a > 0$, the following inequality holds:*

$$\|A + B\|_{F,\lambda}^2 \leq (1+a)\|A\|_{F,\lambda}^2 + \left(1 + \frac{1}{a}\right)\|B\|_{F,\lambda}^2.$$

**Lemma C.4.** *Let $W$ be the matrix constructed in* (3)*. Under the $\lambda$-similarity transform $\widetilde{W} = D_\lambda^{1/2}WD_\lambda^{-1/2}$, the transformed matrix $\widetilde{W}$ is symmetric.*

*Proof.* From the definition in (3), the entries of $W$ are given by

$$W_{i,j} = \begin{cases} \dfrac{1-\varepsilon}{d_i}\min\left(1, \dfrac{\lambda_j d_i}{\lambda_i d_j}\right), & \text{if } i \neq j \text{ and } j \in \mathcal{N}_i, \\ 1 - \sum_{k\in\mathcal{N}_i} W_{i,k}, & \text{if } i = j, \\ 0, & \text{otherwise.} \end{cases}$$

Applying the similarity transformation $\widetilde{W} = D_\lambda^{1/2}WD_\lambda^{-1/2}$ yields

$$\widetilde{W}_{i,j} = \begin{cases} (1-\varepsilon)\min\left(\dfrac{1}{d_i}\sqrt{\dfrac{\lambda_i}{\lambda_j}}, \dfrac{1}{d_j}\sqrt{\dfrac{\lambda_j}{\lambda_i}}\right), & \text{if } i \neq j \text{ and } j \in \mathcal{N}_i, \\ 1 - \sum_{k\in\mathcal{N}_i} W_{i,k}, & \text{if } i = j, \\ 0, & \text{otherwise.} \end{cases}$$

By construction, $[\widetilde{W}]_{i,j} = [\widetilde{W}]_{j,i}$ for all $i, j$. Therefore, $\widetilde{W}$ is symmetric under the $\lambda$-similarity transform. $\square$

**Lemma C.5.** *Let $W_J := W^{\text{ds}} - J$, $J = \frac{\mathbf{1}\mathbf{1}^\top}{n}$, and $D_\lambda = \text{diag}(\lambda_1, \ldots, \lambda_n)$. Then the $\lambda$-weighted spectral norm of $(W_J)^t(I - J)D_\lambda$ satisfies that:*

$$\|W_J^t(I - J)D_\lambda\|_\lambda \leq \lambda_{\max}\rho_J^t,$$

*where $\lambda_{\max} = \max_i \lambda_i$ and $\rho_J = \|W^{\text{ds}} - J\|_2 < 1$ denotes the spectral radius of $W^{\text{ds}} - J$.*

*Proof.* Note that $W^{\text{ds}}$ is a doubly stochastic matrix, we have

$$W^{\text{ds}}J = JW^{\text{ds}} = J,$$

Hence,

$$(W_J)^t(I - J) = (W^{\text{ds}} - J)^t(I - J) = \left[(W^{\text{ds}})^t - J\right](I - J) = (W^{\text{ds}})^t - J.$$

Thus we consider the $\lambda$-weighted spectral norm of $(I - J)D_\lambda$:

$$\begin{aligned} \|(W_J)^t(I - J)D_\lambda\|_\lambda &= \left\|D_\lambda^{1/2}\left[(W^{\text{ds}})^t - J\right]D_\lambda D_\lambda^{-1/2}\right\|_2 \\ &= \left\|D_\lambda^{1/2}\left[(W^{\text{ds}})^t - J\right]D_\lambda^{1/2}\right\|_2 \\ &\leq \|D_\lambda^{1/2}\|_2 \cdot \|(W^{\text{ds}})^t - J\|_2 \cdot \|D_\lambda^{1/2}\|_2 \\ &= \lambda_{\max}\rho_J^t, \end{aligned}$$

where $\lambda_{\max} = \max_i \lambda_i$ and $\rho_J = \|W^{\text{ds}} - J\|_2 < 1$ denotes the spectral radius of $W^{\text{ds}} - J$. $\square$

To facilitate the subsequent proofs, we now introduce the following lemmas concerning series summations.

**Lemma C.6.** *Consider the sequence $ta^t$ for $t = 0, 1, \ldots, n$, where $0 < a < 1$. The closed-form expression for its sum is given by*

$$\sum_{t=0}^{n} ta^{2t} = \frac{a^2\left(1 - (n+1)a^{2n} + na^{2(n+1)}\right)}{(1 - a^2)^2},$$

*and the infinite sum is*

$$\sum_{t=0}^{\infty} ta^{2t} = \frac{a^2}{(1 - a^2)^2}. \tag{10}$$

*Proof.* Let $b = a^2$, the series can be rewritten as

$$\sum_{t=0}^{n} ta^{2t} = \sum_{t=0}^{n} tb^t.$$

We first consider the standard geometric series

$$G(b) = \sum_{t=0}^{n} b^t = \frac{1 - b^{(n+1)}}{1 - b}, \quad 0 < b < 1.$$

Differentiating both sides of the equation with respect to $b$ yields

$$G'(b) = \sum_{t=0}^{n} tb^{t-1} = \frac{1 - (n+1)b^n + nb^{n+1}}{(1 - b)^2}.$$

Multiplying both sides by $b$ gives the desired sum:

$$\sum_{t=0}^{n} ta^{2t} = \sum_{t=0}^{n} tb^t = b\sum_{t=1}^{n} tb^{t-1} = \frac{a^2\left(1 - (n+1)a^{2n} + na^{2(n+1)}\right)}{(1 - a^2)^2}.$$

Taking the limit as $n \to \infty$ yields the infinite sum:

$$\sum_{t=0}^{\infty} ta^{2t} = \frac{a^2}{(1 - a^2)^2}.$$

$\square$

**Lemma C.7.** *Consider the sequence $t^2a^{2t}$ for $t = 0, 1, \ldots, n$, where $0 < a < 1$. The closed-form expression for its sum is given by*

$$\sum_{t=0}^{n} t^2a^{2t} = \frac{a^2(1 + a^2) - (n+1)^2a^{2(n+1)} + (2n^2 + 2n - 1)a^{2(n+2)} - n^2a^{2(n+3)}}{(1 - a^2)^3}.$$

*Taking the limit as $n \to \infty$ yields the infinite sum:*

$$\sum_{t=0}^{\infty} t^2a^{2t} = \frac{a^2(1 + a^2)}{(1 - a^2)^3}. \tag{11}$$

*Proof.* Let $b = a^2$, the series can be rewritten as

$$\sum_{t=0}^{n} t^2a^{2t} = \sum_{t=0}^{n} t^2b^t.$$

We also begin with the standard geometric series

$$G(b) = \sum_{t=0}^{n} b^t = \frac{1 - b^{n+1}}{1 - b}, \quad 0 < b < 1.$$

Differentiating $G(b)$ with respect to $b$ yields

$$G'(b) = \sum_{t=1}^{n} tb^{t-1} = \frac{1 - (n+1)b^n + nb^{n+1}}{(1-b)^2}.$$ (12)

Differentiating once more, we obtain

$$G''(b) = \sum_{t=2}^{n} t(t-1)b^{t-2} = \frac{2 - (n+1)nb^{n-1} + 2(n^2-1)b^n - n(n-1)b^{n+1}}{(1-b)^3}.$$ (13)

Since $t^2 = t(t-1) + t$, we can decompose the sum as

$$\sum_{t=0}^{n} t^2 b^t = \sum_{t=0}^{n} t(t-1)b^t + \sum_{t=0}^{n} tb^t$$
$$= b^2 G''(b) + bG'(b).$$ (14)

Substituting (12) and (13) into (14) and replacing $b$ with $a^2$, we obtain

$$\sum_{t=0}^{n} t^2 a^{2t} = \frac{a^2(1+a^2) - (n+1)^2 a^{2(n+1)} + (2n^2 + 2n - 1)a^{2(n+2)} - n^2 a^{2(n+3)}}{(1-a^2)^3}.$$

Finally, since $a^{2n} \to 0$ as $n \to \infty$ for $0 < a < 1$, the infinite series converges to

$$\sum_{t=0}^{\infty} t^2 a^{2t} = \frac{a^2(1+a^2)}{(1-a^2)^3}.$$

$\square$

**Lemma C.8.** *Consider the sequence $t^3 a^{2t}$ for $t = 0, 1, \ldots$, where $0 < a < 1$. The closed-form expression for its infinite sum is given by*

$$\sum_{t=0}^{\infty} t^3 a^{2t} = \frac{a^2(1 + 4a^2 + a^4)}{(1-a^2)^4}.$$ (15)

*Proof.* Following Lemma C.7, which states that

$$\sum_{t=0}^{\infty} t^2 a^{2t} = \frac{a^2(1+a^2)}{(1-a^2)^3},$$

we differentiate both sides of the equation with respect to $a^2$ to obtain:

$$\frac{\mathrm{d}\sum_{t=0}^{\infty} t^2 a^{2t}}{\mathrm{d}a^2} = \sum_{t=0}^{\infty} t^3 a^{2(t-1)} = \frac{a^4 + 4a^2 + 1}{(1-a^2)^4}.$$

Therefore, we can get the final result

$$\sum_{t=0}^{\infty} t^3 a^{2t} = a^2 \sum_{t=0}^{\infty} t^3 a^{2(t-1)} = \frac{a^2(a^4 + 4a^2 + 1)}{(1-a^2)^4}.$$

$\square$

## C.2. Descent lemma

In this subsection, we present the proof of the descent lemma to obtain the convergence rate with the two strategies. To begin with, we present the following descent lemma for the two strategies.

**Lemma C.9.** *Under Assumptions 6.1 and 6.2 and a constant step size $\alpha$, it holds for Algorithm 1 with both Strategy I and II that:*

$$
\mathbb{E}\left[F(\overline{\theta}_*^{(t+1)})\right] \leq \mathbb{E}\left[F\left(\overline{\theta}_*^{(t)}\right)\right] - \frac{\alpha}{2}\mathbb{E}\left[\left\|\nabla F(\overline{\theta}_*^{(t)})\right\|^2\right] - \frac{\alpha}{2}\mathbb{E}\left[\left\|\sum_{i=1}^{n}\frac{\lambda_i}{n}\nabla F_i(\theta_i^{(t)})\right\|^2\right] + \frac{c_\lambda\alpha^2\beta\upsilon^2}{2}
$$

$$
+ \frac{\alpha^2\beta}{2}\cdot\mathbb{E}\left[\left\|\sum_{i=1}^{n}\frac{\lambda_i}{n}\nabla F_i(\theta_i^{(t)})\right\|^2\right] + \frac{\alpha T_3}{2},
\tag{16}
$$

*where $T_3 = \mathbb{E}\left[\left\|\nabla F(\overline{\theta}_*^{(t)}) - \sum_{i=1}^{n}\frac{\lambda_i}{n}\nabla F_i(\theta_i^{(t)})\right\|^2\right]$, $\overline{\theta}_*^{(t)} = \overline{\theta}^{(t)}$ in Strategy I and $\overline{\theta}_*^{(t)} = \overline{\theta}_\lambda^{(t)}$ in Strategy II.*

*Proof.* According to Assumption 6.1, we can derive

$$
\mathbb{E}\left[F(\overline{\theta}_*^{(t+1)})\right] \leq \mathbb{E}\left[F\left(\overline{\theta}_*^{(t)}\right)\right] + \underbrace{\mathbb{E}\left[\left\langle\nabla F(\overline{\theta}_*^{(t)}), \overline{\theta}_*^{(t+1)} - \overline{\theta}_*^{(t)}\right\rangle\right]}_{:=T_1} + \underbrace{\mathbb{E}\left[\frac{\beta}{2}\left\|\overline{\theta}_*^{(t+1)} - \overline{\theta}_*^{(t)}\right\|^2\right]}_{:=T_2}.
$$

Using $\overline{\theta}_*^{(t+1)} = \overline{\theta}_*^{(t)} - \frac{\alpha}{n}\sum_{i=1}^{n}\lambda_i g_i^{(t)}$, we can derive:

$$
\begin{aligned}
T_1 &= -\alpha\mathbb{E}\left[\left\langle\nabla F(\overline{\theta}_*^{(t)}), \sum_{i=1}^{n}\frac{\lambda_i}{n}g_i^{(t)}\right\rangle\right] \\
&= -\alpha\mathbb{E}\left[\left\langle\nabla F(\overline{\theta}_*^{(t)}), \sum_{i=1}^{n}\frac{\lambda_i}{n}\nabla F_i(\theta_i^{(t)})\right\rangle\right] \\
&= \frac{\alpha}{2}\mathbb{E}\left[\left\|\nabla F(\overline{\theta}_*^{(t)}) - \sum_{i=1}^{n}\frac{\lambda_i}{n}\nabla F_i(\theta_i^{(t)})\right\|^2\right] - \frac{\alpha}{2}\mathbb{E}\left[\left\|\nabla F(\overline{\theta}_*^{(t)})\right\|^2\right] - \frac{\alpha}{2}\mathbb{E}\left[\left\|\sum_{i=1}^{n}\frac{\lambda_i}{n}\nabla F_i(\theta_i^{(t)})\right\|^2\right],
\end{aligned}
\tag{17}
$$

and

$$
\begin{aligned}
T_2 &= \frac{\alpha^2\beta}{2}\mathbb{E}\left[\left\|\sum_{i=1}^{n}\frac{\lambda_i}{n}g_i^{(t)}\right\|^2\right] = \frac{\alpha^2\beta}{2}\mathbb{E}\left[\mathbb{E}_t\left[\left\|\sum_{i=1}^{n}\frac{\lambda_i}{n}\left(g_i^{(t)} - \nabla F_i(\theta_i^{(t)}) + \nabla F_i(\theta_i^{(t)})\right)\right\|^2\right]\right] \\
&= \frac{\alpha^2\beta}{2}\left(\mathbb{E}\left[\left\|\sum_{i=1}^{n}\frac{\lambda_i}{n}\left(g_i^{(t)} - \nabla F_i(\theta_i^{(t)})\right)\right\|^2\right] + \mathbb{E}\left[\left\|\sum_{i=1}^{n}\frac{\lambda_i}{n}\nabla F_i(\theta_i^{(t)})\right\|^2\right]\right) \\
&\quad + 2\mathbb{E}\left[\left\langle\sum_{i=1}^{n}\frac{\lambda_i}{n}\nabla F_i(\theta_i^{(t)}), \mathbb{E}_t\left[\sum_{i=1}^{n}\frac{\lambda_i}{n}\left(g_i^{(t)} - \nabla F_i(\theta_i^{(t)})\right)\right]\right\rangle\right] \\
&\leq \frac{\alpha^2\beta}{2}\left(\frac{1}{n^2}\sum_{i=1}^{n}\lambda_i^2\upsilon^2 + \mathbb{E}\left[\left\|\sum_{i=1}^{n}\frac{\lambda_i}{n}\nabla F_i(\theta_i^{(t)})\right\|^2\right]\right) = \frac{c_\lambda\alpha^2\beta\upsilon^2}{2} + \frac{\alpha^2\beta}{2}\cdot\mathbb{E}\left[\left\|\sum_{i=1}^{n}\frac{\lambda_i}{n}\nabla F_i(\theta_i^{(t)})\right\|^2\right],
\end{aligned}
\tag{18}
$$

where we use the Assumption 6.2. Plugging (17) and (18) into (16) and using the definition of $T_3$, we can prove that (16) holds for both strategies. □

Here, we denote $c_\lambda = \sum_{i=1}^{n}\lambda_i^2/n^2 < 1$. Then with Lemma C.9, we can present a further descent lemma for the both communication strategies.

**Lemma C.10** (Descent lemma of Strategy I). *Under Assumptions 6.1 and 6.2 and a constant step size $\alpha \leq 1/\beta$, it holds for Algorithm 1 with Strategy I that:*

$$\mathbb{E}[F(\bar{\theta}^{(t+1)})] \leq \mathbb{E}[F(\bar{\theta}^{(t)})] - \frac{\alpha}{2}\mathbb{E}[\|\nabla F(\bar{\theta}^{(t)})\|^2] + \frac{\alpha\beta^2}{2n}\mathbb{E}[\|(I-J)\Theta^{(t)}\|_{F,\lambda}^2] + \frac{\alpha^2 c_\lambda \beta \upsilon^2}{2}. \tag{19}$$

*Proof.* We consider the term $T_3$ that has been definited in Lemma C.9. It holds under Assumption 6.1 that:

$$
\begin{aligned}
\boldsymbol{T_3} &= \mathbb{E}\left[\left\|\nabla F(\bar{\theta}^{(t)}) - \sum_{i=1}^n \frac{\lambda_i}{n}\nabla F_i(\theta_i^{(t)})\right\|^2\right] = \mathbb{E}\left[\left\|\sum_{i=1}^n \frac{\lambda_i}{n}\left(\nabla F_i(\bar{\theta}^{(t)}) - \nabla F_i(\theta_i^{(t)})\right)\right\|^2\right] \\
&\leq \frac{\beta^2}{n}\sum_{i=1}^n \lambda_i \mathbb{E}\left[\left\|\bar{\theta}^{(t)} - \theta_i^{(t)}\right\|_2^2\right] = \frac{\beta^2}{n}\mathbb{E}\left[\left\|(I-J)\Theta^{(t)}\right\|_{F,\lambda}^2\right],
\end{aligned}
\tag{20}
$$

where the last inequality follows from Jensen's inequality. Substituting (20) into (16) and it holds that:

$$
\begin{aligned}
\mathbb{E}\left[F(\bar{\theta}^{(t+1)})\right] &\leq \mathbb{E}\left[F(\bar{\theta}^{(t)})\right] + \frac{\alpha\beta^2}{2n^2}\mathbb{E}\left[\left\|(I-J)\Theta^{(t)}\right\|_{F,\lambda}^2\right] + \frac{\alpha(\alpha\beta-1)}{2}\mathbb{E}\left[\left\|\sum_{i=1}^n \frac{\lambda_i}{n}\nabla F_i(\theta_i^{(t)})\right\|^2\right] \\
&\quad - \frac{\alpha}{2}\mathbb{E}\left[\left\|\nabla F(\bar{\theta}^{(t)})\|\right\|^2\right] + \frac{c_\lambda \alpha^2 \beta \upsilon^2}{2} \\
&\stackrel{\alpha \leq \frac{1}{\beta}}{\leq} \mathbb{E}\left[F(\bar{\theta}^{(t)})\right] + \frac{\alpha\beta^2}{2n^2}\mathbb{E}\left[\left\|(I-J)\Theta^{(t)}\right\|_{F,\lambda}^2\right] - \frac{\alpha}{2}\mathbb{E}\left[\left\|\nabla F(\bar{\theta}^{(t)})\right\|^2\right] + \frac{c_\lambda \alpha^2 \beta \upsilon^2}{2}.
\end{aligned}
$$

Then we finish the proof of Eq. (19). □

The following lemma is the descent lemma of strategy II.

**Lemma C.11** (Descent lemma of Strategy II). *Under Assumptions 6.1 and 6.2 and a constant step size $\alpha \leq 1/\beta$, it holds for Algorithm 1 with Strategy II that:*

$$\mathbb{E}[F(\bar{\theta}_\lambda^{(t+1)})] \leq \mathbb{E}[F(\bar{\theta}_\lambda^{(t)})] - \frac{\alpha}{2}\mathbb{E}[\|\nabla F(\bar{\theta}_\lambda^{(t)})\|^2] + \frac{\alpha\beta^2}{2n}\mathbb{E}[\|(I-\Lambda)\Theta^{(t)}\|_{F,\lambda}^2] + \frac{\alpha^2 c_\lambda \beta \upsilon^2}{2}. \tag{21}$$

*Proof.* We consider the term $T_3$ that has been definited in Lemma C.9. It holds under Assumption 6.1 that:

$$
\begin{aligned}
\boldsymbol{T_3} &= \mathbb{E}\left[\left\|\nabla F(\bar{\theta}_\lambda^{(t)}) - \sum_{i=1}^n \frac{\lambda_i}{n}\nabla F_i(\theta_i^{(t)})\right\|^2\right] = \mathbb{E}\left[\left\|\sum_{i=1}^n \frac{\lambda_i}{n}\left(\nabla F_i(\bar{\theta}_\lambda^{(t)}) - \nabla F_i(\theta_i^{(t)})\right)\right\|^2\right] \\
&\leq \frac{\beta^2}{n}\sum_{i=1}^n \lambda_i \mathbb{E}\left[\left\|\bar{\theta}_\lambda^{(t)} - \theta_i^{(t)}\right\|_2^2\right] = \frac{\beta^2}{n}\mathbb{E}\left[\left\|(I-\Lambda)\Theta^{(t)}\right\|_{F,\lambda}^2\right],
\end{aligned}
\tag{22}
$$

Substituting (22) into (16) and it holds that:

$$
\begin{aligned}
\mathbb{E}\left[F(\bar{\theta}_\lambda^{(t+1)})\right] &\leq \mathbb{E}\left[F(\bar{\theta}_\lambda^{(t)})\right] + \frac{\alpha\beta^2}{2n^2}\mathbb{E}\left[\left\|(I-\Lambda)\Theta^{(t)}\right\|_{F,\lambda}^2\right] + \frac{\alpha(\alpha\beta-1)}{2}\mathbb{E}\left[\left\|\sum_{i=1}^n \frac{\lambda_i}{n}\nabla F_i(\theta_i^{(t)})\right\|^2\right] \\
&\quad - \frac{\alpha}{2}\mathbb{E}\left[\left\|\nabla F(\bar{\theta}_\lambda^{(t)})\|\right\|^2\right] + \frac{\alpha^2 c_\lambda \beta \upsilon^2}{2} \\
&\stackrel{\alpha \leq \frac{1}{\beta}}{\leq} \mathbb{E}\left[F(\bar{\theta}_\lambda^{(t)})\right] + \frac{\alpha\beta^2}{2n^2}\mathbb{E}\left[\left\|(I-\Lambda)\Theta^{(t)}\right\|_{F,\lambda}^2\right] - \frac{\alpha}{2}\mathbb{E}\left[\left\|\nabla F(\bar{\theta}_\lambda^{(t)})\right\|^2\right] + \frac{\alpha^2 c_\lambda \beta \upsilon^2}{2}.
\end{aligned}
$$

Then we finish the proof of Eq. (21). □

*Remark* C.12. We observe that, due to the heterogeneous node weights, the consensus error terms in the descent lemmas for both strategies, (20) and (22), naturally take the form of norms in the Hilbert space $L^2(\lambda; \mathbb{R}^d)$. This serves as an important motivation for introducing this weighted Hilbert space.

### C.3. Consensus error analysis

In this subsection, we present the consensus error analysis for Algorithm 1 with different communication strategies.

#### C.3.1. PARAMETER DEVIATIONS AND SPECTRAL NORMS

Firstly, we present the following lemma to characterize the parameter and tracker deviations:

**Lemma C.13.** *The parameter and tracker deviations satisfy the following recursive relations:*

$$\begin{cases} E_{\mathrm{I}}^{(t+1)} = A_{\mathrm{I}} E_{\mathrm{I}}^{(t)} + \alpha B_{\mathrm{I}}^{(t)}, & \text{(Strategy I)}, \\ E_{\mathrm{II}}^{(t+1)} = A_{\mathrm{II}} E_{\mathrm{II}}^{(t)} + \alpha B_{\mathrm{II}}^{(t)}, & \text{(Strategy II)}, \end{cases} \tag{23}$$

*where the matrices involved are defined as follows:*

$$E_{\mathrm{I}}^{(t)} = \begin{bmatrix} (I-J)\Theta^{(t)} \\ \alpha(I-J)Y^{(t)} \end{bmatrix}, \quad E_{\mathrm{II}}^{(t)} = \begin{bmatrix} (I-\Lambda)\Theta^{(t)} \\ \alpha(I-\Lambda)Y^{(t)} \end{bmatrix}, \quad A_{\mathrm{I}} = \begin{bmatrix} W_J & -W_J \\ \mathbf{0} & W_J \end{bmatrix}, \quad A_{\mathrm{II}} = \begin{bmatrix} W_\Lambda & -W_\Lambda \\ \mathbf{0} & W_\Lambda \end{bmatrix},$$

$$B_{\mathrm{I}}^{(t)} = \begin{bmatrix} \mathbf{0} \\ (I-J)D_\lambda\big[\nabla F_\xi(\Theta^{(t+1)}) - \nabla F_\xi(\Theta^{(t)})\big] \end{bmatrix}, \quad B_{\mathrm{II}}^{(t)} = \begin{bmatrix} \mathbf{0} \\ (I-\Lambda)\big[\nabla F_\xi(\Theta^{(t+1)}) - \nabla F_\xi(\Theta^{(t)})\big] \end{bmatrix}.$$

*Proof.* For Strategy I, we have the following update form:

$$\begin{bmatrix} \Theta^{(t+1)} \\ \alpha \cdot Y^{(t+1)} \end{bmatrix} = \begin{bmatrix} W^{\mathrm{ds}} & -W^{\mathrm{ds}} \\ \mathbf{0} & W^{\mathrm{ds}} \end{bmatrix} \begin{bmatrix} \Theta^{(t)} \\ \alpha \cdot Y^{(t)} \end{bmatrix} + \alpha \begin{bmatrix} \mathbf{0} \\ D_\lambda[\nabla F_\xi(\Theta^{(t+1)}) - \nabla F_\xi(\Theta^{(t)})] \end{bmatrix}. \tag{24}$$

Then, we have the following equations:

$$(I-J)^2 = I - 2J + J^2 = I - 2J + J = I - J,$$

$$(I-J)^2 W^{\mathrm{ds}} \overset{\mathbf{1}W^{\mathrm{ds}}=\mathbf{1}}{=} (I-J)(W^{\mathrm{ds}} - J) \overset{W^{\mathrm{ds}}\mathbf{1}=\mathbf{1}}{=} (I-J)W^{\mathrm{ds}}(I-J) = W_J(I-J).$$

Multiplying both sides of (24) by $(I-J)^2$, we obatin:

$$\underbrace{\begin{bmatrix} (I-J)\Theta^{(t+1)} \\ \alpha \cdot (I-J)Y^{(t+1)} \end{bmatrix}}_{E_{\mathrm{I}}^{(t+1)}} = \underbrace{\begin{bmatrix} W_J & -W_J \\ \mathbf{0} & W_J \end{bmatrix}}_{A_{\mathrm{I}}} \underbrace{\begin{bmatrix} (I-J)\Theta^{(t)} \\ \alpha \cdot (I-J)Y^{(t)} \end{bmatrix}}_{E_{\mathrm{I}}^{(t)}} + \alpha \underbrace{\begin{bmatrix} \mathbf{0} \\ (I-J)D_\lambda[\nabla F_\xi(\Theta^{(t+1)}) - \nabla F_\xi(\Theta^{(t)})] \end{bmatrix}}_{B_{\mathrm{I}}^{(t)}}.$$

Similarly for Strategy II, we can give the following update from (2):

$$\begin{bmatrix} \Theta^{(t+1)} \\ \alpha \cdot Y^{(t+1)} \end{bmatrix} = \begin{bmatrix} W & -W \\ \mathbf{0} & W \end{bmatrix} \begin{bmatrix} \Theta^{(t)} \\ \alpha \cdot Y^{(t)} \end{bmatrix} + \alpha \begin{bmatrix} \mathbf{0} \\ \nabla F_\xi(\Theta^{(t+1)}) - \nabla F_\xi(\Theta^{(t)}) \end{bmatrix}. \tag{25}$$

Then, we have the following equations:

$$(I-\Lambda)^2 = I - 2\Lambda + \Lambda^2 = I - 2\Lambda + \frac{\mathbf{1}\lambda}{n} \cdot \frac{\mathbf{1}\lambda}{n} \overset{\lambda^\top \cdot \mathbf{1}=n}{=} I - \Lambda,$$

$$(I-\Lambda)^2 W \overset{\lambda^\top W=\lambda}{=} (I-\Lambda)(W - \Lambda) \overset{W\mathbf{1}=\mathbf{1}}{=} (I-\Lambda)W(I-\Lambda) = W_\Lambda(I-\Lambda).$$

Multiplying both sides of (25) by $(I-\Lambda)^2$ and substituting the two equations above, we arrive at the desired result:

$$\underbrace{\begin{bmatrix} (I-\Lambda)\Theta^{(t+1)} \\ \alpha \cdot (I-\Lambda)Y^{(t+1)} \end{bmatrix}}_{E_{\mathrm{II}}^{(t+1)}} = \underbrace{\begin{bmatrix} W_\Lambda & -W_\Lambda \\ \mathbf{0} & W_\Lambda \end{bmatrix}}_{A_{\mathrm{II}}} \underbrace{\begin{bmatrix} (I-\Lambda)\Theta^{(t)} \\ \alpha \cdot (I-\Lambda)Y^{(t)} \end{bmatrix}}_{E_{\mathrm{II}}^{(t)}} + \alpha \underbrace{\begin{bmatrix} \mathbf{0} \\ (I-\Lambda)[\nabla F_\xi(\Theta^{(t+1)}) - \nabla F_\xi(\Theta^{(t)})] \end{bmatrix}}_{B_{\mathrm{II}}^{(t)}}.$$

Thus Eq. (23) holds for both strategies. $\square$

From Lemma C.13, we obtain the following recursion:

$$
\begin{cases}
E_{\mathrm{II}}^{(t)} = A_{\mathrm{II}}^t E_{\mathrm{II}}^{(0)} + \alpha \sum_{j=1}^{t} A_{\mathrm{II}}^{t-j} B_{\mathrm{II}}^{(j-1)}, & \text{(Strategy II)}, \\
E_{\mathrm{I}}^{(t)} = A_{\mathrm{I}}^t E_{\mathrm{I}}^{(0)} + \alpha \sum_{j=1}^{t} A_{\mathrm{I}}^{t-j} B_{\mathrm{I}}^{(j-1)}, & \text{(Strategy I)}.
\end{cases}
$$

We next examine the spectral norms of $A_{\mathrm{II}}$ and $A_{\mathrm{I}}$. While both matrices have spectral radius strictly less than one, their spectral norms do not necessarily satisfy this property. The following proposition makes this distinction precise.

**Proposition C.14.** *Let $A_{\mathrm{II}}$ and $A_{\mathrm{I}}$ be as defined in Lemma C.13. Then (i) $\rho(A_{\mathrm{II}}) < 1$ and $\rho(A_{\mathrm{I}}) < 1$; (ii) for any integer $t \geq 1$, the $\lambda$-weighted spectral norms satisfy*

$$
\|A_{\mathrm{II}}^t\|_\lambda^2 = \frac{2 + t^2 + t\sqrt{t^2 + 4}}{2} \rho_\Lambda^{2t}, \quad \|A_{\mathrm{I}}^t\|_\lambda^2 \leq \frac{2 + t^2 + t\sqrt{t^2 + 4}}{2} \kappa_\lambda^2 \rho_J^{2t}. \tag{26}
$$

*Proof.* We start with the traditional method for computing eigenvalues:

$$
\det(\sigma I - A_{\mathrm{II}}) = \det \begin{bmatrix} \sigma I - W_\Lambda & \sigma I + W_\Lambda \\ \mathbf{0} & \sigma I - W_\Lambda \end{bmatrix} = [\det(\sigma I - W_\Lambda)]^2.
$$

Therefore, $A$ has the same eigenvalues as $W_\Lambda$, each with algebraic multiplicity two, indicating that $\rho(A) = \rho_\Lambda < 1$.

As for the matrix $A_{\mathrm{I}}$, similarly, we can obtain:

$$
\det(\sigma I - A_{\mathrm{I}}) = \det \begin{bmatrix} \sigma I - W_J & \sigma I + W_J \\ \mathbf{0} & \sigma I - W_J \end{bmatrix} = [\det(\sigma I - W_J)]^2.
$$

Hence, $A_{\mathrm{I}}$ shares the same spectrum as $W_J$, and each eigenvalue of $W_J$ appears twice in $A_{\mathrm{I}}$. This implies that the spectral radius of $A_{\mathrm{I}}$ satisfies $\rho(A_{\mathrm{I}}) = \rho_J < 1$.

**Proof of $\|A_{\mathrm{II}}^t\|_\lambda^2$.** Recall that the $\lambda$-weighted spectral norm is defined as

$$
\|M\|_\lambda = \|D_\lambda^{1/2} M D_\lambda^{-1/2}\|_2.
$$

Consequently, in order to evaluate $\|A_{\mathrm{II}}^t\|_\lambda$, it suffices to compute the spectral radius of the right normal matrix of its $\lambda$-similarity transform:

$$
\|A_{\mathrm{II}}^t\|_\lambda^2 = \|A_{\mathrm{II,sim}}^t\|_2^2 = \rho\big((A_{\mathrm{II,sim}}^t)^\top A_{\mathrm{II,sim}}^t\big).
$$

We start by calculating the power of $A$, by simple recursion, one obtains

$$
A_{\mathrm{II}}^t = \begin{bmatrix} W_\Lambda^t & -t W_\Lambda^t \\ \mathbf{0} & W_\Lambda^t \end{bmatrix},
$$

where $t$ is a positive integer. To compute the $\lambda$-weighted spectral norm, we need to obtain the $\lambda$-similarity transform of $A^t$:

$$
A_{\mathrm{II,sim}}^t = \begin{bmatrix} D_\lambda^{\frac{1}{2}} & \mathbf{0} \\ \mathbf{0} & D_\lambda^{\frac{1}{2}} \end{bmatrix} A_{\mathrm{II}}^t \begin{bmatrix} D_\lambda^{-\frac{1}{2}} & \mathbf{0} \\ \mathbf{0} & D_\lambda^{-\frac{1}{2}} \end{bmatrix} = \begin{bmatrix} \widetilde{W}_\Lambda^t & -t \widetilde{W}_\Lambda^t \\ \mathbf{0} & \widetilde{W}_\Lambda^t \end{bmatrix},
$$

where $\widetilde{W}_\Lambda^t = D_\lambda^{\frac{1}{2}} W_\Lambda^t D_\lambda^{-\frac{1}{2}}$. The right normal matrix associated with $A_{\mathrm{II,sim}}^t$ is

$$
(A_{\mathrm{II,sim}}^t)^\top A_{\mathrm{II,sim}}^t = \begin{bmatrix} (\widetilde{W}_\Lambda^t)^\top \widetilde{W}_\Lambda^t & -t(\widetilde{W}_\Lambda^t)^\top \widetilde{W}_\Lambda^t \\ -t(\widetilde{W}_\Lambda^t)^\top \widetilde{W}_\Lambda^t & (1+t^2)(\widetilde{W}_\Lambda^t)^\top \widetilde{W}_\Lambda^t \end{bmatrix}.
$$

We denote $\mathcal{W} = (\widetilde{W}_\Lambda^t)^\top \widetilde{W}_\Lambda^t$ for simplicity. This matrix admits a Kronecker factorization

$$
(A_{\mathrm{II,sim}}^t)^\top A_{\mathrm{II,sim}}^t = \begin{bmatrix} 1 & -t \\ -t & 1+t^2 \end{bmatrix} \otimes \mathcal{W}.
$$

By the spectral property of Kronecker products (Horn & Johnson, 2012), the eigenvalues of $A \otimes B$ are given by the pairwise products of the eigenvalues of $A$ and $B$.

We first compute the eigenvalues of the $2 \times 2$ coefficient matrix $M_t = \begin{bmatrix} 1 & -t \\ -t & 1+t^2 \end{bmatrix}$. Its characteristic polynomial is

$$\det \begin{bmatrix} 1-\mu & -t \\ -t & 1+t^2-\mu \end{bmatrix} = (1-\mu)(1+t^2-\mu) - t^2 = 0,$$

which yields

$$\mu_{1,2} = \frac{t^2 + 2 \pm t\sqrt{t^2+4}}{2}.$$

For the matrix $\mathcal{W} = (\widetilde{W}_\Lambda^t)^\top \widetilde{W}_\Lambda^t$, the spectral mapping theorem together with the invariance of eigenvalues under similarity transformations implies

$$\sigma_i(\mathcal{W}) = \big(\sigma_i(W_\Lambda)\big)^{2t}, \quad i = 1, \ldots, n.$$

Therefore, the eigenvalues of $(A_{\mathrm{II,sim}}^t)^\top A_{\mathrm{II,sim}}^t$ are given by

$$\sigma(A_{\mathrm{II,sim}}^t) = \Big\{ \mu_k(M_t) \cdot \big(\sigma_i(W_\Lambda)\big)^{2t} \big| k = 1, 2, i = 1, \ldots, n \Big\}.$$

In particular, the spectral radius is

$$\rho\big((A_{\mathrm{II,sim}}^t)^\top A_{\mathrm{II,sim}}^t\big) = \max\{|\mu_1|, |\mu_2|\} \cdot \max_i \big(\sigma_i(W_\Lambda)\big)^{2t} = \frac{t^2 + 2 + t\sqrt{t^2+4}}{2} \rho_\Lambda^{2t}.$$

Consequently, the $\lambda$-weighted spectral norm of $A^t$ satisfies

$$\|A_{\mathrm{II}}^t\|_\lambda^2 = \|A_{\mathrm{II,sim}}^t\|_2^2 = \rho((A_{\mathrm{II,sim}}^t)^\top A_{\mathrm{II,sim}}^t) = \frac{t^2 + 2 + t\sqrt{t^2+4}}{2} \rho_\Lambda^{2t},$$

and we obtain $\|A_{\mathrm{II}}\|_\lambda^2 = \frac{3+\sqrt{5}}{2} \rho_\Lambda^{2t} \approx 2.62 \rho_\Lambda^{2t}$, which can be greater than 1 when the network connectivity is poor.

**Proof of** $\|A_{\mathrm{I}}^t\|_\lambda^2$. Since the matrix $W^{\mathrm{ds}}$ is self-adjoint in the standard Euclidean space, we can similarly obtain:

$$\|A_{\mathrm{I}}^t\|_2^2 = \frac{t^2 + 2 + t\sqrt{t^2+4}}{2} \rho_J^{2t}.$$

Therefore, the $\lambda$-weighted norm of $A_{\mathrm{I}}^t$ satisfies:

$$\|A_{\mathrm{I}}^t\|_\lambda^2 = \left\| \begin{bmatrix} D_\lambda^{\frac{1}{2}} & \mathbf{0} \\ \mathbf{0} & D_\lambda^{\frac{1}{2}} \end{bmatrix} A_{\mathrm{I}}^t \begin{bmatrix} D_\lambda^{-\frac{1}{2}} & \mathbf{0} \\ \mathbf{0} & D_\lambda^{-\frac{1}{2}} \end{bmatrix} \right\|_2^2 \leq \left\| \begin{bmatrix} D_\lambda^{\frac{1}{2}} & \mathbf{0} \\ \mathbf{0} & D_\lambda^{\frac{1}{2}} \end{bmatrix} \right\|_2^2 \|A_{\mathrm{I}}^t\|_2^2 \left\| \begin{bmatrix} D_\lambda^{-\frac{1}{2}} & \mathbf{0} \\ \mathbf{0} & D_\lambda^{-\frac{1}{2}} \end{bmatrix} \right\|_2^2$$
$$\leq \kappa_\lambda^2 \frac{t^2 + 2 + t\sqrt{t^2+4}}{2} \rho_J^{2t}.$$

Thus Eq. (26) holds. □

### C.3.2. CONSENSUS ERROR FOR A SINGLE STEP

Then we can present the consensus error for the $t$-th step for both cases. Firstly, the following lemma gives the consensus error of Strategy II.

**Lemma C.15** (Consensus error for Strategy II). *Suppose that Assumptions 6.1, 6.3, and 6.2 hold, and the step size $\alpha$ satisfies that $\alpha \leq \frac{1}{9\beta} \sqrt{\frac{3(1-\rho_\Lambda)}{(2-\rho_\Lambda)}}$. Then for any $t > 0$, the following bounds on $\mathbb{E}\left[ \|E_{\mathrm{II}}^{(t)}\|_{F,\lambda}^2 \right]$ are satisfied:*

$$\mathbb{E}\left[\|E_{\mathrm{II}}^{(t)}\|_{F,\lambda}^2\right] \leq 3\left(2 + t^2 + t\sqrt{t^2+4}\right)\rho_\Lambda^{2t}\left\|E_{\mathrm{II}}^{(0)}\right\|_{F,\lambda}^2 + 27\alpha^2\beta^2\mathbb{E}\left[\sum_{j=0}^{t-1}\left(H_\Lambda^{(t,j)} + \frac{11}{10}H_\Lambda^{(t,j+1)}\right)\left\|(I-\Lambda)\Theta^{(j)}\right\|_{F,\lambda}^2\right]$$

$$+ 2n\alpha^2\mathbb{E}\left[\sum_{j=1}^{t}H_\Lambda^{(t,j)}\left\|\nabla F(\overline{\theta}_\lambda^{(j-1)})\right\|^2\right] + 18n\alpha^2 v^2\sum_{j=1}^{t}\left(h_\Lambda^{(t,j)} + \frac{3}{2}c_\lambda\alpha^2\beta^2 H_\Lambda^{(t,j)}\right).$$

$$(27)$$

where $H_\Lambda^{(t,j)} := ((t-j)^2+1)\rho_\Lambda^{2(t-j)} + \left(\dfrac{\rho_\Lambda(t-j)}{1-\rho_\Lambda}+1\right)\dfrac{\rho_\Lambda^{t-j}}{1-\rho_\Lambda}$ and $h_\Lambda^{(t,j)} := ((t-j)^2+1)\rho_\Lambda^{2(t-j)}$.

*Proof.* From (23), it can be obtained that:

$$E_{\mathrm{II}}^{(t)} = A_{\mathrm{II}}^t E_{\mathrm{II}}^{(0)} + \alpha\sum_{j=1}^{t}A_{\mathrm{II}}^{t-j}B_{\mathrm{II}}(j-1).$$

Then we obtain

$$\mathbb{E}\left[\left\|E_{\mathrm{II}}^{(t)}\right\|_{F,\lambda}^2\right] \overset{lm.C.3}{\leq} (1+2)\|A_{\mathrm{II}}^t\|_\lambda^2\left\|E_{\mathrm{II}}^{(0)}\right\|_{F,\lambda}^2 + (1+\frac{1}{2})\alpha^2\mathbb{E}\left[\left\|\sum_{j=1}^{t}A_{\mathrm{II}}^{t-j}B_{\mathrm{II}}(j-1)\right\|_{F,\lambda}^2\right]$$

$$= 3\|A_{\mathrm{II}}^t\|_\lambda^2\left\|E_{\mathrm{II}}^{(0)}\right\|_{F,\lambda}^2 + \frac{3}{2}\alpha^2\mathbb{E}\left[\left\|\begin{bmatrix} -\sum_{j=1}^{t}(t-j)W_\Lambda^{t-j}I_\Lambda[\nabla F_\xi(\Theta^{(j)}) - \nabla F_\xi(\Theta^{(j-1)})] \\ \sum_{j=1}^{t}W_\Lambda^{t-j}I_\Lambda[\nabla F_\xi(\Theta^{(j)}) - \nabla F_\xi(\Theta^{(j-1)})]\end{bmatrix}\right\|_{F,\lambda}^2\right]$$

$$\leq 3\|A^t\|_\lambda^2\left\|E_{\mathrm{II}}^{(0)}\right\|_{F,\lambda}^2 + \frac{3}{2}\alpha^2\underbrace{\mathbb{E}\left[\left\|\sum_{j=1}^{t}W_\Lambda^{t-j}[\nabla F_\xi(\Theta^{(j)}) - \nabla F_\xi(\Theta^{(j-1)})]\right\|_{F,\lambda}^2\right]}_{T_1}$$

$$+ \frac{3}{2}\alpha^2\underbrace{\mathbb{E}\left[\left\|\sum_{j=1}^{t}(t-j)W_\Lambda^{t-j}[\nabla F_\xi(\Theta^{(j)}) - \nabla F_\xi(\Theta^{(j-1)})]\right\|_{F,\lambda}^2\right]}_{T_2},$$

$$(28)$$

where we use $W_\Lambda^{t-j}I_\Lambda = I_\Lambda W_\Lambda^{t-j}$ and $\|I_\Lambda\|_\lambda \leq 1$. From Spectral Mapping Theorem, we have $\rho(W_\Lambda^{t-j}) = \rho_\Lambda^{t-j}$. For matrix $W_\Lambda$, the $\lambda$-weighted spectral norm coincides with the spectral radius, implying $\|W_\Lambda^{t-j}\|_\lambda = \rho_\Lambda^{t-j}$. Therefore, we obtain

$$T_1 \leq 3\mathbb{E}\left[\left\|\sum_{j=1}^{t}W_\Lambda^{t-j}[\nabla F(\Theta^{(j-1)}) - \nabla F_\xi(\Theta^{(j-1)})]\right\|_{F,\lambda}^2\right] + 3\mathbb{E}\left[\left\|\sum_{j=1}^{t}W_\Lambda^{t-j}[\nabla F(\Theta^{(j)}) - \nabla F_\xi(\Theta^{(j)})]\right\|_{F,\lambda}^2\right]$$

$$+ 3\underbrace{\mathbb{E}\left[\left\|\sum_{j=1}^{t}W_\Lambda^{t-j}[\nabla F(\Theta^{(j)}) - \nabla F(\Theta^{(j-1)})]\right\|_{F,\lambda}^2\right]}_{T_3}.$$

Then from Lemma C.2 and the unbiasedness of the stochastic gradient, it comes that:

$$
\begin{aligned}
T_1 \leq & 3\sum_{j=1}^{t}\left\|W_\Lambda^{t-j}\right\|_\lambda^2 \mathbb{E}\left[\left\|\nabla F(\Theta^{(j-1)}) - \nabla F_\xi(\Theta^{(j-1)})\right\|_{F,\lambda}^2\right] \\
& + 3\sum_{j=1}^{t}\left\|W_\Lambda^{t-j}\right\|_\lambda^2 \mathbb{E}\left[\left\|\nabla F(\Theta^{(j)}) - \nabla F_\xi(\Theta^{(j)})\right\|_{F,\lambda}^2\right] + 3\mathbb{E}[T_3] \\
\leq & 3\sum_{j=1}^{t}\rho_\Lambda^{2(t-j)}n\upsilon^2 + 3\sum_{j=1}^{t}\rho_\Lambda^{2(t-j)}n\upsilon^2 + 3\mathbb{E}[T_3] = 6\sum_{j=1}^{t}\rho_\Lambda^{2(t-j)}n\upsilon^2 + 3\mathbb{E}[T_3],
\end{aligned}
\tag{29}
$$

As for the term $T_3$, it holds that:

$$
\begin{aligned}
T_3 = & \mathbb{E}\left[\left\|\sum_{j=1}^{t}W_\Lambda^{t-j}[\nabla F(\Theta^{(j)}) - \nabla F(\Theta^{(j-1)})]\right\|_{F,\lambda}^2\right] \\
= & \mathbb{E}\left[\sum_{j=1}^{t}\left\|W_\Lambda^{t-j}[\nabla F(\Theta^{(j)}) - \nabla F(\Theta^{(j-1)})]\right\|_{F,\lambda}^2\right] \\
& + \mathbb{E}\left[\underbrace{\sum_{j\neq\iota}\left\langle W_\Lambda^{t-j}[\nabla F(\Theta^{(j)}) - \nabla F(\Theta^{(j-1)})], W_\Lambda^{t-\iota}[\nabla F(\Theta^{(\iota)}) - \nabla F(\Theta^{(\iota-1)})]\right\rangle_{\lambda,d}}_{T_4}\right].
\end{aligned}
\tag{30}
$$

We bound $T_4$:

$$
\begin{aligned}
T_4 = & \sum_{j\neq\iota}^{t}\left\langle W_\Lambda^{t-j}[\nabla F(\Theta^{(j)}) - \nabla F(\Theta^{(j-1)})], W_\Lambda^{t-\iota}[\nabla F(\Theta^{(\iota)}) - \nabla F(\Theta^{(\iota-1)})]\right\rangle_{\lambda,d} \\
\leq & \sum_{j\neq\iota}^{t}\left\|W_\Lambda^{t-j}\right\|_\lambda\left\|\nabla F(\Theta^{(j)}) - \nabla F(\Theta^{(j-1)})\right\|_{F,\lambda}\left\|W_\Lambda^{t-\iota}\right\|_\lambda\left\|\nabla F(\Theta^{(\iota)}) - \nabla F(\Theta^{(\iota-1)})\right\|_{F,\lambda} \\
\leq & \sum_{j\neq\iota}^{t}\rho_\Lambda^{2t-(j+\iota)}\frac{\left\|\nabla F(\Theta^{(j)}) - \nabla F(\Theta^{(j-1)})\right\|_{F,\lambda}^2}{2} + \sum_{j\neq\iota}^{t}\rho_\Lambda^{2t-(j+\iota)}\frac{\left\|\nabla F(\Theta^{(\iota)}) - \nabla F(\Theta^{(\iota-1)})\right\|_{F,\lambda}^2}{2} \\
= & \sum_{j\neq\iota}^{t}\rho_\Lambda^{2t-(j+\iota)}\left\|\nabla F(\Theta^{(j)}) - \nabla F(\Theta^{(j-1)})\right\|_{F,\lambda}^2 = \sum_{j=1}^{t}\sum_{\substack{\iota=1\\\iota\neq j}}^{t}\rho_\Lambda^{2t-(j+\iota)}\left\|\nabla F(\Theta^{(j)}) - \nabla F(\Theta^{(j-1)})\right\|_{F,\lambda}^2 \\
\leq & \frac{1}{1-\rho_\Lambda}\sum_{j=1}^{t}\rho_\Lambda^{t-j}\left\|\nabla F(\Theta^{(j)}) - \nabla F(\Theta^{(j-1)})\right\|_{F,\lambda}^2.
\end{aligned}
\tag{31}
$$

Combing (30) and (31), we can get the upper bound for $\mathbb{E}[T_3]$:

$$
T_3 \leq \sum_{j=1}^{t}\left(\rho_\Lambda^{2(t-j)} + \frac{1}{1-\rho_\Lambda}\rho_\Lambda^{t-j}\right)\mathbb{E}\left[\left\|\nabla F(\Theta^{(j)}) - \nabla F(\Theta^{(j-1)})\right\|_{F,\lambda}^2\right].
\tag{32}
$$

Similarly, we can bound $T_2$ as:

$$
T_2 \leq 6\sum_{j=1}^{t}(t-j)^2\rho_\Lambda^{2(t-j)}n\upsilon^2 + 3\mathbb{E}\left[\underbrace{\left\|\sum_{j=1}^{t}(t-j)W_\Lambda^{t-j}[\nabla F(\Theta^{(j)}) - \nabla F(\Theta^{(j-1)})]\right\|_{F,\lambda}^2}_{T_3'}\right]
\tag{33}
$$

As for $T_3'$, we have

$$
\begin{aligned}
T_3' \leq &\sum_{j=1}^{t}(t-j)^2 \rho_\Lambda^{2(t-j)} \left\| \nabla F(\Theta^{(j)}) - \nabla F(\Theta^{(j-1)}) \right\|_{F,\lambda}^2 \\
&+ \sum_{j=1}^{t}(t-j)\sum_{\substack{\iota=1 \\ \iota \neq j}}^{t}(t-\iota)\rho_\Lambda^{2t-(j+\iota)} \left\| \nabla F(\Theta^{(j)}) - \nabla F(\Theta^{(j-1)}) \right\|_{F,\lambda}^2 \\
\leq &\sum_{j=1}^{t}\left((t-j)^2 \rho_\Lambda^{2(t-j)} + \frac{\rho_\Lambda}{(1-\rho_\Lambda)^2}(t-j)\rho_\Lambda^{t-j}\right) \left\| \nabla F(\Theta^{(j)}) - \nabla F(\Theta^{(j-1)}) \right\|_{F,\lambda}^2.
\end{aligned}
\tag{34}
$$

Therefore, it holds from (29), (32), (33), and (34) that

$$
\begin{aligned}
T_1 + T_2 \leq &6\sum_{j=1}^{t}((t-j)^2+1)\rho_\Lambda^{2(t-j)}nv^2 \\
&+ 3\sum_{j=1}^{t}\left[((t-j)^2+1)\rho_\Lambda^{2(t-j)} + \left(\frac{\rho_\Lambda(t-j)}{1-\rho_\Lambda}+1\right)\frac{\rho_\Lambda^{t-j}}{1-\rho_\Lambda}\right]\mathbb{E}\left[\underbrace{\left\| \nabla F(\Theta^{(j)}) - \nabla F(\Theta^{(j-1)}) \right\|_{F,\lambda}^2}_{T_5}\right] \\
\leq &3\sum_{j=1}^{t}H_\Lambda^{(t,j)}\mathbb{E}[T_5] + 6\sum_{j=1}^{t}h_\Lambda^{(t,j)}nv^2,
\end{aligned}
\tag{35}
$$

where we denote $H_\Lambda^{(t,j)} := ((t-j)^2+1)\rho_\Lambda^{2(t-j)} + \left(\frac{\rho_\Lambda(t-j)}{1-\rho_\Lambda}+1\right)\frac{\rho_\Lambda^{t-j}}{1-\rho_\Lambda}$ and $h_\Lambda^{(t,j)} := ((t-j)^2+1)\rho_\Lambda^{2(t-j)}$.

As for the term $T_5$, we can derive

$$
\begin{aligned}
T_5 \leq &3\left\| \nabla F(\Theta^{(j)}) - \nabla F(\overline{\Theta}_\lambda(j)) \right\|_{F,\lambda}^2 + 3\left\| \nabla F(\Theta^{(j-1)}) - \nabla F(\overline{\Theta}_\lambda^{(j-1)}) \right\|_{F,\lambda}^2 + 3\left\| \nabla F(\overline{\Theta}_\lambda^{(j-1)}) - \nabla F(\overline{\Theta}_\lambda(j)) \right\|_{F,\lambda}^2 \\
\leq &3\beta^2\left[\left\| (I-\Lambda)\Theta^{(j)} \right\|_{F,\lambda}^2 + \left\| (I-\Lambda)\Theta^{(j-1)} \right\|_{F,\lambda}^2 + \left\| \overline{\Theta}_\lambda(j) - \overline{\Theta}_\lambda^{(j-1)} \right\|_{F,\lambda}^2\right].
\end{aligned}
\tag{36}
$$

We consider the last term of (36) and obtain that:

$$
\begin{aligned}
\mathbb{E}\left[\left\| \overline{\Theta}_\lambda(j) - \overline{\Theta}_\lambda^{(j-1)} \right\|_{F,\lambda}^2\right] &= \sum_{i=1}^{n}\lambda_i \mathbb{E}\left[\left\| \overline{\theta}_\lambda(j) - \overline{\theta}_\lambda^{(j-1)} \right\|^2\right] = n\alpha^2 \mathbb{E}\left[\left\| \sum_{i=1}^{n}\frac{\lambda_i}{n}g_i^{(j-1)} \right\|^2\right] \\
&\leq n\alpha^2 \mathbb{E}\left[\left\| \sum_{i=1}^{n}\frac{\lambda_i}{n}\nabla F_i(\theta_i^{(j-1)}) \right\|^2 + \left\| \sum_{i=1}^{n}\frac{\lambda_i}{n}[g_i^{(j-1)} - \nabla F_i(\theta_i^{(j-1)})] \right\|^2\right] \\
&\leq n\alpha^2 \mathbb{E}\left[\left\| \sum_{i=1}^{n}\frac{\lambda_i}{n}\nabla F_i(\theta_i^{(j-1)}) \right\|^2\right] + c_\lambda n\alpha^2 v^2 \\
&\leq 2n\alpha^2\left(\mathbb{E}\left[\left\| \sum_{i=1}^{n}\frac{\lambda_i}{n}\left[\nabla F_i(\theta_i^{(j-1)}) - \nabla F_i(\overline{\theta}_\lambda^{(j-1)})\right] \right\|^2 + \left\| \nabla F_\lambda(\overline{\theta}_\lambda^{(j-1)}) \right\|^2\right] + \frac{c_\lambda v^2}{2}\right)
\end{aligned}
\tag{37}
$$

Combining (36) and (37) together, it holds that:

$$
\begin{aligned}
\mathbb{E}[T_5] = &3\beta^2 \mathbb{E}\left[\left\| (I-\Lambda)\Theta^{(j)} \right\|_{F,\lambda}^2\right] + 3\beta^2(1+2\alpha^2\beta^2)\mathbb{E}\left[\left\| (I-\Lambda)\Theta^{(j-1)} \right\|_{F,\lambda}^2\right] + 6n\alpha^2\beta^2 \mathbb{E}\left[\left\| \nabla F_\lambda(\overline{\theta}_\lambda^{(j-1)}) \right\|^2\right] \\
&+ 3c_\lambda n\alpha^2\beta^2 v^2.
\end{aligned}
\tag{38}
$$

Substituting (38) and (35) into (28), it holds that:

$$
\mathbb{E}\left[\|E^{(t)}\|_{F,\lambda}^2\right] \leq 3\left\|A^t\right\|_\lambda^2 \left\|E_{\mathrm{II}}^{(0)}\right\|_{F,\lambda}^2 + \frac{27}{2}\alpha^2\beta^2\mathbb{E}\left[\sum_{j=0}^{t-1}\left(H_\Lambda^{(t,j)} + (1+2\alpha^2\beta^2)H^{(t,j+1)}\right)\left\|(I-\Lambda)\Theta^{(j)}\right\|_{F,\lambda}^2\right]
$$

$$
+ \frac{27(2-\rho_\Lambda)}{2(1-\rho_\Lambda)}\alpha^2\beta^2\mathbb{E}\left[\left\|(I-\Lambda)\Theta^{(t)}\right\|_{F,\lambda}^2\right] + 27n\alpha^4\beta^2\mathbb{E}\left[\sum_{j=1}^{t}H_\Lambda^{(t,j)}\left\|\nabla F_\lambda(\overline{\theta}_\lambda^{(j-1)})\right\|^2\right] \tag{39}
$$

$$
+ 9n\alpha^2\upsilon^2\sum_{j=1}^{t}\left(h_\Lambda^{(t,j)} + \frac{3}{2}c_\lambda\alpha^2\beta^2 H_\Lambda^{(t,j)}\right).
$$

Furthermore, by noting that $\mathbb{E}\left[\left\|(I-\Lambda)\Theta^{(t)}\right\|_{F,\lambda}^2\right] \leq \mathbb{E}\left[\left\|E^{(t)}\right\|_{F,\lambda}^2\right]$ and assuming that $\alpha \leq \frac{1}{9\beta}\sqrt{\frac{3(1-\rho_\Lambda)}{2-\rho_\Lambda}}$, we can further derive from (39) that:

$$
\mathbb{E}\left[\|E(t)\|_{F,\lambda}^2\right] \leq 3\left(2+t^2+t\sqrt{t^2+4}\right)\rho_\Lambda^{2t}\left\|E_{\mathrm{II}}^{(0)}\right\|_{F,\lambda}^2 + 27\alpha^2\beta^2\mathbb{E}\left[\sum_{j=0}^{t-1}\left(H_\Lambda^{(t,j)} + \frac{11}{10}H^{(t,j+1)}\right)\left\|(I-\Lambda)\Theta^{(j)}\right\|_{F,\lambda}^2\right]
$$

$$
+ 2n\alpha^2\mathbb{E}\left[\sum_{j=1}^{t}H_\Lambda^{(t,j)}\left\|\nabla F(\overline{\theta}_\lambda^{(j-1)})\right\|^2\right] + 18n\alpha^2\upsilon^2\sum_{j=1}^{t}\left(h_\Lambda^{(t,j)} + \frac{3}{2}c_\lambda\alpha^2\beta^2 H_\Lambda^{(t,j)}\right).
$$

Thus we prove that (27) holds. $\qquad\square$

The following lemma present the single-step consensus error for Strategy I.

**Lemma C.16** (Consensus error for Strategy I)**.** *Suppose that Assumptions 6.1, 6.3, and 6.2 hold, and the step size $\alpha$ satisfies that $\alpha \leq \frac{1}{9\beta}\sqrt{\frac{3(1-\rho_J)}{(2-\rho_J)\lambda_{\max}^2}}$. Then for any $t > 0$, the following bounds on $\mathbb{E}\left[\|E_{\mathrm{I}}^{(t)}\|_{F,\lambda}^2\right]$ are satisfied:*

$$
\mathbb{E}\left[\left\|E_{\mathrm{I}}^{(t)}\right\|_{F,\lambda}^2\right] \leq 3\kappa_\lambda^2\left(2+t^2+t\sqrt{t^2+4}\right)\rho_J^{2t}\left\|E_{\mathrm{I}}^{(0)}\right\|_{F,\lambda}^2 + 27\alpha^2\beta^2\mathbb{E}\left[\sum_{j=0}^{t-1}\left(H_J^{(t,j)} + \frac{11}{10}H_J^{(t,j+1)}\right)\left\|(I-J)\Theta^{(j)}\right\|_{F,\lambda}^2\right]
$$

$$
+ 2n\alpha^2\mathbb{E}\left[\sum_{j=1}^{t}H_J^{(t,j)}\left\|\nabla F(\overline{\theta}_\lambda^{(j-1)})\right\|^2\right] + 18n\alpha^2\upsilon^2\sum_{j=1}^{t}\left(h_J^{(t,j)} + \frac{3}{2}c_\lambda\alpha^2\beta^2 H_J^{(t,j)}\right),
$$

$$\tag{40}$$

*where* $H_J^{(t,j)} := \lambda_{\max}^2\left[((t-j)^2+1)\rho_J^{2(t-j)} + \left(\frac{\rho_J(t-j)}{1-\rho_J}+1\right)\frac{\rho_J^{t-j}}{1-\rho_J}\right]$ *and* $h_J^{(t,j)} := \lambda_{\max}^2((t-j)^2+1)\rho_J^{2(t-j)}$.

*Proof.* Similar to that of Strategy II, we can derive:

$$
\mathbb{E}\left[\left\|E_{\mathrm{I}}^{(t)}\right\|_{F,\lambda}^2\right] \leq 3\|A_{\mathrm{I}}^t\|_\lambda^2\left\|E_{\mathrm{I}}^{(0)}\right\|_{F,\lambda}^2 + \frac{3}{2}\alpha^2\mathbb{E}\left[\left\|\sum_{j=1}^{t}A^{t-j}B(j-1)\right\|_{F,\lambda}^2\right]
$$

$$
\leq 3\|A_{\mathrm{I}}^t\|_\lambda^2\left\|E_{\mathrm{I}}^{(0)}\right\|_{F,\lambda}^2 + \frac{3}{2}\alpha^2\underbrace{\mathbb{E}\left[\left\|\sum_{j=1}^{t}W_J^{t-j}I_J D_\lambda[\nabla F_\xi(\Theta^{(j)}) - \nabla F_\xi(\Theta^{(j-1)})]\right\|_{F,\lambda}^2\right]}_{T_1} \tag{41}
$$

$$
+ \frac{3}{2}\alpha^2\underbrace{\mathbb{E}\left[\left\|\sum_{j=1}^{t}(t-j)W_J^{t-j}I_J D_\lambda[\nabla F_\xi(\Theta^{(j)}) - \nabla F_\xi(\Theta^{(j-1)})]\right\|_{F,\lambda}^2\right]}_{T_2},
$$

For the term $T_1$, we can present an upper bound as follows:

$$
\begin{aligned}
T_1 =& \mathbb{E}\left[\left\|\sum_{j=1}^{t} W_J^{t-j} I_J D_\lambda [\nabla F_\xi(\Theta^{(j)}) - \nabla F_\xi(\Theta^{(j-1)})]\right\|_{F,\lambda}^2\right] \\
\leq& \sum_{j=1}^{t} \left\|W_J^{t-j} I_J D_\lambda\right\|_\lambda^2 \mathbb{E}\left[\left\|\nabla F_\xi(\Theta^{(j)}) - \nabla F_\xi(\Theta^{(j-1)})\right\|_{F,\lambda}^2\right] \\
&+ \mathbb{E}\left[\underbrace{\sum_{j\neq\iota} \left\langle W_J^{t-j} I_J D_\lambda [\nabla F(\Theta^{(j)}) - \nabla F(\Theta^{(j-1)})], W_J^{t-\iota} I_J D_\lambda [\nabla F(\Theta^{(\iota)}) - \nabla F(\Theta^{(\iota-1)})]\right\rangle_{\lambda,d}}_{T_4}\right].
\end{aligned}
\tag{42}
$$

The proof is essentially identical to that of Strategy II, except that every $W_\Lambda$ term is replaced by $W_J$, and an extra factor $I_J D_\lambda$ is applied. Consequently, the argument reduces to bounding the corresponding $\lambda$-weighted spectral norm. Applying Lemma C.5, we obtain:

$$
\|W_J^{t-j} I_J D_\lambda\|_\lambda \leq \lambda_{\max}\rho_J^{t-j}, \quad \|(t-j) W_J^{t-j} I_J D_\lambda\|_\lambda \leq (t-j)\lambda_{\max}\rho_J^{t-j}.
$$

If $\alpha$ satisfies that $\alpha \leq \dfrac{1}{9\beta}\sqrt{\dfrac{3(1-\rho_J)}{(2-\rho_J)\lambda_{\max}^2}}$, we can obtain from (41) and (42) that:

$$
\begin{aligned}
\mathbb{E}\left[\left\|E_{\mathrm{I}}^{(t)}\right\|_{F,\lambda}^2\right] \leq& 3\kappa_\lambda^2\left(2 + t^2 + t\sqrt{t^2+4}\right)\rho_J^{2t}\left\|E_{\mathrm{I}}^{(0)}\right\|_{F,\lambda}^2 + 27\alpha^2\beta^2\mathbb{E}\left[\sum_{j=0}^{t-1}\left(H_J^{(t,j)} + \frac{11}{10}H_J^{(t,j+1)}\right)\left\|(I-J)\Theta^{(j)}\right\|_{F,\lambda}^2\right] \\
&+ 2n\alpha^2\mathbb{E}\left[\sum_{j=1}^{t} H_J^{(t,j)}\left\|\nabla F(\overline{\theta}_\lambda^{(j-1)})\right\|^2\right] + 18n\alpha^2\upsilon^2\sum_{j=1}^{t}\left(h_J^{(t,j)} + \frac{3}{2}c_\lambda\alpha^2\beta^2 H_J^{(t,j)}\right),
\end{aligned}
$$

where $H_J^{(t,j)} := ((t-j)^2+1)\rho_J^{2(t-j)} + \left(\dfrac{\rho_J(t-j)}{1-\rho_J} + 1\right)\dfrac{\rho_J^{t-j}}{1-\rho_J}$ and $h_J^{(t,j)} := ((t-j)^2+1)\rho_J^{2(t-j)}$. Thus (40) holds. $\square$

### C.3.3. TOTAL CONSENSUS ERROR ANALYSIS

Finally, we can present total consensus error analysis and complete the proof of Proposition 6.4.

*Proof.* **Proof of Strategy II.** By taking the summation on both sides of Eq. (27) from $t=0$ to $T-1$, we obtain:

$$
\begin{aligned}
&\sum_{t=0}^{T-1}\mathbb{E}\left[\|E_{\mathrm{II}}^{(t)}\|_{F,\lambda}^2\right] \\
\leq& 6\sum_{t=0}^{T-1}\frac{2+t^2+t\sqrt{t^2+4}}{2}\rho_\Lambda^{2t}\left\|E_{\mathrm{II}}^{(0)}\right\|_{F,\lambda}^2 + 27\alpha^2\beta^2\sum_{t=0}^{T-1}\sum_{j=0}^{t-1}\left(H_\Lambda^{(t,j)} + \frac{11}{10}H^{(t,j+1)}\right)\mathbb{E}\left[\left\|(I-\Lambda)\Theta^{(j)}\right\|_{F,\lambda}^2\right] \\
&+ 2n\alpha^2\sum_{t=0}^{T-1}\sum_{j=1}^{t} H_\Lambda^{(t,j)}\mathbb{E}\left[\left\|\nabla F(\overline{\theta}_\lambda^{(-1)})\right\|^2\right] + 18n\alpha^2\upsilon^2\sum_{t=0}^{T-1}\sum_{j=1}^{t}\left(h_\Lambda^{(t,j)} + \frac{3}{2}c_\lambda\alpha^2\beta^2 H_\Lambda^{(t,j)}\right) \\
\leq& 6\underbrace{\sum_{t=0}^{T-1}\frac{2+t^2+t\sqrt{t^2+4}}{2}\rho_\Lambda^{2t}\left\|E_{\mathrm{II}}^{(0)}\right\|_{F,\lambda}^2}_{T_1} + 27\alpha^2\beta^2\underbrace{\sum_{j=0}^{T-2}\mathbb{E}[\|(I-\Lambda)\Theta^{(j)}\|_{F,\lambda}^2]\sum_{t=j+1}^{T-1}\left(H_\Lambda^{(t,j)} + \frac{11}{10}H^{(t,j+1)}\right)}_{T_2} \\
&+ 2n\alpha^2\underbrace{\sum_{j=1}^{T-1}\mathbb{E}\left[\left\|\nabla F(\overline{\theta}_\lambda^{(-1)})\right\|^2\right]\sum_{t=j}^{T-1} H_\Lambda^{(t,j)}}_{T_3} + 18n\alpha^2\upsilon^2\underbrace{\sum_{t=0}^{T-1}\sum_{j=1}^{t}\left(h_\Lambda^{(t,j)} + \frac{3}{2}c_\lambda\alpha^2\beta^2 H_\Lambda^{(t,j)}\right)}_{T_4}.
\end{aligned}
\tag{43}
$$

We first consider the term $T_1$. From (10) and (11) it holds that:

$$T_1 = \sum_{t=0}^{T-1} \frac{2 + t^2 + t\sqrt{t^2+4}}{2} \rho_\Lambda^{2t} \leq \sum_{t=0}^{\infty} (t^2+3)\rho_\Lambda^{2t} = \frac{\rho_\Lambda^2(1+\rho_\Lambda^2)}{(1-\rho_\Lambda^2)^3} + \frac{3}{(1-\rho_\Lambda^2)} = \frac{4\rho_\Lambda^4 - 5\rho_\Lambda^2 + 3}{(1-\rho_\Lambda^2)^3}. \tag{44}$$

Then, the term $T_2$ holds from (10) and (11) that:

$$
\begin{aligned}
T_2 &= \sum_{t=j+1}^{T-1} \left( H_\Lambda^{(t,j)} + \frac{11}{10} H^{(t,j+1)} \right) \\
&\leq \frac{11}{10} \sum_{m=1}^{T-j-1} \left( (m^2+1)\rho_\Lambda^{2m} + \left( \frac{\rho_\Lambda m}{1-\rho_\Lambda} + 1 \right) \frac{\rho_\Lambda^m}{1-\rho_\Lambda} + ((m-1)^2+1)\rho_\Lambda^{2(m-1)} + \left( \frac{\rho_\Lambda(m-1)}{1-\rho_\Lambda} + 1 \right) \frac{\rho_\Lambda^{(m-1)}}{1-\rho_\Lambda} \right) \\
&\leq \frac{11}{5} \sum_{m=0}^{T-j-1} \left( (m^2+1)\rho_\Lambda^{2m} + \left( \frac{\rho_\Lambda m}{1-\rho_\Lambda} + 1 \right) \frac{\rho_\Lambda^m}{1-\rho_\Lambda} \right) \\
&\leq \frac{11}{5} \left( \frac{\rho_\Lambda^2(1+\rho_\Lambda^2)}{(1-\rho_\Lambda^2)^3} + \frac{1}{1-\rho_\Lambda^2} + \frac{\rho_\Lambda^2}{(1-\rho_\Lambda)^4} + \frac{1}{(1-\rho_\Lambda)^2} \right) \leq \frac{22}{5} \cdot \frac{1+3\rho_\Lambda^4}{(1-\rho_\Lambda^2)^3(1-\rho_\Lambda)}.
\end{aligned}
\tag{45}
$$

The term $T_3$ holds from (10) and (11) that:

$$
\begin{aligned}
T_3 &= \sum_{t=j}^{T-1} H_\Lambda^{(t,j)} = \sum_{t=j}^{T-1} \left( ((t-j)^2+1)\rho_\Lambda^{2(t-j)} + \left( \frac{\rho_\Lambda(t-j)}{1-\rho_\Lambda} + 1 \right) \frac{\rho_\Lambda^{t-j}}{1-\rho_\Lambda} \right) \\
&= \sum_{m=0}^{T-j-1} \left( (m^2+1)\rho_\Lambda^{2m} + \left( \frac{\rho_\Lambda m}{1-\rho_\Lambda} + 1 \right) \frac{\rho_\Lambda^m}{1-\rho_\Lambda} \right) \\
&\leq \left( \frac{\rho_\Lambda^2(1+\rho_\Lambda^2)}{(1-\rho_\Lambda^2)^3} + \frac{1}{1-\rho_\Lambda^2} + \frac{\rho_\Lambda^2}{(1-\rho_\Lambda)^4} + \frac{1}{(1-\rho_\Lambda)^2} \right) \leq \frac{2(1+3\rho_\Lambda^4)}{(1-\rho_\Lambda^2)^3(1-\rho_\Lambda)}.
\end{aligned}
\tag{46}
$$

The last term $T_4$ holds from (10), (11), and (15) that:

$$
\begin{aligned}
T_4 &= \sum_{t=0}^{T-1} \sum_{j=1}^{t} \left( h_\Lambda^{(t,j)} + \frac{3}{2} c_\lambda \alpha^2 \beta^2 H_\Lambda^{(t,j)} \right) \\
&= \sum_{t=0}^{T-1} \sum_{j=1}^{t} \left( ((t-j)^2+1)\rho_\Lambda^{2(t-j)} + \frac{3}{2} c_\lambda \alpha^2 \beta^2 \left( ((t-j)^2+1)\rho_\Lambda^{2(t-j)} + \left( \frac{\rho_\Lambda(t-j)}{1-\rho_\Lambda} + 1 \right) \frac{\rho_\Lambda^{t-j}}{1-\rho_\Lambda} \right) \right) \\
&= \sum_{k=0}^{T-2} \sum_{t=k+1}^{T-1} \left( (k^2+1)\rho_\Lambda^{2k} + \frac{3}{2} c_\lambda \alpha^2 \beta^2 \left( (k^2+1)\rho_\Lambda^{2k} + \left( \frac{\rho_\Lambda k}{1-\rho_\Lambda} + 1 \right) \frac{\rho_\Lambda^k}{1-\rho_\Lambda} \right) \right) \\
&= \sum_{k=0}^{T-2} (T-k-1) \left( (k^2+1)\rho_\Lambda^{2k} + \frac{3}{2} c_\lambda \alpha^2 \beta^2 \left( (k^2+1)\rho_\Lambda^{2k} + \left( \frac{\rho_\Lambda k}{1-\rho_\Lambda} + 1 \right) \frac{\rho_\Lambda^k}{1-\rho_\Lambda} \right) \right) \\
&\leq \left( \frac{\rho_\Lambda^2(1+\rho_\Lambda^2)}{(1-\rho_\Lambda^2)^3} + \frac{1}{1-\rho_\Lambda^2} + \frac{3}{2} c_\lambda \alpha^2 \beta^2 \left( \frac{\rho_\Lambda^2(1+\rho_\Lambda^2)}{(1-\rho_\Lambda^2)^3} + \frac{1}{1-\rho_\Lambda^2} + \frac{\rho_\Lambda^2}{(1-\rho_\Lambda)^4} + \frac{1}{(1-\rho_\Lambda)^2} \right) \right) T \\
&\leq \left( \frac{1+\rho_\Lambda^2}{(1-\rho_\Lambda^2)^3} + \frac{3 c_\lambda \alpha^2 \beta^2 (1+3\rho_\Lambda^4)}{(1-\rho_\Lambda^2)^3(1-\rho_\Lambda)} \right) T
\end{aligned}
\tag{47}
$$

For simplicity of notation, we denote $A(\rho_\Lambda) := \dfrac{1+\rho_\Lambda^2}{(1-\rho_\Lambda^2)^3}$ and $B(\rho_\Lambda) := \dfrac{2(1+3\rho_\Lambda^4)}{(1-\rho_\Lambda^2)^3(1-\rho_\Lambda)}$. Plugging (44), (45), (46),

and (47) into (43), then we can derive:

$$
\sum_{t=0}^{T-1} \mathbb{E}\left[\left\|E_{\mathrm{II}}^{(t)}\right\|_{F,\lambda}^2\right] \leq 6\frac{4\rho_\Lambda^4 - 5\rho_\Lambda^2 + 3}{(1-\rho_\Lambda^2)^3}\left\|E_{\mathrm{II}}^{(0)}\right\|_{F,\lambda}^2 + 60\alpha^2\beta^2 B(\rho_\Lambda)\sum_{t=0}^{T-1}\mathbb{E}\left[\left\|(I-\Lambda)\Theta^{(t)}\right\|_{F,\lambda}^2\right]
$$
$$
+ 2n\alpha^2 B(\rho_\Lambda)\sum_{j=0}^{T-1}\mathbb{E}\left[\left\|\nabla F(\overline{\theta}_\lambda^{(t)})\right\|^2\right] + 18n\alpha^2 v^2\left(A(\rho_\Lambda) + \frac{3}{2}c_\lambda\alpha^2\beta^2 B(\rho_\Lambda)\right)T.
$$
(48)

Rearranging (48) and using that $\|(I-\Lambda)\Theta^{(t)}\|_{F,\lambda}^2 \leq \|E_{\mathrm{II}}^{(t)}\|_{F,\lambda}^2$, we can derive

$$
\left(1 - 60\alpha^2\beta^2 B(\rho_\Lambda)\right)\sum_{t=0}^{T-1}\mathbb{E}\left[\left\|E_{\mathrm{II}}^{(t)}\right\|_{F,\lambda}^2\right] \leq 6\frac{4\rho_\Lambda^4 - 5\rho_\Lambda^2 + 3}{(1-\rho_\Lambda^2)^3}\left\|E_{\mathrm{II}}^{(0)}\right\|_{F,\lambda}^2 + 2n\alpha^2 B(\rho_\Lambda)\sum_{j=0}^{T-1}\mathbb{E}\left[\left\|\nabla F(\overline{\theta}_\lambda^{(t)})\right\|^2\right]
$$
$$
+ 18n\alpha^2 v^2\left(A(\rho_\Lambda) + \frac{3}{2}c_\lambda\alpha^2\beta^2 B(\rho_\Lambda)\right)T.
$$

Thus, Eq. (5) holds.

**Proof of Strategy I.** The modification is purely mechanical: replace $\rho_\Lambda$ with $\rho_J$, multiply the summation term by $\lambda_{\max}^2$, and multiply the $E_0'$ term by $\kappa_\lambda^2$. The derivation mirrors the previous case and is omitted for brevity. The resulting expression is:

$$
\left(1 - 60\lambda_{\max}^2\alpha^2\beta^2 B(\rho_J)\right)\sum_{t=0}^{T-1}\mathbb{E}\left[\left\|E_{\mathrm{I}}^{(t)}\right\|_{F,\lambda}^2\right]
$$
$$
\leq 6\frac{4\rho_J^4 - 5\rho_J^2 + 3}{(1-\rho_J^2)^3}\kappa_\lambda^2\left\|E_{\mathrm{I}}^{(0)}\right\|_{F,\lambda}^2 + 2n\lambda_{\max}^2\alpha^2 B(\rho_J)\sum_{j=0}^{T-1}\mathbb{E}\left[\left\|\nabla F(\overline{\theta}^{(t)})\right\|_{F,\lambda}^2\right]
$$
$$
+ 18n\lambda_{\max}^2\alpha^2 v^2\left(A(\rho_J) + \frac{3}{2}c_\lambda\alpha^2\beta^2 B(\rho_J)\right)T.
$$

Thus Eq. (4) also holds. We finish the proof of Proposition 6.4.

**Proof of step size.** We show that the two upper bounds of the step size satisfy $M_1(\rho) > M_2(\rho)$ for all $0 < \rho < 1$, where

$$
M_1(\rho) = \tfrac{1}{9}\sqrt{\tfrac{3(1-\rho)}{2-\rho}}, \qquad M_2(\rho) = \tfrac{1}{2}\sqrt{\tfrac{1}{15B(\rho)}}, \quad B(\rho) = \tfrac{2(1+3\rho^4)}{(1-\rho^2)^3(1-\rho)}.
$$

Since both terms are positive, it suffices to show $[M_1(\rho)]^2 > [M_2(\rho)]^2$. Direct computation yields

$$
[M_1(\rho)]^2 = \tfrac{1-\rho}{27(2-\rho)}, \qquad [M_2(\rho)]^2 = \tfrac{(1-\rho^2)^3(1-\rho)}{120(1+3\rho^4)}.
$$

Hence

$$
[M_1(\rho)]^2 - [M_2(\rho)]^2 = (1-\rho)\left[\tfrac{1}{27(2-\rho)} - \tfrac{(1-\rho^2)^3}{120(1+3\rho^4)}\right] = \tfrac{(1-\rho)Q(\rho)}{1080(2-\rho)(1+3\rho^4)},
$$

where

$$
Q(\rho) = -9\rho^7 + 18\rho^6 + 27\rho^5 + 66\rho^4 - 27\rho^3 + 54\rho^2 + 9\rho + 22.
$$

Since the denominator is positive on $(0,1)$, it suffices to check $Q(\rho) > 0$. Noting that $\rho^7 < \rho^6$ and $\rho^3 < \rho^2$ for $0 < \rho < 1$, we have

$$
Q(\rho) > 9\rho^6 + 27\rho^5 + 66\rho^4 + 27\rho^2 + 9\rho + 22 > 0.
$$

Thus, $M_1(\rho) > M_2(\rho)$ holds for all $0 < \rho < 1$. Consequently, the step size $\alpha$ should satisfy

$$
\alpha < \frac{1}{\lambda_{\max}\beta}\min\left\{\frac{1}{9}\sqrt{\frac{3(1-\rho_J)}{2-\rho_J}}, \frac{1}{2}\sqrt{\frac{1}{15B(\rho_J)}}\right\} = \frac{1}{2\lambda_{\max}\beta}\sqrt{\frac{1}{15B(\rho_J)}},
$$

for *Strategy I*, and

$$\alpha < \frac{1}{\beta} \min \left\{ \frac{1}{9} \sqrt{\frac{3(1 - \rho_\Lambda)}{2 - \rho_\Lambda}}, \frac{1}{2} \sqrt{\frac{1}{15 B(\rho_\Lambda)}} \right\} = \frac{1}{2\beta} \sqrt{\frac{1}{15 B(\rho_\Lambda)}},$$

for *Strategy II*.

$\square$

### C.4. Obtaining the final convergence error.

In this subsection, we combine the consensus error analysis and the single-step convergence analysis and present the final convergence error and complete the proof of Theorem 6.5.

The following lemma present the convergence rate under Strategy II.

**Lemma C.17** (Convergence rate under Strategy II). *Suppose Assumptions 6.1–6.3 are all hold and the step-size* $\alpha <$ $\frac{1}{\beta} \sqrt{\frac{1}{62 B(\rho_\Lambda)}}$, *then*

$$\frac{1}{T} \sum_{t=0}^{T-1} \mathbb{E}\left[\|\nabla F(\overline{\theta}_\lambda^{(t)})\|^2\right] \le \frac{2[F(\overline{\theta}_\lambda^{(0)}) - F(\theta^\star)]}{\alpha C_2 T} + \frac{6[4\rho_\Lambda^4 - 5\rho_\Lambda^2 + 3]\beta^2}{(1 - \rho_\Lambda^2)^3 C_1 C_2 n T} \left\|E_{\mathrm{II}}^{(0)}\right\|_{F,\lambda}^2 + \frac{\alpha c_\lambda \beta v^2}{C_2}$$
$$+ \frac{18\alpha^2 \beta^2 v^2}{C_1 C_2} \left(A(\rho_\Lambda) + \frac{3}{2} c_\lambda \alpha^2 \beta^2 B(\rho_\Lambda)\right),$$

*where* $C_2 = 1 - \dfrac{2\alpha^2 \beta^2 B(\rho_\Lambda)}{C_1} > 0.$

*Proof.* By taking the summation on both sides of (21) from $t = 0$ to $T - 1$ and use (27), we can derive:

$$\frac{1}{T} \sum_{t=0}^{T-1} \mathbb{E}\left[\|\nabla F(\overline{\theta}_\lambda^{(t)})\|^2\right] \le \frac{2}{\alpha T} \sum_{t=0}^{T-1} \left(\mathbb{E}\left[F(\overline{\theta}_\lambda^{(t)})\right] - \mathbb{E}\left[F(\overline{\theta}_\lambda^{(t+1)})\right]\right) + \frac{\beta^2}{n T} \sum_{t=0}^{T-1} \mathbb{E}\left[\|(I - \Lambda)\Theta^{(t)}\|_{F,\lambda}^2\right] + \alpha c_\lambda \beta v^2$$

$$\le \frac{2[F(\overline{\theta}_\lambda^{(0)}) - F(\theta^\star)]}{\alpha T} + \frac{\beta^2}{n T} \sum_{t=0}^{T-1} \mathbb{E}\left[\|E_{\mathrm{II}}^{(t)}\|_{F,\lambda}^2\right] + \alpha c_\lambda \beta v^2$$

$$\le \frac{2[F(\overline{\theta}_\lambda^{(0)}) - F(\theta^\star)]}{\alpha T} + \frac{6[4\rho_\Lambda^4 - 5\rho_\Lambda^2 + 3]\beta^2}{(1 - \rho_\Lambda^2)^3 C_1 n T} \left\|E_{\mathrm{II}}^{(0)}\right\|_{F,\lambda}^2 + \alpha c_\lambda \beta v^2$$

$$+ \frac{2\alpha^2 \beta^2 B(\rho_\Lambda)}{C_1 T} \sum_{j=0}^{T-1} \mathbb{E}\left[\|\nabla F(\overline{\theta}_\lambda^{(t)})\|^2\right] + \frac{18\alpha^2 \beta^2 v^2}{C_1} \left[A(\rho_\Lambda) + \frac{3}{2} c_\lambda \alpha^2 \beta^2 B(\rho_\Lambda)\right].$$

Then it follows that:

$$\left(1 - \frac{2\alpha^2 \beta^2 B(\rho_\Lambda)}{C_1}\right) \frac{1}{T} \sum_{t=0}^{T-1} \mathbb{E}\left[\left\|\nabla F(\overline{\theta}_\lambda^{(t)})\right\|^2\right]$$

$$\le \frac{2[F(\overline{\theta}_\lambda^{(0)}) - F(\theta^\star)]}{\alpha T} + 18\alpha^2 \beta^2 v^2 \left(A(\rho_\Lambda) + \frac{3}{2} c_\lambda \alpha^2 \beta^2 B(\rho_\Lambda)\right) + \alpha c_\lambda \beta v^2 + \frac{6[4\rho_\Lambda^4 - 5\rho_\Lambda^2 + 3]\beta^2}{(1 - \rho_\Lambda^2)^3 C_1 n T} \left\|E_{\mathrm{II}}^{(0)}\right\|_{F,\lambda}^2,$$

which yields Eq. (7). $\square$

Similarly, we can present the following lemma to present the convergence rate under Strategy I.

**Lemma C.18** (Convergence rate under Strategy I)**.** *Suppose Assumptions 6.1–6.3 are all hold and the step-size* $\alpha <$ $\frac{1}{\lambda_{\max}\beta}\sqrt{\frac{1}{62B(\rho_J)}}$, *then*

$$\frac{1}{T}\sum_{t=0}^{T-1}\mathbb{E}[\|\nabla F(\overline{\theta}^{(t)})\|^2] \leq \frac{2[F(\overline{\theta}^{(0)}) - F(\theta^\star)]}{\alpha C_2' T} + \frac{6\kappa_\lambda^2[4\rho_J^4 - 5\rho_J^2 + 3]\beta^2}{(1-\rho_J^2)^3 C_1' C_2' nT}\|E_{\mathrm{I}}^{(0)}\|_{F,\lambda}^2 + \frac{\alpha c_\lambda \beta v^2}{C_2'}$$

$$+ \frac{18\lambda_{\max}^2\alpha^2\beta^2 v^2}{C_1' C_2'}\left(A(\rho_J) + \frac{3}{2}c_\lambda\alpha^2\beta^2 B(\rho_J)\right),$$

*where* $C_2' = 1 - \frac{2\boldsymbol{\lambda_{\max}^2}\alpha^2\beta^2 B(\rho_J)}{C_1'} > 0$.

### C.5. Proof of Corollary 6.7

*Proof.* According to (6) and (7), the convergence rate for both strategies under uniform weight can be given as:

$$\frac{1}{T}\sum_{t=0}^{T-1}\mathbb{E}[\|\nabla F(\overline{\theta}^{(t)})\|^2] \leq \frac{2[F(\overline{\theta}^{(0)}) - F(\theta^\star)]}{\alpha C_2 T} + \frac{6[4\rho_J^4 - 5\rho_J^2 + 3]\beta^2}{(1-\rho_J^2)^3 C_1 C_2 nT}\|E_{\mathrm{I}}^{(0)}\|_{F,\lambda}^2 + \frac{\alpha\beta v^2}{nC_2}$$

$$+ \frac{18\alpha^2\beta^2 v^2}{C_1 C_2}\left(A(\rho_J) + \frac{3}{2n}\alpha^2\beta^2 B(\rho_J)\right).$$

We first consider the terms $C_1$ and $C_2$, we set:

$$C_1 = 1 - 60\alpha^2\beta^2 B(\rho_J) \geq \frac{1}{2} \text{ and } C_2 = 1 - \frac{2\alpha^2\beta^2 B(\rho_J)}{C_1} \geq \frac{1}{2} \implies \alpha \leq \frac{1}{2\beta}\sqrt{\frac{1}{30B(\rho_J)}}.$$

Then, we let $\Delta = F(\overline{\theta}^{(0)}) - F(\theta^\star)$ and define

$$\alpha_1 = \left(\frac{2n\Delta}{\beta v^2 T}\right)^{\frac{1}{2}}, \quad \alpha_2 = \left(\frac{n\Delta}{18\beta v^2 A(\rho_J)T}\right)^{\frac{1}{3}}, \quad \alpha_3 = \left(\frac{n\Delta}{27\beta^4 v^2 B(\rho_J)T}\right)^{\frac{1}{5}}.$$

If we set

$$\alpha := \frac{1}{\frac{1}{\alpha_1} + \frac{1}{\alpha_2} + \frac{1}{\alpha_3} + 2\beta\sqrt{30B(\rho_J)}},$$

we can obtain the following convergence rate:

$$\frac{1}{T}\sum_{t=0}^{T-1}\mathbb{E}[\|\nabla F(\overline{\theta}^{(t)})\|^2] \leq \frac{4\Delta}{T}\left(\frac{1}{\alpha_1} + \frac{1}{\alpha_2} + \frac{1}{\alpha_3} + 2\beta\sqrt{30B(\rho_J)}\right) + \frac{2\alpha_1\beta v^2}{n} + 72\alpha_2^2\beta^2 v^2 A(\rho_J) + \frac{108\alpha_3^4\beta^4 v^2 B(\rho_J)}{n}$$

$$+ \frac{24[4\rho_J^4 - 5\rho_J^2 + 3]\beta^2}{(1-\rho_J^2)^3 nT}\|E_{\mathrm{I}}^{(0)}\|_{F,\lambda}^2$$

$$\lesssim_{n,T} \frac{1}{\sqrt{nT}}.$$

Thus, we finish the proof of this corollary. $\qquad\square$

## D. Missing Proofs in the Comparison of the Two Strategies

In this section, we present the missing proofs in Section 7, which are used for the comparison of the convergence rate of Algorithm 1 under two communication strategies.

### D.1. Proof of Theorem 7.1

Following standard practice (Lian et al., 2017; Koloskova et al., 2020), we omit the term dependent on initialization. From the two convergence results, we obtain that Strategy II achieves a faster convergence than Strategy I if the following inequalities hold:

$$\lambda_{\max}^2 \underbrace{\frac{1 + 3\rho_J^4}{(1 - \rho_J^2)^3(1 - \rho_J)}}_{B(\rho_J)} \geq \underbrace{\frac{1 + 3\rho_\Lambda^4}{(1 - \rho_\Lambda^2)^3(1 - \rho_\Lambda)}}_{B(\rho_\Lambda)}, \tag{49a}$$

$$\lambda_{\max}^2 \underbrace{\frac{1 + \rho_J^2}{(1 - \rho_J^2)^3}}_{A(\rho_J)} \geq \underbrace{\frac{1 + \rho_\Lambda^2}{(1 - \rho_\Lambda^2)^3}}_{A(\rho_\Lambda)}, \tag{49b}$$

It is clear that when $\rho_\Lambda \leq \rho_J$, the two inequalities hold trivially; thus we focus on the case $\rho_\Lambda > \rho_J$.

**Analysis for inequality** (49a). Inequality (49a) is equivalent to

$$(1 - \rho_\Lambda)^4 \geq \frac{\phi_1(\rho_\Lambda)}{\lambda_{\max}^2 \phi_1(\rho_J)}(1 - \rho_J)^4,$$

where $\phi_1(\rho) := \frac{1 + 3\rho^4}{(1 + \rho)^3}$. We now study the monotonicity of $\phi_1(\rho)$ for $\rho \in (0, 1)$. It holds that:

$$\phi_1'(\rho) = \frac{3(4\rho^3 + \rho^4 - 1)}{(1 + \rho)^4} = \frac{3H_1(\rho)}{(1 + \rho)^4}.$$

Since $3(1 + \rho)^4 > 0$ on $(0, 1)$, the sign of $\phi_1'$ equals that of $H_1$. We have

$$H_1'(\rho) = 4\rho^3 + 12\rho^2 > 0 \quad \text{for all } \rho \in (0, 1),$$

so $H_1$ is strictly increasing. Moreover, $H_1(0) = -1 < 0$ and $H_1(1) = 4 > 0$, implying a unique zero $\rho_\star \in (0, 1)$. Numerically, $\rho_\star \approx 0.605829$. Therefore, $\phi_1'(\rho) < 0$ on $(0, \rho_\star)$ and $\phi_1'(\rho) > 0$ on $(\rho_\star, 1)$.

Now we consider the bounds and extremum: $\phi_1(0) = 1$, $\phi_1(1) = \frac{1}{2} = 0.5$, and the global minimum $\phi_1(\rho_\star) \approx 0.34$.

Therefore, we can derive the supremum of the ratio:

$$\sup_{0 < \rho_J < \rho_\Lambda < 1} \frac{\phi_1(\rho_\Lambda)}{\phi_1(\rho_J)} = \frac{\lim_{\rho \to 1^-} \phi_1(\rho)}{\phi_1(\rho_\star)} \approx \frac{0.5}{0.34} \approx 1.471.$$

Define

$$\eta := \left(\sup_{0 < \rho_J < \rho_\Lambda < 1} \frac{\phi_1(\rho_\Lambda)}{\phi_1(\rho_J)}\right)^{1/4} - 1 \approx 1.471^{1/4} - 1 \approx 0.102.$$

Then any condition satisfying

$$1 - \rho_\Lambda \geq (1 + \eta)\lambda_{\max}^{-1/2}(1 - \rho_J)$$

implies

$$(1 - \rho_\Lambda)^4 \geq \frac{(1 + \eta)^4}{\lambda_{\max}^2}(1 - \rho_J)^4 \geq \frac{\phi_1(\rho_\Lambda)}{\lambda_{\max}^2 \phi_1(\rho_J)}(1 - \rho_J)^4.$$

Thus, inequality (49a) holds under the sufficient condition $(1 - \rho_\Lambda) \geq (1 + \eta)\lambda_{\max}^{-1/2}(1 - \rho_J)$.

**Analysis for inequality** (49b). Similarly, inequality (49b) is equivalent to

$$(1 - \rho_\Lambda)^3 \geq \frac{\phi_2(\rho_\Lambda)}{\lambda_{\max}^2 \phi_2(\rho_J)}(1 - \rho_J)^3,$$

where $\phi_2(\rho) = \dfrac{1+\rho^2}{(1+\rho)^3}$. Consider the differentiation of $\phi_2(\rho) = (1+\rho^2)(1+\rho)^{-3}$:

$$\phi_2'(\rho) = 2\rho(1+\rho)^{-3} - 3(1+\rho^2)(1+\rho)^{-4} = \frac{2\rho(1+\rho) - 3(1+\rho^2)}{(1+\rho)^4}.$$

The numerator simplifies to

$$2\rho(1+\rho) - 3(1+\rho^2) = 2\rho + 2\rho^2 - 3 - 3\rho^2 = -(\rho^2 - 2\rho + 3) = -((\rho-1)^2 + 2) < 0,$$

and the denominator $(1+\rho)^4 > 0$ for $\rho \in (0,1)$. Hence $\phi_2'(\rho) < 0$ on $(0,1)$, $\phi_2$ is strictly decreasing, i.e., $\frac{\phi_2(\rho_\Lambda)}{\phi_2(\rho_J)} \leq 1$. Therefore, inequality (49b) holds if

$$1 - \rho_\Lambda \geq \lambda_{\max}^{-2/3}(1 - \rho_J).$$

Finally, since $\sum_{i=1}^n \lambda_i = n$, we have $\lambda_{\max} \geq 1$, and hence

$$(1+\eta)\lambda_{\max}^{-1/2} \geq \lambda_{\max}^{-2/3}.$$

Thus, the sufficient condition for (49a) also implies the sufficient condition for (49b). Finally, combining this with the case $\rho_\Lambda \leq \rho_J$, we obtain the sufficient condition

$$1 - \rho_\Lambda \geq \min\left\{(1+\eta)\lambda_{\max}^{-1/2},\, 1\right\}(1 - \rho_J).$$

Under this condition, both inequalities (49a) and (49b) hold, and Strategy II has the faster convergence rate.

### D.2. Proof of Theorem 7.2

*Proof.* We first consider the matrix $\widetilde{W}$ obtained from $W$ via a similarity transformation, $\widetilde{W} := D_\lambda^{1/2} W D_\lambda^{-1/2}$, where $D_\lambda = \mathrm{diag}(\lambda_1, \ldots, \lambda_n)$. By construction, $\widetilde{W}$ and $W$ share the same spectrum. We can get the entries of $\widetilde{W}$ as

$$\widetilde{W}_{i,j} = \begin{cases} W_{i,i} & \text{if } i = j \\ \frac{1-\varepsilon}{d_i}\sqrt{\frac{\lambda_i}{\lambda_j}}\min\left(1, \frac{\lambda_j d_i}{\lambda_i d_j}\right) & \text{if } i \neq j \text{ and } j \in \mathcal{N}_i \\ 0 & \text{otherwise.} \end{cases}$$

We then define the Laplacian matrix as

$$\mathcal{L}(\lambda) := I - \widetilde{W},$$

whose elements can be given by:

$$\mathcal{L}(\lambda)_{i,j} = \begin{cases} -\widetilde{W}_{i,j} & \text{if } i \neq j, \\ 1 - \widetilde{W}_{i,i} = \sum_{k \neq i}\widetilde{W}_{i,k} & \text{if } i = j. \end{cases}$$

It follows that the spectrum of $\mathcal{L}$ satisfies

$$\sigma(\mathcal{L}(\lambda)) = 1 - \sigma(\widetilde{W}) = 1 - \sigma(W) \geq 0.$$

Moreover, we note that $|\sigma_{\min}(W)| \leq |\sigma_2(W)|$, and thus the second largest eigenvalue in absolute value of $W$ corresponds to the second smallest eigenvalue of $\mathcal{L}(\lambda)$, i.e., $1 - \sigma_2(W) = \sigma_{n-1}(\mathcal{L}(\lambda))$. According to the Courant–Fischer theorem, the variational characterization of $\widetilde{W}$ can be expressed in terms of the Rayleigh quotient (Mohar, 1991; Chung, 1997). In particular, the second smallest eigenvalue corresponds to the minimum Rayleigh quotient over the subspace orthogonal to the trivial eigenvector, which is $D_\lambda^{\frac{1}{2}}$:

$$\mathcal{L}(\lambda)D_\lambda^{1/2}\mathbf{1} = \left(I - D_\lambda^{1/2}W D_\lambda^{-1/2}\right)D_\lambda^{1/2}\mathbf{1} = D_\lambda^{1/2}\mathbf{1} - D_\lambda^{1/2}W\mathbf{1} = 0.$$

For the weighted case, this yields:

$$\sigma_{n-1}(\mathcal{L}(\lambda)) = \inf_{\substack{z \neq 0 \\ \langle z, D_\lambda^{1/2} \mathbf{1} \rangle = 0}} \frac{z^\top \mathcal{L}(\lambda) z}{\|z\|_2^2}, \quad \forall z \in \mathbb{R}^n, \tag{50}$$

whereas in the uniform-weight case ($\lambda \equiv \mathbf{1}$) we have

$$\sigma_{n-1}(\mathcal{L}(\mathbf{1})) = \inf_{\substack{z' \neq 0 \\ \langle z', \mathbf{1} \rangle = 0}} \frac{z'^\top \mathcal{L}(\mathbf{1}) z'}{\|z'\|_2^2}, \qquad z' \in \mathbb{R}^n. \tag{51}$$

The quadratic form $z^\top \mathcal{L}(\lambda) z$ can be expanded as follows:

$$\begin{aligned}
z^\top \mathcal{L}(\lambda) z &= \sum_{i,j} z_i \mathcal{L}(\lambda)_{i,j} z_j = \sum_i z_i^2 \mathcal{L}(\lambda)_{i,i} + \sum_{i \neq j} z_i \mathcal{L}(\lambda)_{i,j} z_j \\
&= \sum_i z_i^2 \sum_{j \neq i} \widetilde{W}_{i,j} - \sum_{i \neq j} z_i \widetilde{W}_{i,j} z_j = \sum_{i \neq j} \widetilde{W}_{i,j} z_i^2 - \sum_{i \neq j} \widetilde{W}_{i,j} z_i z_j.
\end{aligned}$$

By interchanging the summation indices, we obtain

$$\sum_{i \neq j} \widetilde{W}_{i,j} z_i^2 = \sum_{i \neq j} \widetilde{W}_{j,i} z_j^2,$$

which leads to

$$\begin{aligned}
z^\top \mathcal{L}(\lambda) z &= \frac{1}{2} \sum_i \sum_{j \neq i} \sum_{i \neq j} (\widetilde{W}_{i,j} z_i^2 + \widetilde{W}_{j,i} z_j^2) - \sum_{i \neq j} \widetilde{W}_{i,j} z_i z_j = \frac{1}{2} \sum_{i \neq j} \widetilde{W}_{i,j} (z_i^2 + z_j^2) - \sum_{i \neq j} \widetilde{W}_{i,j} z_i z_j \\
&= \frac{1}{2} \sum_{i \neq j} \widetilde{W}_{i,j} (z_i^2 + z_j^2 - 2 z_i z_j) = \frac{1}{2} \sum_{i \neq j} \widetilde{W}_{i,j} (z_i - z_j)^2,
\end{aligned} \tag{52}$$

where the symmetry property of $\widetilde{W}$ is utilized, i.e., $\widetilde{W}_{j,i} = \widetilde{W}_{i,j}$. For simplicity ,we denote $R = \max\left\{ (1+\eta)\kappa_\lambda^{-1/3}, \ \lambda_{\max}^{-1/2} \right\}$.

According to (50) and (51), we have

$$\inf_{\substack{z \neq 0 \\ \langle z, D_\lambda^{1/2} \mathbf{1} \rangle = 0}} \frac{z^\top \mathcal{L}(\lambda) z}{\|z\|_2^2} \geq R \inf_{\substack{z' \neq 0 \\ \langle z', \mathbf{1} \rangle = 0}} \frac{z'^\top \mathcal{L}(\mathbf{1}) z'}{\|z'\|_2^2} \iff \sigma_{n-1}(\mathcal{L}(\lambda)) \geq R \cdot \sigma_{n-1}(\mathcal{L}(\mathbf{1})).$$

By substituting the explicit form of the quadratic terms and noting that $1 - \rho_\Lambda = \sigma_{n-1}(\mathcal{L}(\lambda))$ and $1 - \rho_J = \sigma_{n-1}(\mathcal{L}(\mathbf{1}))$, we obtain the necessary and sufficient condition of $\rho_\Lambda \leq \rho_J$:

$$\inf_{\substack{z \neq 0 \\ \langle z, D_\lambda^{1/2} \mathbf{1} \rangle = 0}} \frac{\sum_{i,j} \min\left\{ \frac{1}{d_i} \sqrt{\frac{\lambda_i}{\lambda_j}}, \frac{1}{d_j} \sqrt{\frac{\lambda_j}{\lambda_i}} \right\} (z_i - z_j)^2}{\|z\|_2^2} \geq R \inf_{\substack{z' \neq 0 \\ \langle z', \mathbf{1} \rangle}} \frac{\sum_{i,j} \min\left\{ \frac{1}{d_i}, \frac{1}{d_j} \right\} (z_i' - z_j')^2}{\|z'\|_2^2},$$

which yields the final result. $\qquad \square$

### D.3. Proof of Corollary 7.3

*Proof.* We consider the Loewner order (Horn & Johnson, 2012) between the two Laplacian matrices $\mathcal{L}(\lambda)$ and $R\mathcal{L}(\mathbf{1})$

$$\mathcal{L}(\lambda) \succeq R\mathcal{L}(\mathbf{1}) \tag{53}$$

holds, which means that the matrix difference $\mathcal{L}(\lambda) - R\mathcal{L}(\mathbf{1})$ is positive semi-definite, then by standard eigenvalue monotonicity under the Loewner order (Fan, 1949) we have

$$\sigma_i\big(\mathcal{L}(\lambda)\big) \geq R\sigma_i\big(\mathcal{L}(\mathbf{1})\big), \qquad \forall i = 1, \dots, n.$$

In particular,

$$1 - \rho_\Lambda = \sigma_{n-1}\big(\mathcal{L}(\lambda)\big) \geq R\sigma_{n-1}\big(\mathcal{L}(\mathbf{1})\big) = R(1 - \rho_J),$$

which implies $\rho_\Lambda \leq \rho_J$ and yields exactly the scaled spectral-gap relation required in Theorem 7.1.

It remains to translate the Loewner-order relation (53) into conditions on the node degrees and weights. By definition of the semidefinite order,

$$\mathcal{L}(\lambda) \succeq R \cdot \mathcal{L}(\mathbf{1}) \iff z^\top \left[\mathcal{L}(\lambda) - R \cdot \mathcal{L}(\mathbf{1})\right] z \geq 0 \quad \forall z \in \mathbb{R}^n.$$

Substituting the quadratic-form representation (52) into the above inequality yields a sufficient condition that can be checked elementwise:

$$\min\left\{\frac{1}{d_i}\sqrt{\frac{\lambda_i}{\lambda_j}}, \frac{1}{d_j}\sqrt{\frac{\lambda_j}{\lambda_i}}\right\} \geq R\min\left\{\frac{1}{d_i}, \frac{1}{d_j}\right\}, \qquad \forall i, j. \tag{54}$$

Let $r := \sqrt{\lambda_i/\lambda_j}$ and define

$$I(r) := \min\left\{\frac{r}{d_i}, \frac{1}{rd_j}\right\}.$$

Then (54) is equivalent to

$$I(r) \geq R\min\left\{\frac{1}{d_i}, \frac{1}{d_j}\right\}.$$

We now analyze the range of $r$ for which this inequality holds.

**Case 1:** $d_i \geq d_j$**.** In this case, $\min\{1/d_i, 1/d_j\} = 1/d_i$, and $I(r)$ is piecewise:

$$I(r) = \begin{cases} \frac{r}{d_i}, & r \leq \frac{d_i}{d_j}, \\ \frac{1}{rd_j}, & r \geq \frac{d_i}{d_j}. \end{cases}$$

The inequality $I(r) \geq R/d_i$ holds if and only if

$$\frac{r}{d_i} \geq \frac{R}{d_i} \quad \text{and} \quad \frac{1}{rd_j} \geq \frac{R}{d_i},$$

which simplifies to

$$R \leq r \leq \frac{d_i}{Rd_j}.$$

**Case 2:** $d_i < d_j$**.** Now $\min\{1/d_i, 1/d_j\} = 1/d_j$, and a symmetric argument shows that $I(r) \geq R/d_j$ holds if and only if

$$\frac{Rd_i}{d_j} \leq r \leq \frac{1}{R}.$$

Combining the two cases yields the unified condition

$$R\min\left\{\frac{d_i}{d_j}, 1\right\} \leq r \leq R^{-1}\max\left\{\frac{d_i}{d_j}, 1\right\} \implies R\min\left\{\frac{d_i}{d_j}, 1\right\} \leq \sqrt{\frac{\lambda_i}{\lambda_j}} \leq R^{-1}\max\left\{\frac{d_i}{d_j}, 1\right\},$$

which is exactly (9). This proves the sufficient condition for $1 - \rho_\Lambda \geq R(1 - \rho_J)$.

Finally, we show that the spectral gap of $W$ is maximized when $\lambda$ is proportional to the degree vector $d$. This corresponds to maximizing the edge weights, or equivalently maximizing $I(r)$ with respect to $r > 0$. Since

$$I(r) = \min \left\{ \frac{r}{d_i}, \frac{1}{rd_j} \right\},$$

the two branches intersect when

$$\frac{r}{d_i} = \frac{1}{rd_j} \iff r^2 = \frac{d_i}{d_j} \iff r^\star = \sqrt{\frac{d_i}{d_j}}.$$

At this point,

$$I(r^\star) = \frac{1}{\sqrt{d_i d_j}},$$

which is the maximum possible value of $I(r)$ over $r > 0$. In terms of $\lambda$, this corresponds exactly to

$$\frac{\lambda_i}{\lambda_j} = (r^\star)^2 = \frac{d_i}{d_j},$$

i.e., $\lambda$ is proportional to $d$. This proves the "moreover" part of the corollary. $\square$

## E. Details of Algorithm 2

This appendix provides the pseudocode of the four subroutines used in Algorithm 2. SCALETODEGREES rescales the weight vector into an integer degree sequence with an even total sum. HAVELHAKIMICONSTRUCT attempts to realize this degree sequence as a simple graph via a Havel–Hakimi procedure. MAKECONNECTED then applies degree-preserving edge swaps to connect different components while keeping all node degrees fixed. Finally, FALLBACKCONNECTEDGRAPH gives a simple ring-based construction used when the previous steps fail to produce a connected realization.

---

**Algorithm 3** SCALETODEGREES

---

**Require:** Weight vector $\lambda = (\lambda_1, \ldots, \lambda_n)$, target average degree $\bar{d}$
**Ensure:** Candidate integer degree sequence $d_1, \ldots, d_n$
1: $S \leftarrow$ nearest even integer to $n\bar{d}$
2: $c \leftarrow S / \sum_{i=1}^{n} \lambda_i$
3: **for** $i = 1$ to $n$ **do**
4:     $d_i \leftarrow \text{round}(c\lambda_i)$
5:     $d_i \leftarrow \min\{n - 1, \max\{1, d_i\}\}$ {clip to $[1, n-1]$}
6: **end for**
7: **if** $\sum_{i=1}^{n} d_i$ is odd **then**
8:     Choose an index $k$ with $1 \leq d_k < n - 1$
9:     $d_k \leftarrow d_k + 1$ {make the total degree sum even}
10: **end if**
11: **return** $(d_1, \ldots, d_n)$

---

---

**Algorithm 4** HAVELHAKIMICONSTRUCT

---

**Require:** Degree sequence $d_1, \ldots, d_n$ with even total sum
**Ensure:** Simple graph $\mathcal{G}_\lambda = (\mathcal{V}, \mathcal{E}_\lambda)$ realizing $d$ or failure
1: $\mathcal{V} \leftarrow \{1, \ldots, n\}, \mathcal{E}_\lambda \leftarrow \emptyset$
2: rem $\leftarrow \{(d_i, i) : i = 1, \ldots, n\}$ {residual degrees}
3: **while** some vertex has positive residual degree **do**
4:     Shuffle rem randomly
5:     Sort rem in non-increasing order by residual degree
6:     $(r, u) \leftarrow$ first element of rem; remove it from rem
7:     **if** $r > |$rem$|$ **then**
8:         **fail** {degree sequence is not graphical}
9:     **end if**
10:    Let $(v_1, \ldots, v_r)$ be the indices of the $r$ vertices in rem with largest residual degrees such that $(u, v_j) \notin \mathcal{E}_\lambda$ for all $j$
11:    **if** fewer than $r$ such vertices exist **then**
12:       **fail** {cannot add edges without violating simplicity}
13:    **end if**
14:    **for** $j = 1$ to $r$ **do**
15:       Add edge $(u, v_j)$ to $\mathcal{E}_\lambda$
16:       Decrease the residual degree of $v_j$ in rem by 1
17:       **if** some residual degree becomes negative **then**
18:         **fail**
19:       **end if**
20:    **end for**
21: **end while**
22: **return** $\mathcal{G}_\lambda = (\mathcal{V}, \mathcal{E}_\lambda)$

---

**Algorithm 5** MAKECONNECTED

---

**Require:** Simple graph $\mathcal{G}_\lambda = (\mathcal{V}, \mathcal{E}_\lambda)$ realizing the target degrees
**Ensure:** Connected simple graph $\mathcal{G}'_\lambda = (\mathcal{V}, \mathcal{E}'_\lambda)$ with the same degrees
1: $\mathcal{G}'_\lambda \leftarrow \mathcal{G}_\lambda$
2: **for** iter $= 1, 2, \ldots, K$ **do**
3:    Compute the connected components of $\mathcal{G}'_\lambda$
4:    **if** there is only one component **then**
5:       **return** $\mathcal{G}'_\lambda$
6:    **end if**
7:    Select two distinct components $C_1$ and $C_2$
8:    Select an internal edge $(a, b) \in \mathcal{E}'_\lambda$ with $a, b \in C_1$ {e.g., uniformly at random}
9:    Select an internal edge $(c, d) \in \mathcal{E}'_\lambda$ with $c, d \in C_2$ {e.g., uniformly at random}
10:   success $\leftarrow$ **false**
11:   **for** each choice of pairing, i.e., $(e_1, e_2) \in \{\{(a, c), (b, d)\}, \{(a, d), (b, c)\}\}$ **do**
12:     **if** neither $e_1$ nor $e_2$ already belongs to $\mathcal{E}'_\lambda$ and they are not self-loops **then**
13:       $\mathcal{E}'_\lambda \leftarrow \mathcal{E}'_\lambda \setminus \{(a, b), (c, d)\}$
14:       $\mathcal{E}'_\lambda \leftarrow \mathcal{E}'_\lambda \cup \{e_1, e_2\}$
15:       success $\leftarrow$ **true**
16:       **break** {degrees are preserved by this edge swap}
17:     **end if**
18:    **end for**
19:    **if not** success **then**
20:       {no valid pairing for this choice of edges; try different edges in the next iteration}
21:    **end if**
22: **end for**
23: **return** $\mathcal{G}'_\lambda$ {may still be disconnected in the worst case}

---

---

**Algorithm 6** FALLBACKCONNECTEDGRAPH

---

**Require:** Target degree sequence $d_1, \ldots, d_n$
**Ensure:** Connected simple graph $\mathcal{G}_\lambda = (\mathcal{V}, \mathcal{E}_\lambda)$ approximating $d$
 1: $\mathcal{V} \leftarrow \{1, \ldots, n\}, \mathcal{E}_\lambda \leftarrow \emptyset$
 2: Initialize $\mathcal{E}_\lambda$ as a simple ring:
 3: **for** $i = 1$ to $n - 1$ **do**
 4:     add edge $(i, i + 1)$
 5: **end for**
 6: add edge $(n, 1)$
 7: **for** $i = 1$ to $n$ **do**
 8:     $\text{need}_i \leftarrow \max\{0, d_i - \deg_{\mathcal{G}_\lambda}(i)\}$ {remaining degree to be assigned}
 9: **end for**
10: **for** $t = 1$ to $n(n-1)/2$ **do**
11:     Choose $u$ with maximal $\text{need}_u$
12:     **if** $\text{need}_u \leq 0$ **then**
13:        **break**
14:     **end if**
15:     Choose $v \neq u$ with maximal $\text{need}_v$, $\text{need}_v > 0$, and $(u, v) \notin \mathcal{E}_\lambda$
16:     **if** such $v$ exists **then**
17:        Add edge $(u, v)$ to $\mathcal{E}_\lambda$
18:        $\text{need}_u \leftarrow \text{need}_u - 1, \text{need}_v \leftarrow \text{need}_v - 1$
19:     **else**
20:        **break**
21:     **end if**
22: **end for**
23: **return** $\mathcal{G}_\lambda = (\mathcal{V}, \mathcal{E}_\lambda)$

---

# F. Experimental Details and Additional Results

In this section, we provide the details on the experiment as well as additional experimental results.

### F.1. Topology

In the experiments, we use three types of topologies that commonly used in existing works including *ring graphs*, *static exponential graphs*, and *grid graphs*. Moreover, we also introduce *Erdős–Rényi random graph* and *random geometric graph* to the experiments, as well as the customized topology constructed according to the weight $\lambda$ described in Section 7.2. Each of them represents a distinct level of sparsity and connectivity structure:

1. **Ring graph.** Each node $i$ is connected only to its two immediate neighbors $(i - 1)$ and $(i + 1)$ with cyclic wrapping, forming a one-dimensional circular structure. This topology is regular and sparse, and is often used to study information propagation under limited local communication. Formally, the adjacency satisfies $\mathcal{E}_{\text{ring}} = \{(i, (i \pm 1) \mod n)\}$.

2. **Static exponential graph.** (Ying et al., 2021) Each node $i$ is connected to nodes whose distances are powers of two, i.e., $(i \pm 2^p) \mod n$ for integer $p \geq 0$ up to $\lfloor \log_2(n/2) \rfloor$. This design introduces logarithmic shortcut links while maintaining deterministic sparsity, improving the spectral gap compared with the ring topology.

3. **Grid graph.** Nodes are arranged on a $\sqrt{n} \times \sqrt{n}$ two-dimensional lattice, where each node connects to its four orthogonal neighbors (up, down, left, right) with periodic boundary conditions. This structure is widely used to model spatially local communication and resembles decentralized sensor or mesh networks.

4. **Erdős–Rényi random graph.** (Beveridge & Youngblood, 2016) A stochastic topology where each undirected edge between any pair of nodes is independently included with probability $p \in (0, 1)$. The resulting graph $\mathcal{G}_{\text{ER}}(n, p)$ is connected with high probability when $p > \frac{\log n}{n}$, and exhibits an expected node degree of $(n - 1)p$. This topology captures random and dynamic communication patterns often observed in large-scale or unreliable networks.

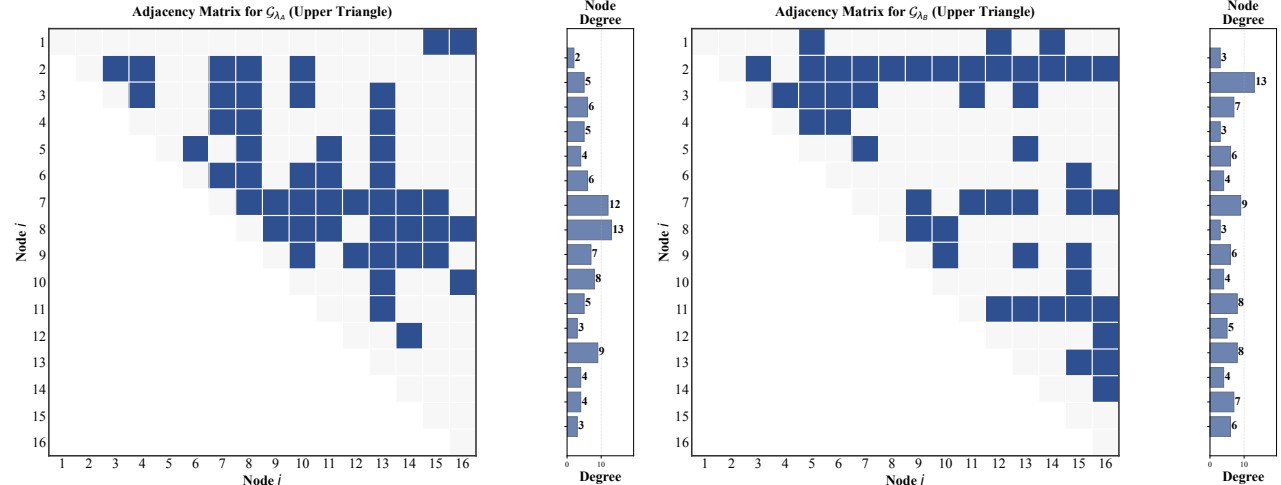

*Figure 4.* Adjacency matrix of $\mathcal{G}_{\lambda_A}$. If node $i$ is connected with node $j$, the $(i, j)$ block is blue.

*Figure 5.* Adjacency matrix of $\mathcal{G}_{\lambda_B}$. If node $i$ is connected with node $j$, the $(i, j)$ block is blue.

5. **Random geometric graph.** (Boyd et al., 2005) Nodes are uniformly sampled from the unit square $[0, 1]^2$, and an undirected edge is established between any two nodes whose Euclidean distance is less than a given radius $r = 0.3$. Formally, the adjacency satisfies $\mathcal{E}_{\text{RGG}} = \{(i, j) \mid \|x_i - x_j\|_2 \leq r\}$. This topology captures spatial locality in communication and is commonly used to model wireless or sensor networks, where connectivity depends on physical proximity rather than explicit node indices. When $r = \Theta\left(\sqrt{\frac{\log n}{n}}\right)$, the graph is connected with high probability.

6. **Custom graph $\mathcal{G}_\lambda = (\mathcal{V}, \mathcal{E}_\lambda)$.** (Section 7.2)

We use prescribed heterogeneous weight vectors to control the level of weight imbalance: $\lambda_A$ and $\lambda_B$ for the 16-node experiments, $\lambda_C$ for the 32-node experiments, and $\lambda_D$ for the 64-node experiments. All weight vectors are generated once and kept fixed across all random seeds and topology comparisons. The weight ratios increase from 7.3 to 20 and 50 in the larger-scale settings.

$$\lambda_A = [0.3, 0.8, 1.0, 0.9, 0.7, 1.0, 2, 2.2, 1.2, 1.4, 0.8, 0.5, 1.5, 0.6, 0.6, 0.5]^\top$$

The corresponding communication topology $\mathcal{G}_{\lambda_A} = (\mathcal{V}, \mathcal{E}_{\lambda_A})$ generated based on these weights is shown in the Figure 4.

The prescribed weight vector for the second 16-node setting is:

$$\lambda_B = [0.4, 2.2, 1.2, 0.5, 1.0, 0.6, 1.5, 0.5, 1.0, 0.7, 1.3, 0.9, 1.4, 0.6, 1.2, 1.0]^\top$$

The corresponding topology $\mathcal{G}_{\lambda_B} = (\mathcal{V}, \mathcal{E}_{\lambda_B})$ is shown in the Figure 5.

The prescribed weight vector for the 32-node setting is:

$$\begin{aligned}
\lambda_C = [&1.7, 2.5, 0.5, 0.4, 1.1, 0.4, 0.9, 0.6, 0.5, 0.6, 0.7, 2.6, 0.2, 1.7, 2.0, 0.8, \\
&0.5, 0.4, 0.3, 1.4, 0.2, 1.5, 0.7, 0.4, 0.4, 0.2, 4.0, 1.7, 0.9, 0.4, 1.5, 0.3]^\top
\end{aligned}$$

The prescribed weight vector for the 64-node setting is:

$$\begin{aligned}
\lambda_D = [&0.5, 1.2, 0.4, 2.5, 0.1, 0.7, 1.5, 0.6, 3.0, 0.2, 0.4, 0.8, 0.5, 1.1, 0.3, 2.0, \\
&0.7, 0.5, 0.6, 5.0, 0.2, 0.4, 0.9, 1.2, 0.1, 0.7, 0.4, 1.4, 0.6, 0.5, 2.5, 0.3, \\
&0.8, 0.2, 3.5, 0.6, 0.5, 0.7, 1.0, 0.4, 0.2, 1.8, 0.5, 0.6, 0.3, 0.4, 1.1, 2.2, \\
&0.1, 0.7, 0.5, 1.5, 0.4, 2.8, 0.2, 0.6, 1.2, 0.4, 2.0, 0.5, 0.7, 1.0, 0.3, 4.5]^\top
\end{aligned}$$

*Table 1.* Spectral gaps of different network topologies under weights $\lambda_A$ and $\lambda_B$.

| Topology | Mixing matrices | | |
|:---:|:---:|:---:|:---:|
| | $W(\lambda_A)$ | $W^{\text{ds}}$ | $W(\lambda_B)$ |
| Ring | 0.034 | **0.053** | 0.027 |
| Grid | 0.075 | **0.119** | 0.086 |
| Exp | 0.248 | **0.400** | 0.202 |
| $\mathcal{G}_{\lambda_A}$ | **0.311** | 0.108 | / |
| $\mathcal{G}_{\lambda_B}$ | / | 0.130 | **0.293** |

The boldface numbers highlight the largest spectral gaps within each row.

For readability, we do not include the full adjacency matrices for the $32-$node and $64-$node tailored graphs. Instead, we report the node degrees, which compactly summarize the resulting topologies and reflect the degree–weight structure produced by Algorithm 2.

The node degrees of $\mathcal{G}_{\lambda_C}$ are:

$$[17, 25, 5, 4, 11, 4, 9, 6, 5, 6, 7, 26, 3, 17, 20, 8, 5, 4, 3, 14, 2, 15, 7, 4, 4, 2, 31, 17, 9, 4, 15, 3].$$

The node degrees of $\mathcal{G}_{\lambda_D}$ are:

$$[5, 12, 4, 25, 1, 7, 15, 6, 30, 2, 4, 8, 5, 11, 3, 20, 7, 5, 6, 50, 2, 4, 9, 12, 1, 7, 4, 14, 6, 5, 25, 3,$$
$$8, 2, 35, 6, 5, 7, 10, 4, 2, 18, 5, 6, 3, 4, 11, 22, 1, 7, 5, 15, 4, 28, 2, 6, 12, 4, 20, 5, 7, 10, 3, 45].$$

### F.2. Comparison on Spectral Gaps

Existing studies typically adopt a conservative choice of the inertia coefficient (e.g., $\varepsilon = 0.5$) (Alghunaim & Yuan, 2022; Yuan et al., 2023). In contrast, we set $\varepsilon = 0.3$, which achieves a more balanced trade-off between stability and convergence speed (Levin & Peres, 2017).

The results in Table 1 validate the observation discussed in Section 7: for regular graphs, uniform weights yield the largest spectral gap. Moreover, the matrices constructed on the tailored graphs achieve larger spectral gaps in heterogeneous settings.

### F.3. Synthetic Quadratic Experiment

We consider a decentralized least-squares loss with heterogeneous local optima and optional gradient noise (Koloskova et al., 2020). For each node $i \in \{1, \ldots, n\}$, the local loss is

$$F_i(\theta) = \frac{1}{2}\|A_i\theta - b_i\|_2^2 + \frac{\rho}{2}\|\theta\|_2^2, \qquad A_i \in \mathbb{R}^{d \times d},\, b_i \in \mathbb{R}^d.$$

In our construction we take $A_i = \zeta_i^{\frac{1}{2}} I_d$ and write $b_i = A_i c_i$, where $c_i$ denotes the unregularized local center.

#### F.3.1. DATA GENERATION

We synthesize problem instances as follows.

1. **Curvature (no directional heterogeneity).** For each node, sample a scalar curvature $\zeta_i \sim \mathcal{U}[\zeta_{\min}, \zeta_{\max}]$ and set $A_i = \zeta_i^{\frac{1}{2}} I_d$. In our default setting, $\zeta_{\min} = 5.5$ and $\zeta_{\max} = 12.5$.

2. **Local offsets (heterogeneous optima).** Draw a shared reference center $c_{\text{base}} \sim \mathcal{N}(0, I_d)$. For each node $i$, assign $c_i = c_{\text{base}} + \mu_i$, where $\mu_i = \mu_0 v_i$, $v_i \sim \mathcal{N}(0, I_d)/\|v_i\|_2$.

3. **Initialization (heterogeneous).** Each node starts from an independent random vector $\theta_i^{(0)} \sim \mathcal{N}(0, I_d)$.

4. **Global weighted optimum (closed form).** For an explicit loss-weight vector $\lambda$, the weighted global minimizer is

$$\theta^\star = \Big( \sum_{i=1}^{n} \lambda_i \zeta_i I_d + \rho I_d \Big)^{-1} \Big( \sum_{i=1}^{n} \lambda_i \zeta_i c_i \Big).$$

### F.3.2. GRADIENT EVALUATION

At iteration $t$, node $i$ computes a local stochastic gradient by adding noise:

$$g_i^{(t)} = \nabla F_i\left(\theta_i^{(t)}\right) + \sigma \xi_{i,t} = \zeta_i\big(\theta_i^{(t)} - c_i\big) + \rho\theta_i^{(t)} + \sigma\xi_{i,t}, \quad \xi_{i,t} \sim \mathcal{N}(0, I_d).$$

To ensure reproducibility while keeping node- and iteration-wise diversity, $\xi_{i,t}$ is drawn using a deterministic composite seed $s_{i,t} = s_0 + 1000i + 10t$ for a base seed $s_0$. This makes the sequence $\{\xi_{i,t}\}$ independent across nodes/iterations.

### F.3.3. IMPLEMENTATION DETAILS

Unless otherwise stated, we use:

- Number of nodes: $n = 16, 32, 64$, dimension: $d = 10$

- Iterations: $T = 300$, evaluation interval: $3$

- Regularization: $\rho = 0.01$, gradient-noise std: $\sigma = 1.0$

- Curvature range: $\zeta_i \in [5.5, 12.5]$, local offsets: $\mu_0 = 3.0$

### F.3.4. EVALUATION METRICS

We monitor the global gradient norm defined as:

$$\left\| \frac{\lambda^\top}{n} \nabla F(\Theta^{(t)}) \right\| = \left\| \frac{1}{n} \sum_{i=1}^{n} \lambda_i \nabla F_i(\theta_i^{(t)}) \right\|,$$

and the distance to the optimum $\theta^\star$: $\|\bar{\theta}^{(t)} - \theta^\star\|$ for Strategy I and $\|\bar{\theta}_\lambda^{(t)} - \theta^\star\|$ for Strategy II.

### F.4. Extra Experiment Results

Each experiment is repeated 10 times over independent random seeds and the results are averaged. We additionally report the distance between the corresponding network average and the closed-form optimum $\theta^\star$ under different weight settings. These results verify that both strategies approach the same optimum, while Strategy II typically reaches a smaller steady-state error.

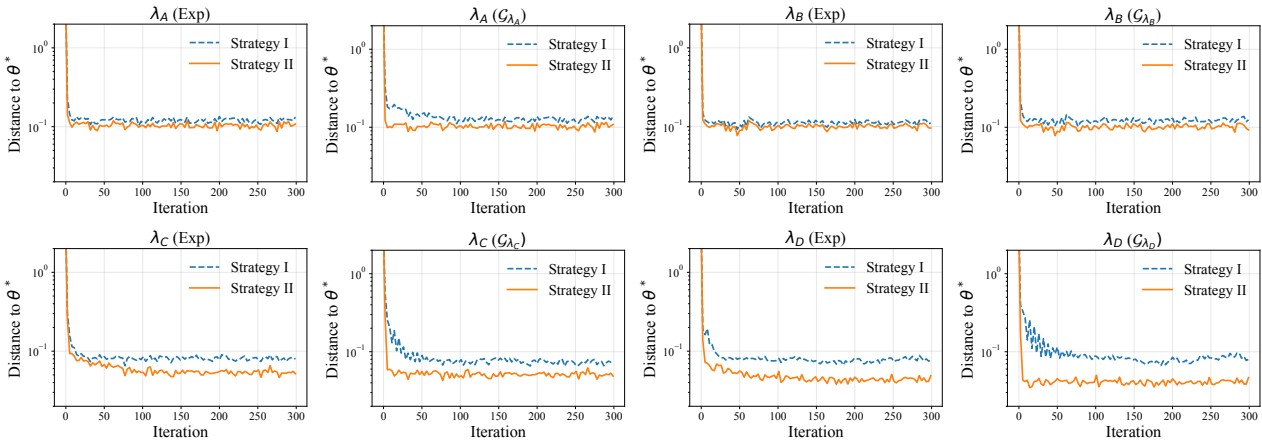

*Figure 6.* Distance between the corresponding network average and the closed-form optimum $\theta^\star$ in the least-squares experiments. The 16-node experiments are conducted under $\lambda_A$ and $\lambda_B$, while the 32-node and 64-node experiments use $\lambda_C$ and $\lambda_D$, respectively.

## F.5. CIFAR-10 Classification Experiment

We evaluate the proposed methods on the CIFAR-10 (Krizhevsky et al., 2009) dataset using a ResNet-18 (He et al., 2016) architecture with Batch Normalization (ResNet18WithBN). The network adopts the standard four-stage structure with channel widths $\{64, 128, 256, 512\}$, ReLU activations, and global average pooling. Following common practice (He et al., 2016; Zhang et al., 2019), we apply a weight decay of $5 \times 10^{-4}$ to convolutional and linear weights, while excluding BatchNorm parameters and bias terms from regularization.

The dataset is randomly partitioned across $n$ nodes according to a weighting vector $\lambda = [\lambda_1, \ldots, \lambda_n]^\top$, so that each node receives a fraction $\lambda_i/n$ of the total samples. All models are trained using vanilla stochastic gradient descent with a mini-batch size of 128. Every 30 iterations, all nodes are jointly evaluated on the full test set. The reported metrics include both accuracy and loss, where the loss (*interval loss*) is computed as the average over each 30-iteration interval, and both accuracy and loss are aggregated across nodes via $\lambda$-weighted averaging.

The detailed experimental results are as follows.

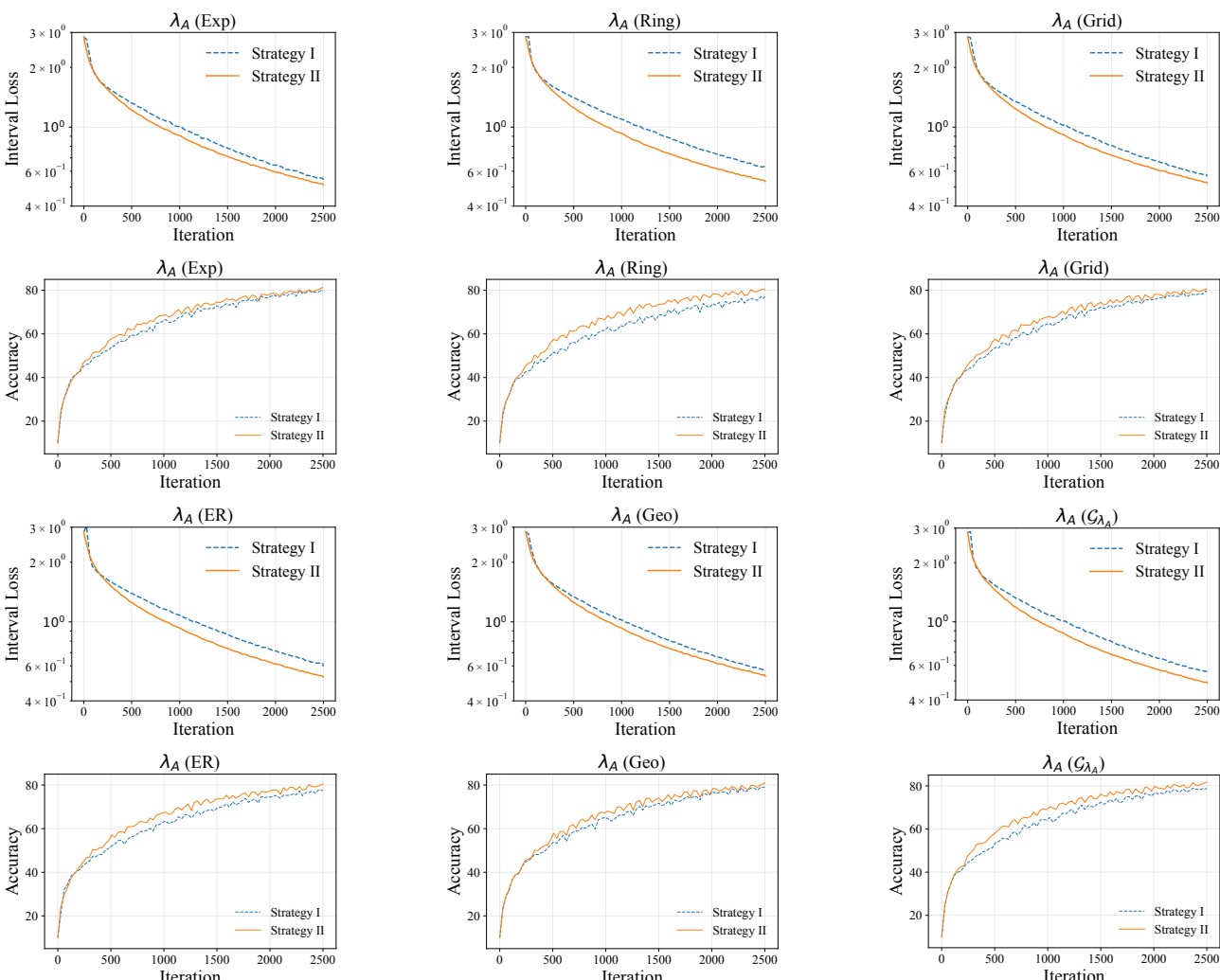

*Figure 7.* Training loss and test accuracy for models trained by Strategy I and II on CIFAR-10 dataset under weight $\lambda_A$ (16 nodes). Strategy II outperforms on all topologies.

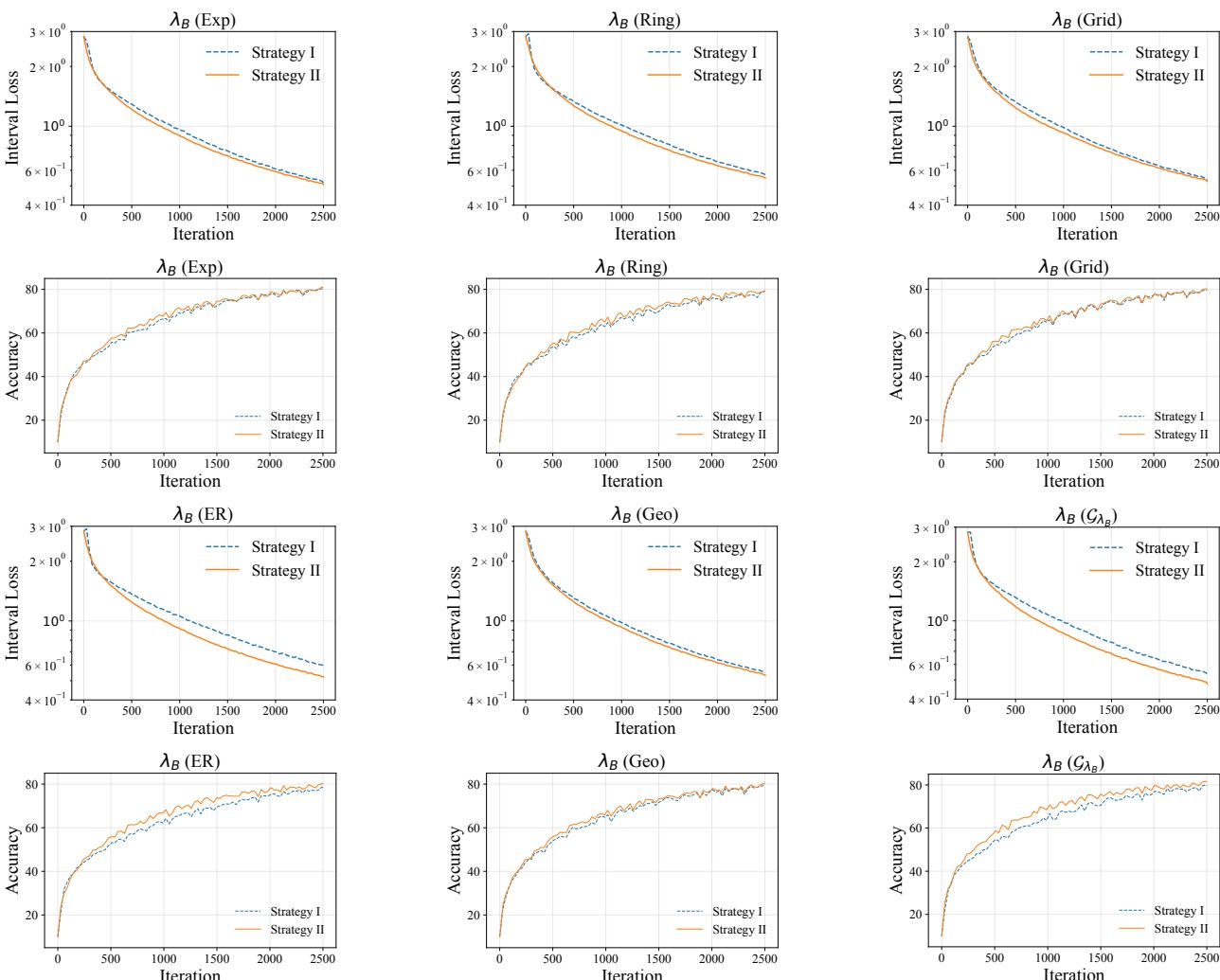

*Figure 8.* Training loss and test accuracy for models trained by Strategy I and II on CIFAR-10 dataset under weight $\lambda_B$ (16 nodes). Strategy II outperforms on all topologies.

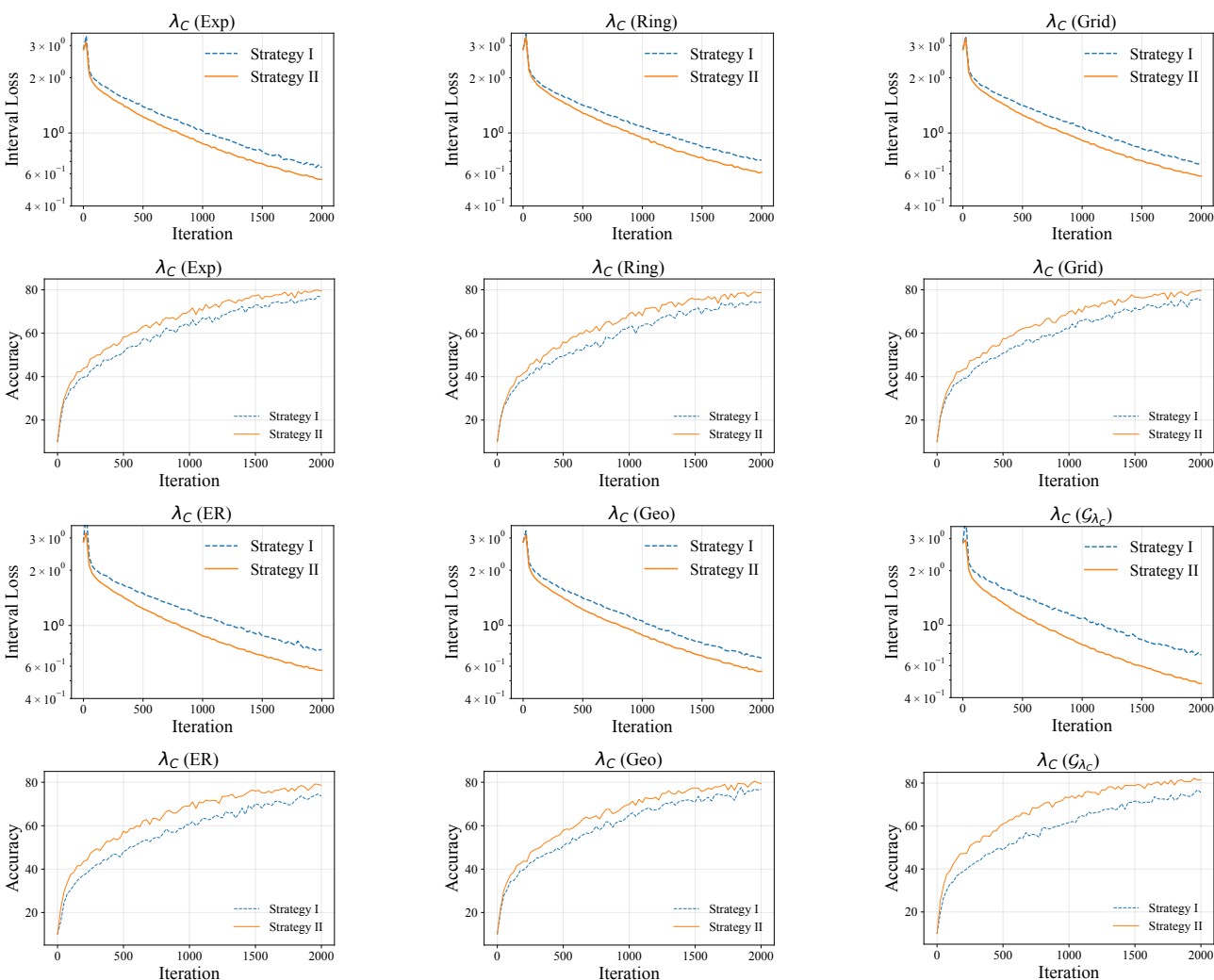

*Figure 9.* Training loss and test accuracy for models trained by Strategy I and II on the CIFAR-10 dataset under weight $\lambda_C$ (32 nodes). Strategy II outperforms on all topologies.

