# OpenReview forum: "Row-Stochastic Matrices Can Provably Outperform Doubly Stochastic Matrices in Decentralized Learning"
_ICML.cc/2026/Conference — ICML 2026 regular_

### Official Review · Reviewer_aBHn · 2026-02-15

**Soundness:** 2
**Presentation:** 3
**Significance:** 1
**Originality:** 2
**Overall Recommendation:** 2
**Confidence:** 5

**Summary:**

This paper discusses decentralized optimization and learning algorithms under so-called weights. The authors discuss two injection mechanisms for such weights and obtain different convergence rates.

**Compliance With Llm Reviewing Policy:**

Affirmed.

**Final Justification:**

The optimal solution changes when different row-stochastic weight vectors are used, which can be readily established through a straightforward fixed-point analysis. Consequently, employing such a weight modifies the original loss function rather than preserving its intended form. As a result, the proposed approach amounts to a mathematical artifact rather than a physically meaningful formulation.

**Key Questions For Authors:**

see weakness

**Limitations:**

My main concern is the rationale of the design and the applicability of the results. See weakness

**Strengths And Weaknesses:**

Strengths:

1. Decentralized optimization and leaning is an interesting problem.

2. The presentation seems clear.

Weakness:

My main concern is the motivation and rationale of adding the so-called "weights", which is where the core contribution lies:

1. In the motivation, it is mentioned that “In practical systems, nodes often possess unequal data volumes and computational capacities, making heterogeneous weighting a more practical choice.” This is not clear to the reviewer. In fact, in most existing decentralized optimization and learning algorithms, the mixing matrix is completely independent of the local objective functions. Therefore, this statement seems questionable.

2. In the problem formulation (1), weights lambda are incorporated before the local objective functions. The motivation for this new parameter is questionable. In fact, this weight will completely change the local objective functions and hence alter the optimization problem in the general case where the local objective functions are different.

3. The two studied strategies seem to lead to different dynamics and, consequently, different optimal solutions. Therefore, it is unclear what the point is of studying these two strategies, which do not even yield the same solution.

4. In practice, the weights that ensure accurate convergence to the desired optimum are dependent on the global topology. Hence, it is difficult to see how such weights can be designed effectively in a fully decentralized manner where no agents have the global topology information (Algorithm 2 does seem to be an efficient algorithm for effectively building a desired topology, and no comparisons are given with respect to the state of the art).

---

> ### Author Rebuttal · Authors · 2026-03-31
>
> We are grateful for the reviewer's insightful feedback and constructive comments. We address each point in detail below. **All newly added experiments can be found in the Rebuttal Experiment Sheet** <https://anonymous.4open.science/r/ICML_Figures-0FBD>.
>
> **Q1: Motivation for heterogeneous weighting and weight-dependent matrices.**
>
> **A1:** Thanks for the comment. We first explain the motivation for using heterogeneous weights:
>
> 1.  In distributed learning, the global loss is naturally a weighted sum of local losses based on data volumes / distributions [1-2]. Recent work also uses heterogeneous weighting to reflect node influence [3].
>
> 2.  Some decentralized algorithms intentionally introduce node-specific weights to handle heterogeneity [4-5].
>
> 3.  Even without explicit modeling, decentralization can implicitly induce effective weights via topology, step sizes, or stationary distribution of the mixing [6-7].
>
> We then discuss the connection between the mixing matrix and local loss:
>
> Most existing work assumes uniform weights, so the matrices are weight-independent [8-9]. When heterogeneous weighting is considered, incorporating weights into the mixing is an established approach in several prior work [5-7]. The core question is whether to absorb weights into the mixing under heterogeneity. We compare two strategies that optimize the same weighted global loss:
>
> - Strategy I absorbs $\lambda_i$ into local losses with a weight-independent doubly stochastic matrix.
>
> - Strategy II keeps local losses unscaled and absorbs $\lambda_i$ into a row-stochastic matrix.
>
> We show that within our standard single-loop gradient-tracking framework, the doubly stochastic matrix is non-self-adjoint in the weighted Hilbert space, introducing multiplicative penalties in the convergence rates. Strategy II restores self-adjointness and removes these penalties.
>
> We will revise the introduction to avoid ambiguity.
>
> **Q2: Motivation and impact of $\lambda$-weighted objective.**
>
> **A2:** Thanks for the comment. We do not change the global loss, but consider weighted loss following [1-5] with two strategies to solve it: Strategy I scales local losses, while Strategy II employs a $\lambda$-induced mixing matrix. We provide the details in the next response.
>
> **Q3: Concern on different dynamics and optimal solutions.**
>
> **A3:** Thanks for the comment. The two strategies are not different optimization problems; they are two decentralized realizations of the same weighted objective in Eq. (1). Local trajectories and aggregate states ($\bar{\theta}^{(t)}$ and $\bar{\theta}^{(t)}_\lambda$) generally differ, so we do not claim identical iterate-by-iterate dynamics.
>
> What is shared is the $\lambda$-weighted aggregation structure in the mean recursion. Each method updates its own aggregate by subtracting a $\lambda$-weighted average of local stochastic gradients generated along its own trajectory (detailed in Section 4). Thus both methods target the same weighted global objective and optimal solutions of Eq. (1), differing only in consensus error propagation, leading to different transient behavior and asymptotic error neighborhoods.
>
> In the least-squares experiment, Eq. (1) has a unique optimum $\theta^\*$. Our added plots confirm that both strategies converge to neighborhoods of $\theta^\*$, with Strategy II reaching a visibly tighter neighborhood, consistent with the extra penalty terms in our rates.
>
> **Q4: Decentralized design of weights and Algorithm 2.**
>
> **A4:** Thanks for the comment. We clarify that $\lambda$ is a prescribed weight vector defining the target objective, not a topology-dependent tuning variable, following [1-5]. Nodes do not choose $\lambda_i$ arbitrarily after seeing the graph; $\lambda$ is part of the problem specification.
>
> Given $\lambda$ and the graph, constructing the row-stochastic matrix $W$ is fully decentralized and offline. Each node only needs its own and neighbors' degrees/weights (Eq. (3)).
>
> Algorithm 2 plays a different role: it is a heuristic to approximately satisfy the sufficient condition in Section 7.1 for obtaining a larger spectral gap. Our main result in Theorem 6.5 does not rely on Algorithm 2; the advantage of Strategy II comes from self-adjointness in the weighted space, while topology only enters via the spectral terms. We will refine the introduction to avoid confusion.
>
> **References**
>
> [1] Communication-efficient learning of deep networks from decentralized data
>
> [2] Variance-reduced stochastic learning by networked agents under
> random reshuffling
>
> [3] DICE: Data influence cascade in decentralized learning
>
> [4] Distributed pareto optimization via diffusion strategies
>
> [5] Exact diffusion for distributed optimization and learning---part i: Algorithm development
>
> [6] Differentially private decentralized learning with random walks
>
> [7] Adaptive network
>
> [8] Optimal gradient tracking for decentralized optimization
>
> [9] A unified theory of decentralized sgd with changing topology and local updates

---

> > ### Author Rebuttal · Reviewer_aBHn · 2026-04-01
> >
> > Thank you for your response. I am still having concerns that using different weight strategies changes the optimal solution, and hence using row-stochastic weights just makes the corresponding optimal value of global objective function smaller. Can you prove that the two strategies lead to exactly the same optimal solution under deterministic F_i in (1)? Otherwise, the comparison is not physically meaningful.

---

> > > ### Author Response · Authors · 2026-04-03
> > >
> > > Thanks for the follow-up. Let $F(\theta)=\frac{1}{n}\sum_{i=1}^n\lambda_i F_i(\theta)$, which is exactly the fixed global objective in Eq. (1). The optimal value depends on the objective itself, not on the strategy. The two strategies are two decentralized ways to realize the same objective. **Importantly, the convergence proofs provided throughout our paper are entirely based on this global objective $F(\theta)$.** We now give a brief proof.
> > >
> > > For clarity, let $\bar{\theta}\_{\rm{I}}^{(t)}$ and $\bar{\theta}\_{\rm{II}}^{(t)}$ denote the simple mean and the $\lambda$-weighted mean of the node parameters under Strategies I and II, respectively. Accordingly, we denote their corresponding matrices as $W_{\rm{I}}=W^{\rm{ds}}$ and $W_{\rm{II}}=W$, with left-invariant vectors $a_{\rm{I}}^\top=\mathbf{1}^\top / n$ and $a_{\rm{II}}^\top=\lambda^\top / n$, respectively.
> > >
> > > As detailed in Section 4, under deterministic gradients and the GT initialization, the aggregate recursions are
> > >
> > > $$\bar{\theta}\_s^{(t+1)}=a_s^\top W\_s[\Theta_s^{(t)}-\alpha Y_s^{(t)}]=\bar{\theta}\_s^{(t)}-\frac{\alpha}{n}\sum\_{i=1}^n\lambda_i \nabla F_i(\theta_{s,i}^{(t)}),\quad s\in\\{\mathrm{I},\mathrm{II}\\}.$$
> > >
> > > The local gradient weights of each node completely correspond to $F$ under both strategies, which is the core. Next, we prove that both strategies can achieve consensus asymptotically.
> > >
> > > With deterministic gradients ($v=0$), Theorem 6.5 establishes that for both strategies:
> > >
> > > $$\frac{1}{T}\sum_{t=0}^{T-1} \\|\nabla F(\overline{\theta}_{s}^{(t)})\\|^2=\mathcal{O}\left(\frac{1}{T}\right),\quad s\in\\{\rm I,\rm{II}\\}$$
> > >
> > > This implies the infinite series of squared gradients is strictly bounded:$\sum_{t=0}^{\infty} \\|\nabla F(\overline{\theta}_{s}^{(t)})\\|^2<\infty.$
> > >
> > > Substituting $v=0$ and the result above into Proposition 6.4, the cumulative consensus error is also bounded: $\sum_{t=0}^{\infty}\\|E_{s}^{(t)}\\|_{F,\lambda}^2<\infty.$ Since all terms are non-negative, the necessary condition for series convergence gives:
> > >
> > > $$\lim\_{t \to \infty}\left\\|E\_{s}^{(t)}\right\\|\_{F, \lambda}^2=0 \implies \lim_{t \to \infty} \\|\theta_{s,i}^{(t)}-\bar{\theta}_s^{(t)}\\|=0,\quad i\in\\{1,\cdots,n\\},$$
> > >
> > > implying that both strategies would reach consensus asymptotically. By the smoothness of the objectives, the local gradients approach the gradient at the mean: $\lim_{t \to \infty} \\|\nabla F_i(\theta_{s,i}^{(t)}) - \nabla F_i(\bar{\theta}_s^{(t)})\\|=0$. Consequently, the aggregate recursion can be written as:
> > >
> > > $$\bar{\theta}\_{s}^{(t+1)}
> > > = \bar{\theta}\_s^{(t)}-\frac{\alpha}{n}\sum_{i=1}^n\lambda_i \nabla F\_i(\bar{\theta}_s^{(t)}) +e_s^{(t)},$$
> > >
> > > where $e_s^{(t)}:=\frac{\alpha}{n}\sum\_{i=1}^n\lambda_i [\nabla F\_i(\bar{\theta}\_s^{(t)})- \nabla F_i(\theta_{s,i}^{(t)})] $ vanishes as $t\to\infty$. Thus, asymptotically, both strategies reduce to the centralized gradient descent on the **same global objective**. Since Theorem 6.5 gives that the gradient norm vanishes asymptotically, any consensus limit point $\theta^\dagger_s$ of either strategy $s$ must satisfy:
> > >
> > > $$\frac{1}{n}\sum_{i=1}^n\lambda_i \nabla F_i(\theta^\dagger_s)=\nabla F(\theta^\dagger_s)=\mathbf{0},\quad s\in\\{\mathrm I,\mathrm{II}\\}.$$
> > >
> > > Thus, any consensus limit point of either strategy is a stationary point of **the same objective $F$**. Because both strategies run over the exact same landscape of $F$, they share the same set of stationary points. In the specific case where $F$ is strongly convex with a unique optimum $\theta^\*$, the condition $\nabla F(\theta^\dagger_s) = \mathbf{0}$ uniquely identifies $\theta^\*$. **It strictly follows that the sequences for both strategies converge to this single optimum $\theta^\*$.** We will clarify this equivalence explicitly in the revised manuscript.
> > >
> > > Existing literature [1-3] has also established the equivalence of the two strategies: under the same step sizes, the stationary distribution $\pi$ of the mixing matrix determines the global objective $\min_\theta \sum_{i=1}^n \pi_i F_i(\theta)$. Thus, matching $\pi$ to the weights ($\pi_i = \lambda_i / n$, Strategy II) is equivalent to using a uniform distribution ($\pi_i = 1/n$) with scaled local losses $\lambda_i F_i(\theta)$ (Strategy I). Our contribution is to show their different pre-convergence dynamics. We prove that Strategy II is self-adjoint in $L^2(\lambda;\mathbb{R}^d)$, removing the multiplicative penalties that slow Strategy I.
> > >
> > > We also ran the deterministic least-squares experiment and updated the Rebuttal Experiment Sheet <https://anonymous.4open.science/r/ICML_Figures-0FBD>. Both strategies converge to the same closed-form $\theta^*=\Big(\sum_{i=1}^n \lambda_i \zeta_i I_d + \rho I_d \Big)^{-1}\Big(\sum_{i=1}^n \lambda_i \zeta_i c_i \Big)$, while Strategy II reaches it faster.
> > >
> > >
> > > **References**
> > >
> > > [1] Differentially private decentralized learning with random walks
> > >
> > > [2] Adaptive network
> > >
> > > [3] Information exchange and learning dynamics over weakly-connected adaptive networks

---

### Official Review · Reviewer_Bouw · 2026-03-06

**Soundness:** 4
**Presentation:** 4
**Significance:** 3
**Originality:** 3
**Overall Recommendation:** 4
**Confidence:** 4

**Summary:**

This paper studies decentralized optimization, with weighted local functions.
The main results of the paper are a general analysis using the local weights as a targeted stationary probability distribution of the weighted communication graph. Comparing the results obtained by 1) reweighting local objective functions to have uniform stationary distribution and 2) not reweighting, the authors show that 2) yields faster convergence than 1).
Theory is corroborated with empirical illustrations.

**Compliance With Llm Reviewing Policy:**

Affirmed.

**Final Justification:**

I thank the authors for their detailed answers to my questions, which answered the tightness and terminology concerns.

**Key Questions For Authors:**

- Related to the main weakness: in order to prove that row stochastic matrices are better than doubly-stochasitic ones, it would be necessary to show that a) the given analysis is tight and b) the given algorithms are optimal.
Indeed, if a) is not true, then maybe the given bounds do not reflect the actual behavior. I believe that a) can in essence be tight and captures the right quantities. However, b) is not tight (acceleration possible, etc), meaning that maybe for the proposed algorithm row-stochastic matrices are better. But this is restricted to this algorithm, tweaking doubly stochastic algorithms in another way could yield better results maybe.

- *Heterogeneous node weights induce a non-Euclidean geometry* or other related sentences: the proposed norm **is Euclidean**. With a diagonal matrix, the Mahalanobis norm is simply a Euclidean norm after rescaling each coordinate, so it is generally not described as non-Euclidean. (small terminology issue).

- References: missing references on SGD with Markovian data for decentralizd optimization. For this type of algorithm, does the observed phenomenon in the current submission still hold?

**Limitations:**

Yes

**Strengths And Weaknesses:**

Strengths:

- Setting: decentralized optimization with local weights. A very common objective indeed, and thus improving current methods is of real interest. Would be nice to motivate further weights (sample size, but maybe other ones are used sometimes? Personalization?)

- Proof that in the given setting, for the given algorithm and the given analysis, one can do better with weighted metric than without


Main weakness:

- The optimality of the analysis or the algorithm are not shown. This is a weakness, if the goal of the paper is to prove than 1) is better than 2)

---

> ### Author Rebuttal · Authors · 2026-03-31
>
> We are grateful for the reviewer's insightful feedback and constructive comments. We address each point in detail below.
>
> **Q1: Broader motivation for heterogeneous weights.**
>
> **A1:** Thanks for the insightful suggestion. Beyond sample sizes, heterogeneous weights also enable *influence-aware* or *priority-aware* decentralized learning [1]. Unlike fully personalized methods that train distinct local models, our approach learns a single shared model with varying emphasis on specific nodes. We will revise the introduction to broaden our motivation.
>
> **Q2: On the analytical tightness and algorithmic optimality.**
>
> **A2:** Thanks for the insightful comment. Our result is a mechanism-level comparison within the fixed single-loop gradient tracking (GT) template, rather than a universal minimax statement:
>
> 1.  **Decisive gap in the consensus step:** Within our standard single-loop GT framework, the performance gap is an algebraic consequence of the consensus step, not an analytical artifact. Since both strategies share the same global loss and gradient aggregation (Section 4), their difference reduces entirely to the mixing matrices during the consensus step. Specifically, because Strategy I's doubly-stochastic matrix is non-self-adjoint under the $\lambda-$weighted space, it incurs unavoidable $\kappa_\lambda$-/ $\lambda_{\max}$- type penalties. Thus, the comparative gap is decisive within the current framework. We provide a pure consensus illustration in our response to DAW7 to demonstrate the penalty directly.
>
> 2.  **On the tightness and optimality:** In decentralized optimization, tightness or lower bounds are often conditional: They usually depend on the problem class, extra communication rounds, particular topology, acceleration, and so on [2-4]. Therefore, one could potentially tweak Strategy I by using these factors. But adding these factors alters the algorithm itself and brings extra overhead, which is beyond our scope. Instead, our goal is to focus on a specific question within the single-loop GT framework: Given a fixed weighted global loss, how should we handle the weights to achieve faster convergence?
>
> 3.  **Rigorousness of the analytical framework:** Furthermore, the convergence rates we provide are also sufficiently rigorous:
>
>     - Our framework builds upon and improves the standard paradigm for decentralized learning [5]. By deriving closed-form weighted spectral norms, we eliminate the need for the window-averaging technique, leading to a more precise tracking of the transient dynamics.
>
>     - As shown in Corollary 6.6, our analysis successfully recovers the standard convergence rates for decentralized SGD in the uniform-weighting case.
>
>     - We prove in Section 6.4 that the convergence rates derived under our weighted Hilbert space framework are strictly tighter than those in standard Euclidean space analysis for Strategy I.
>
> We will revise the wording in the manuscript and add a discussion regarding the tightness, lower bounds, optimality, and potential tweaks to Strategy I.
>
> **Q3: Terminology clarification regarding non-Euclidean geometry.**
>
> **A3:** Thanks for pointing out the small terminology issue. The $\lambda$-induced norm is indeed a weighted Euclidean (diagonal Mahalanobis) norm. We will replace "non-Euclidean geometry\" with "weighted Euclidean geometry\" throughout the revised manuscript.
>
> **Q4: Extension to Markovian data and missing references.**
>
> **A4:** Thanks for pointing out this missing line of work. Our current theory does not directly cover this setting, as Assumption 6.2 relies on unbiased stochastic gradients rather than the biased, temporally dependent gradients of Markovian sampling [6-8].
>
> However, the comparative gap is driven by the mixing matrix, not the data model. Strategy I incurs the extra penalties because doubly-stochastic mixing is non-self-adjoint. Existing Markovian SGD analyses generally add a data-mixing penalty alongside explicit topology terms. Therefore, we suspect the gap between the two strategies persists under Markovian sampling, though a formal proof with Markovian bias is beyond the current scope. We will add the missing references and clarify this scope in the revision.
>
> **References**
>
> [1] DICE: Data influence cascade in decentralized learning
>
> [2] Optimal gradient tracking for decentralized optimization
>
> [3] Revisiting optimal convergence rate for smooth and non-convex stochastic decentralized optimization
>
> [4] Optimal algorithms for smooth and strongly convex distributed optimization in networks
>
> [5] An improved analysis of gradient tracking for decentralized machine learning
>
> [6] On the decentralized stochastic gradient descent with markov chain sampling
>
> [7] A randomized incremental subgradient method for distributed optimization in networked systems
>
> [8] On markov chain gradient descent

---

> > ### Author Rebuttal · Reviewer_Bouw · 2026-04-01
> >
> > I thank the authors for their detailed answers to my questions, which answered the tightness and terminology concerns.

---

> > > ### Author Response · Authors · 2026-04-01
> > >
> > > We thank the reviewer for the positive feedback and for acknowledging that we have addressed the concerns on tightness and terminology. The constructive comments have greatly improved the quality of our paper.

---

### Official Review · Reviewer_DAW7 · 2026-03-10

**Soundness:** 3
**Presentation:** 4
**Significance:** 3
**Originality:** 3
**Overall Recommendation:** 5
**Confidence:** 4

**Summary:**

This paper focuses on decentralied optimization where a group of agents work collectively to minimize weighted sum of local objective functions. Normally, heterogeneous node weights are considered part of loss function. This paper instead focuses on moving weights into edge weights of mixing matrix, and shows that moving heterogeneous node weights into edge weights of mixing matrix can provably accelerate convergence. Technically, they mainly utilize a weighted Hilbert space in their analysis which is natural and straightforward. Their analysis can be potentially applied to more settings in decentralized optimization. Generally, this paper is well-written with solid theoretical support and numerical results.

**Compliance With Llm Reviewing Policy:**

Affirmed.

**Final Justification:**

I will keep the rating, my main concerns have been addressed.

**Key Questions For Authors:**

1. Can your results be extended to directed graphs?
2. Is it possible to show tightness of bounds for doubly stochastic cases? Comparison between loose bounds is not meaningful actually.
3. Why does gradient norm of Strategy I and II converge to different value in Figure 1? Loss functions of least square quadratic experiments is convex, they should converge to the neighborhood of the same optimal point.
4. Is performance gap still preserved with more nodes in your numerical experiments? As the number of nodes increase, the effect of moving weights into graphs should be more significant intuitively because the ratio between largest and smallest weights tend to be larger for networks with more nodes.
5. Weights can also be viewed as part of functions and change the smoothness constant of the functions. If the smoothness constants are different for each $f_i$, is it provable that moving weights into graphs still outperform moving them into smoothness constants of functions?

**Limitations:**

Yes

**Strengths And Weaknesses:**

Strength:
1. It establishes solid theoretical foundation for convergence analysis of decentralized optimization with row-stochastic mixing matrices. This problem is well-motivated and meaningful in practice, especially considering that data distribution of different clients normally vary a lot.
2. Presentation is clear. The proofs are technically sound and easy to follow. Their proof mainly rely on defining a weighted Hilbert norm which has the potential to be extended to more general settings in decentralized optimization.
3. They consider multiple standard graph topology and show consistent performance improvement. This makes the experiment results more convincing.

Weakness:
1. Results are only shown for undirected graphs.
2. No tightness for bound on doubly stochastic cases is shown. Comparison between two loose bounds is not decisive.
3. Experiments are limited to toy settings where there are only 16 nodes.
4. The analysis starts directly from gradient tracking algorithm. Although gradient tracking is one of the most popular decentralized optimization algorithms, it would be clearer if it starts from analysis of consensus rates and then moves to complexities of some specific algorithm on loss functions.

---

> ### Author Rebuttal · Authors · 2026-03-31
>
> We are grateful for the reviewer's insightful feedback and constructive comments. We address each point in detail below. **All newly added experiments can be found in the Rebuttal Experiment Sheet** <https://anonymous.4open.science/r/ICML_Figures-0FBD>.
>
> **Q1: Extension to directed graphs.**
>
> **A1:** Thanks for the comment. Extending to directed graphs is non-trivial because of the lack of bidirectional communication and the self-adjointness.  Please refer to a detailed discussion in response A2 to Reviewer g7Yd.
>
> **Q2: On the tightness of the convergence rate.**
>
> **A2:** Thanks for the comment. Within the standard single-loop GT framework, our comparison is meaningful because the penalties are rooted in the weighted-space consensus operator. Since multiple reviewers raised this question, please see our detailed discussion in response A3 to Reviewer Bouw.
>
> **Q3: Convergence to different norms in Fig. 1.**
>
> **A3:** Thanks for the comment. Both strategies target exactly the same optimum $\theta^\*=\Big(\sum\_{i=1}^n\lambda\_i\zeta\_iI_d+\rho I\_d\Big)^{-1}\Big(\sum_{i=1}^n\lambda_i\zeta_i c_i\Big)$ and converge to a neighborhood of it.
>
> Theorem 6.5 shows that Strategy I incurs multiplicative asymptotic penalties, forcing it into a larger neighborhood of the optimum under constant step sizes within our single-loop GT template. Strategy II removes the penalties to yield a tighter neighborhood and lower gradient norm. Our added plots track the distance of the (weighted) mean to $\theta^\*$, confirming that both strategies converge toward the same optimum, but Strategy II achieves a smaller steady-state error.
>
> **Q4: Performance gap under larger networks.**
>
> **A4:** Thanks for the comment. To test the suggested regime, we scaled our 16-node baseline (weight ratio of 7.3) to 32 and 64 nodes, yielding larger weight ratios of 20 and 50, respectively.
>
> Least-squares: We scaled the network to 32 and 64 nodes. The new figures confirm that the convergence gap between strategies grows significantly as the weight ratio scales.
>
> CIFAR-10: We added a 32-node CIFAR-10 experiment. Consistent with the synthetic results, the gap in both loss and test accuracy is wider than our 16-node baseline.
>
> **Q5: Starting with simple consensus analysis.**
>
> **A5:** Thanks for the comment. We give a simple consensus problem in the $\lambda-$weighted space below and will add it to the revision.
>
> Let $\Lambda := \frac{\bf{1}\lambda^\top}{n}$ and $J := \frac{\bf{1}\bf{1}^\top}{n}$. We first analyze the pure consensus of Strategy II ($\Theta^{(t+1)} = W \Theta^{(t)}$). The consensus error is evaluated as:
>
> $\left\\|(I-\Lambda)\\Theta^{(t)}\right\\|\_{F,\lambda}^2=\left\\|(I-\Lambda)W^t\\Theta^{(0)}\right\\|\_{F,\lambda}^2=\left\\|(W-\Lambda)^t\\Theta^{(0)}\right\\|\_{F,\lambda}^2\le\\rho\_{\Lambda}^{2t}\left\\|\\Theta^{(0)}\right\\|\_{F,\lambda}^2.$
>
> Similarly, for Strategy I ($\Theta^{(t+1)} = W^\mathrm{ds} \Theta^{(t)}$), the error is bounded by:
>
> $\left\\|(I-J)\\Theta^{(t)}\right\\|\_{F,\lambda}^2=\left\\|(W^\mathrm{ds}-J)^t\\Theta^{(0)}\right\\|\_{F,\lambda}^2\le\\kappa\_{\lambda}^2\\rho\_{J}^{2t}\left\\|\\Theta^{(0)}\right\\|\_{F,\lambda}^2.$
>
> Because $W^{\rm{ds}}$ is non-self-adjoint in the $\lambda$-weighted space, the penalty $\kappa_\lambda^2$ is unavoidable. The extra penalty $\lambda_{\max}^2$ in Theorem 6.5 arises from the GT dynamics, not the pure consensus step.
>
> **Q6: On the heterogeneous smoothness constants.**
>
> **A6:** Thanks for the comment. Indeed, writing Strategy I with scaled local losses yields effective smoothness constants $\tilde\beta_i=\lambda_i\beta_i$, introducing additional curvature heterogeneity.
>
> However, this is a broader question than the one we isolate. Rather than proposing a fully $\beta_i$-aware design, our analysis specifically isolates where to place the weight vector $\lambda$ when comparing two strategies for the same loss under a standard shared smoothness bound [1-2].
>
> A fully $\beta_i$-aware analysis is more subtle, requiring extra mechanisms like adaptive step sizes or smoothness-based mixing, mostly limited to convex settings [3-4]. Standard non-convex analyses typically revert to a common smoothness bound [5]. Thus we do not claim dominance across all $\beta_i$-aware variants. Our focus is narrower: within the standard single-loop GT template, absorbing $\lambda$ into the matrix brings self-adjointness, while absorbing it into the losses incurs extra penalties. Empirically, Strategy II consistently remains faster in our quadratic experiments with heterogeneous curvatures $\zeta_i$. We will clarify this scope in the revision.
>
> **References**
>
> [1] Optimal gradient tracking for decentralized optimization
>
> [2] Optimal complexity in decentralized training
>
> [3] AdGT: Decentralized gradient tracking with tuning-free per-agent stepsize
>
> [4] Random walk gradient descent for decentralized learning on graphs
>
> [5] Decentralized stochastic gradient tracking for non-convex empirical risk minimization

---

> > ### Author Rebuttal · Reviewer_DAW7 · 2026-04-01
> >
> > My main concerns have been addressed. The additional experimental results are satisfactory and discussions on extending to directed graphs and tightness of bounds make sense

---

> > > ### Author Response · Authors · 2026-04-02
> > >
> > > We thank the reviewer for the positive feedback and for acknowledging that our additional experiments and discussions on directed graphs and theoretical tightness have fully addressed the initial concerns. These constructive comments have significantly improved the quality of our paper.

---

### Official Review · Reviewer_g7Yd · 2026-03-11

**Soundness:** 3
**Presentation:** 4
**Significance:** 3
**Originality:** 4
**Overall Recommendation:** 4
**Confidence:** 3

**Summary:**

This paper studies decentralised optimisation with heterogeneous node weights. It compares two strategies. Strategy I absorbs the weights into the local objective functions and works with doubly stochastic mixing matrices. Strategy II retains the original local objectives and instead employs row-stochastic mixing.

The authors argue that standard Euclidean analysis is too coarse to meaningfully distinguish between these two approaches. By introducing a weighted Hilbert space, they show that row-stochastic mixing matrices are preferable, as they remain self-adjoint in this space. In contrast, doubly stochastic matrices lose self-adjointness, which leads to an additional multiplicative penalty term that amplifies the consensus error bound.

**Compliance With Llm Reviewing Policy:**

Affirmed.

**Key Questions For Authors:**

1. If node weights $\lambda$ are not known, how does the convergence rate degrade if the weights $\lambda$ used to build the matrix $A$ differ from the actual loss weights?
2. How is the computational cost of finding a row-stochastic matrix $A$ satisfying $A^\top \lambda = \lambda$?
3. Can the authors provide a lower bound showing that no doubly stochastic matrix can match the rate of Strategy II?

**Limitations:**

The paper ignores the communication overhead of weight-sharing required to build the row-stochastic matrix.

**Strengths And Weaknesses:**

(1a) Soundness - strengths: The technical claims are supported by a rigorous new theoretical framework (weighted Hilbert space). By moving away from Euclidean analysis, the authors provide tighter and more representative convergence rates for heterogeneous settings.

(1b) Soundness - weaknesses: In reality, node weights (often based on dataset size) may be unknown or change as data is streamed. If there is a mismatch between the matrix stationary distribution and the actual $\lambda$, the bias reduction property collapses.

(2a) Presentation - strengths: The paper is well-structured, clearly distinguishing between the two strategies and providing intuitive explanations for why the row-stochastic approach succeeds. In particular, the penalty term explanation was helpful.

(3a) Significance - strengths: Addressing data heterogeneity is a critical real-world problem in decentralized learning. Providing a provable way to improve convergence simply by changing the mixing matrix design is highly practical.

(3b) Significance - weaknesses: The real-world practicality might depend on the ability to construct specific row-stochastic matrices which might be harder than constructing doubly stochastic ones in directed graphs.

(4a) Originality - strengths: The paper makes significant move from standard literature that typically assumes uniform weighting or doubly stochastic mixing. Identifying the self-adjoint property of row-stochastic matrices in weighted spaces is a novel and insightful observation.

---

> ### Author Rebuttal · Authors · 2026-03-31
>
> We are grateful for the reviewer's insightful feedback and constructive comments. Below, we address each point in detail.
>
> **Q1: Convergence degradation under mismatched or dynamic node weights.**
>
> **A1:** Thanks for the insightful comment. In highly dynamic streaming settings, true weights can change. We clarify how this affects our theoretical guarantees and the comparison between the two strategies.
>
> 1.  Like most work in decentralized optimization [1-3], we assume a fixed $\lambda$ in the weighted global loss. Our goal is to compare the convergence rates of the two strategies within the standard single-loop gradient-tracking (GT) framework. Handling time-varying weights would shift the problem to dynamic tracking or online learning, which is outside our scope.
>
> 2.  If the weights used in the algorithm ($\hat{\lambda}$) differ from the intended weights ($\lambda_{\mathrm{true}}$), the dynamics do not break down, and both strategies simply target a surrogate global loss $F\_{\hat{\lambda}} = \frac{1}{n}\sum\_i \hat{\lambda}\_i F\_i(\theta)$. The gradient mismatch is $\nabla F\_{\hat\lambda} - \nabla F\_{\lambda\_{\mathrm{true}}} = \frac1n\sum\_i (\hat\lambda\_i-\lambda\_{\mathrm{true},i})\nabla F\_i$, which appears as an additive bias floor.
>
>     - The rate to optimize $F_{\hat{\lambda}}$ does not degrade. It is strictly governed by the spectral properties of the matrix built with $\hat{\lambda}$ (as quantified by our Theorem 6.5), entirely independent of $\lambda_{\mathrm{true}}$.
>
>     - Both strategies target the same surrogate global loss $F_{\hat{\lambda}}$, and the theorem applies with respect to $F_{\hat{\lambda}}$; The gap to the intended loss $F_{\lambda_{\mathrm{true}}}$ is a modeling mismatch rather than an optimization instability.
>
>     Even with an imperfectly estimated $\lambda$, both strategies target the same biased surrogate loss, preserving the validity of our comparison. Strategy I still incurs penalty terms within our single-loop GT template due to the non-self-adjointness. We will clarify this in the revised manuscript.
>
> **Q2: Practical feasibility and overhead of the matrix construction.**
>
> **A2:** Thanks for the insightful comment. We clarify the feasibility and overhead from three perspectives:
>
> 1.  In general directed graphs, the lack of bidirectional communication breaks detailed balance, rendering local Metropolis-Hastings (MH) construction (the standard paradigm of building a doubly stochastic matrix [4-5]) inapplicable for both strategies [6]. Constructing mixing matrices with prescribed stationary distributions on directed topologies is non-trivial, requiring online paradigms like Push-Sum or Pull-Diag to converge via general singly stochastic matrices [6]. Furthermore, directed topologies induce non-reversible dynamics, destroying the self-adjointness essential to our current analysis. Extending to non-reversible Markov chains (via pseudo-spectra or complex eigenvalue) is beyond our current scope.
>
> 2.  On undirected networks, building our $\lambda$-induced row-stochastic matrix is just as feasible as constructing a doubly stochastic one. Our modified MH rule in Eq. (3) supports both: setting $\lambda \equiv \mathbf{1}$ recovers the standard doubly stochastic matrix, while using the prescribed weights yields our $\lambda$-induced row-stochastic matrix.
>
> 3.  Finding the row-stochastic matrix $W$ with $\lambda^\top W = \lambda^\top$ incurs negligible overhead, as it is computed offline and locally:
>
>     - Communication: Each node sends its weight $\lambda_i$ and its degree $d_i$ to its neighbors. That's one extra float per neighbor compared to the standard doubly stochastic setup.
>
>     - Computation: The edge weight $W_{i,j}$ is $\frac{1-\epsilon}{d_i} \min(1, \frac{\lambda_j d_i}{\lambda_i d_j})$. That's two extra scalar multiplications per edge compared to the uniform case.
>
> **Q3: On the lower bound of the convergence rate.**
>
> **A3:** Thanks for the insightful comment. Since similar questions were also raised by other reviewers, please see our response A3 to Reviewer Bouw. In short, within the standard single-loop GT framework, the gap between the two strategies is an unavoidable consequence of the consensus step. Establishing lower bounds would require modifying the algorithm further, such as multiple communication rounds, acceleration, or specific topologies.
>
> **References**
>
> [1] Communication-efficient learning of deep networks from decentralized data
>
> [2] Variance-reduced stochastic learning by networked agents under random reshuffling
>
> [3] DICE: Data influence cascade in decentralized learning
>
> [4] A unified theory of decentralized sgd with changing topology and local updates
>
> [5] An improved analysis of gradient tracking for decentralized machine learning
>
> [6] Achieving linear speedup and near-optimal complexity for decentralized optimization over row-stochastic networks

---

> > ### Author Rebuttal · Reviewer_g7Yd · 2026-04-03
> >
> > The rebuttal satisfactorily addresses my main concern regarding weight mismatch, clarifying that it introduces a bias towards a surrogate objective without affecting convergence rates. The discussion on constructing row-stochastic matrices in undirected settings is also helpful and improves the practical perspective.
> >
> > However, some issues remain only partially addressed. In particular, there is still no formal lower bound separating the two strategies, and the analysis relies on static weights and undirected graphs.
> >
> > Overall, my key concern is resolved, but some limitations remain, so my score remains unchanged.

---

> > > ### Author Response · Authors · 2026-04-06
> > >
> > > We thank the reviewer for the positive feedback and for acknowledging that our clarifications regarding weight mismatch and matrix construction have resolved the main concern. Your constructive comments have significantly improved the quality of our paper.
> > >
> > > The remaining points, namely formal lower bounds, dynamic weights, and directed graphs, are important and challenging open problems. However, addressing them would require moving beyond our current standard single-loop gradient tracking framework. Within this setting, our comparison remains fair: both strategies target the same objective, and the advantage of Strategy II is structural, arising from self-adjointness in the weighted space. We view these as important future directions, and will explicitly discuss them in the revised manuscript.

---

### Decision · Program_Chairs · 2026-04-30

**Decision:**

Accept (regular)

**Comment:**

This paper studies decentralized learning with a weighted global loss involving heterogeneous node weights $\lambda$. It develops two strategies to handle these weights: (i) embedding them into the local losses to retain uniform weights (and thus a doubly stochastic matrix), and (ii) keeping the original losses while employing a $\lambda$-induced row-stochastic matrix. The authors establish the convergence rates of these two strategies and identify conditions under which the row-stochastic design converges faster, even with a smaller spectral gap.

Overall, the paper provides a deeper understanding of row-stochastic matrices in decentralized learning. Its contributions are interesting and may benefit further developments in this area.

One concern, raised by Reviewer aBHn, is whether the optimal solution changes when using a $\lambda$-induced row-stochastic matrix. The authors responded to this concern during the rebuttal phase, and it was further discussed among the reviewers. However, Reviewer aBHn remained unconvinced by the rebuttal and the subsequent discussion.

After carefully reading the paper, the reviews, and the discussion, I find that using a different communication matrix does not affect convergence. In particular, the paper establishes convergence in terms of the stationary point $|\nabla F(\theta)|$ of the original loss function in Theorem 6.5. This is sufficient to guarantee that both communication strategies converge to a stationary point of the original objective.

I agree that different communication matrices may alter the optimization trajectory. However, regardless of the trajectory taken, the key question is whether the algorithm converges to a stationary point of the original problem. Theorem 6.5 confirms that the proposed algorithm does ensure this. Therefore, I do not believe the issue raised by Reviewer aBHn is valid.

An analogy may help illustrate this point. In gradient descent, various gradient estimators can be used, such as standard stochastic gradients, momentum-based methods, and variance-reduced techniques. Some of these estimators are even biased approximations of the full gradient, and different choices can lead to different optimization trajectories. However, what ultimately matters is whether they converge to a stationary point of the original objective. If they do, they effectively solve the original problem. A similar principle applies to the choice of communication matrices in decentralized learning: regardless of the specific matrix used, if the method converges to a stationary point of the original objective, then it successfully solves the problem.

Considering that this paper provides new insights into decentralized learning and presents a novel convergence analysis, I recommend acceptance.